# Oligomer formation from the gas-phase reactions of Criegee intermediates with hydroperoxide esters: mechanism and kinetics

Long Chen,[1,2] Yu Huang,*[1,2] Yonggang Xue,[1,2] Zhihui Jia,[3] Wenliang Wang[4]

[1] *State Key Lab of Loess and Quaternary Geology (SKLLQG), Institute of Earth Environment, Chinese Academy of Sciences (CAS), Xi'an, 710061, China*

[2] *CAS Center for Excellence in Quaternary Science and Global Change, Xi'an, 710061, China*

[3] *School of Materials Science and Engineering, Shaanxi Normal University, Xi'an, Shaanxi, 710119, China*

[4] *School of Chemistry and Chemical Engineering, Key Laboratory for Macromolecular Science of Shaanxi Province, Shaanxi Normal University, Xi'an, Shaanxi, 710119, China*

Submitted to *Atmospheric Chemistry & Physics*

*Corresponding author:

Prof. Yu Huang, E-mail address: huangyu@ieecas.cn

**Abstract**

Hydroperoxide esters, formed in the reactions of carbonyl oxides (also called Criegee intermediates, CIs) with formic acid, play a crucial role in the formation of secondary organic aerosol (SOA) in the atmosphere. However, the transformation mechanism of hydroperoxide esters in the presence of stabilized Criegee intermediates (SCIs) is not well understood. Herein, the oligomerization reaction mechanisms and kinetics of distinct SCIs ($CH_2OO$, *syn*-$CH_3CHOO$, *anti*-$CH_3CHOO$ and $(CH_3)_2COO$) reactions with their respective hydroperoxide esters as well as with hydroperoxymethyl formate (HPMF) are investigated in the gas phase using quantum chemical and kinetics modeling methods. The calculations show that the addition reactions of SCIs with hydroperoxide esters proceed through successive insertion of SCIs into hydroperoxide ester to form oligomers that involve SCIs as the repeated chain unit. The exothermicity of oligomerization reactions significantly decreases when the number of methyl substituents increases, and the exothermicity of *anti*-methyl substituted carbonyl oxides is obviously higher than that of *syn*-methyl substituted carbonyl oxides. The –OOH insertion reaction is energetically more feasible than the –CH insertion pathway in the SCIs oligomerization reactions, and the barrier heights increase with increasing the number of SCIs added to the oligomer except *syn*-$CH_3CHOO$. For the reactions of distinct SCIs with HPMF, the barrier of -OOH insertion pathway shows a dramatic decrease when a methyl substituent occurs at the *anti*-position, while it reveals a significant increase when a methyl group is introduced at the *syn*-position and dimethyl substituent. Compared with the rate coefficients of the $CH_2OO$ + HPMF reaction, the rate coefficients increase by about one order of magnitude when a methyl substituent occurs at the *anti*-position, whereas the rate coefficients decrease by 1-2 orders of magnitude when a methyl group is introduced at the *syn*-position. These new findings advance our current understanding on the influence of Criegee-chemistry on the formation processes and chemical compositions of SOA.

# 1. Introduction

Alkenes are an important class of volatile organic compounds (VOCs) that are emitted into the atmosphere from large quantities of biogenic and anthropogenic sources (Lester and Klippenstein, 2018). The reaction with ozone is one of the dominant degradation pathways for alkenes in the atmosphere (Johnson and Marston, 2008; Atkinson and Arey, 2003). Ozonolysis of alkene proceeds through the electrophilic 1,3-cycloaddition of ozone to C=C bond of alkenes to form a primary ozonides (POZ), and then it rapidly decomposes into a carbonyl compound and a carbonyl oxide (also called Criegee intermediates, CIs) (Criegee, 1975; Osborn and Taatjes, 2015; Giorio et al., 2017). A part of the initially energized CIs (~ 37-50%) may promptly dissociate to OH radicals, which are thought to be an important nonphotolytic source of OH radicals in the atmosphere (Novelli et al., 2014; Liu et al., 2014). The remaining CIs (~ 63-50%) are collisionally stabilized prior to the thermal unimolecular decay (Lester and Klippenstein, 2018; Novelli et al., 2014; Anglada and Solé, 2016). The stabilized Criegee intermediates (SCIs) can proceed bimolecular reactions with various trace species such as $H_2O$, $NO_2$, $SO_2$, and HCOOH to generate secondary organic aerosol (SOA), thus profoundly influencing air quality, global climate and human health (Osborn and Taatjes, 2015; Khan et al., 2018; Lin and Chao, 2017; Liu et al., 2019; Chhantyal-Pun et al., 2018; Gong and Chen, 2021; Taatjes, 2017).

Formic acid (HCOOH), one of the most abundant carboxylic acids, has a significant influence on rainwater acidity in remote areas, where pH reduces by 0.25-0.5 in the presence of HCOOH (Stavrakou et al., 2012; Wang et al., 2020; Chaliyakunnel et al., 2016). It also plays an important role in the formation of cloud condensation nuclei (CCN), indirectly influencing radiative forcing and climate change (Yu, 2000). The primary sources of HCOOH include biomass burning, human activities, tropical and boreal forests, as well as the secondary sources involve the photochemical oxidation of non-methane hydrocarbons, such as ketene-enols, vinyl alcohol, isoprene, and terpenoids (Stavrakou et al., 2012; Wang et al., 2020; Chaliyakunnel et al., 2016; So et al., 2014; Paulot et al., 2011). According to satellite measurements, the production

of HCOOH is up to 100-120 Tg yr$^{-1}$, and the value is expected to increase due to the acceleration of industrialization and urbanization (Stavrakou et al., 2012). Recent kinetics measurements have revealed that the reaction with HCOOH is a more important loss process for SCI than is presently assumed, especially in terrestrial equatorial areas and in high SCI concentration areas (Welz et al., 2014; Chung et al., 2019). The formed hydroperoxide esters have been identified as the low-volatility and high-oxygenated compounds, contributing to the formation and growth of SOA (Welz et al., 2014; Vansco et al., 2021; Sakamoto et al., 2017; Riva et al., 2017).

Welz et al. (2014) directly determined the rate coefficients for the reactions of $CH_2OO$ and $CH_3CHOO$ with formic and acetic acid by employing multiplexed photoionization mass spectrometry and cavity-enhanced broadband ultraviolet absorption spectroscopy. They found that the measured rate coefficients are in the excess of $1.0 \times 10^{-10}$ cm$^3$ molecule$^{-1}$ s$^{-1}$, which are several orders of magnitude greater than those derived from previous experimental studies (Johnson et al., 2001; Tobias and Ziemann, 2001). Sipilä et al. (2014) conducted a competitive reaction kinetics experiment to investigate the reactions of acetone oxide (($CH_3$)$_2$OO) with $SO_2$, HCOOH and $CH_3COOH$, and they concluded that the rate coefficients of the ($CH_3$)$_2$OO + HCOOH/$CH_3OOH$ reactions are faster than that of the ($CH_3$)$_2$OO + $SO_2$ system by about three times. These high rate coefficients could make the reaction with carboxylic acids a substantial dominant chemical sink for carbonyl oxides in the atmosphere (Welz et al., 2014; Taatjes et al., 2019; Chhantyal-Pun et al., 2017). Quantum chemical calculations show that the reaction of $CH_2OO$ with HCOOH proceeds through a facile transfer of hydrogen atom from the acidic OH group to the terminal oxygen of $CH_2OO$ to form hydroperoxymethyl formate (HPMF) (Long et al., 2009; Vereecken, 2017; Porterfield et al., 2019). Chen et al. (2018) concluded the same by investigating the reactions of various carbonyl oxides with HCOOH that the barrierless 1,4-insertion reaction is the most favorable pathway, and the primary products are hydroperoxide esters. Caravan et al. (2020) employed high-level ab initio CCSD(T)-F12 methods to study the reaction of methyl vinyl ketone oxide (MVK-oxide) with HCOOH, and they

found that the barrierless net insertion of MVK-oxide into HCOOH leading to the formation of a functionalized hydroperoxide is dominant over fragmentation to produce an alkoxy radical and OH radicals. Moreover, oligomerization reactions with hydroperoxides and peroxy radicals are identified as one of the dominant loss processes for carbonyl oxides under atmospheric conditions (Sakamoto et al., 2013; Sadezky et al., 2008; Zhao et al., 2015; Chen et al., 2017 and 2019). All the above milestone investigations provide important information for understanding the chemistry of Criegee intermediate in the presence of carboxylic acids. However, to the best of our knowledge, there are few studies on the oligomerization reactions of SCIs with hydroperoxide esters, which are important with regard to organic new particle and cloud condensation nuclei formations. Moreover, the relationship between the reactivity of SCIs and the nature of substituents remains uncertain in the SCIs oligomerization reactions.

In the present study, we mainly focus on the oligomerization reaction mechanisms and kinetics of four carbonyl oxides reactions with their respective hydroperoxide esters as well as with HPMF by employing quantum chemical calculations and kinetics modeling methods. For the initiation reactions of carbonyl oxides with formic acid, four kinds of pathways including 1,4 O-H insertion, 1,2 O-H insertion, C-H insertion, and C=O cycloaddition are considered. For the oligomerization reactions of the successive insertion of carbonyl oxides into hydroperoxide esters, two types of reactions involving –OOH and –CH insertions are taken into account. The selected carbonyl oxides, including $CH_2OO$, *syn*-, *anti*-$CH_3CHOO$ and $(CH_3)_2CHOO$, are anticipated upon the ozonolysis of ethylene, propylene, and 2,3-dimethyl-2-butene, whereas the hydroperoxide esters are assumed to arise from the bimolecular reactions of carbonyl oxides with formic acid in the atmosphere.

## 2. Computational details

### 2.1 Electronic structure and energy calculations

The geometries of all stationary points, including reactants (R), intermediates (IM), transition states (TS), and products (P), are optimized at the M06-2X/6-311+G(2df,2p)

level of theory, since the M06-2X functional has the reliable performance for predicting thermochemistry, kinetics and hydrogen bonding interactions (Zhao and Truhlar, 2008). Harmonic vibrational frequencies are performed at the same level to verify the nature of transition state (NIMAG = 1) and minimum (NIMAG = 0), and to provide zero-point vibrational energy (ZPVE) and Gibbs free energies corrections ($G_{corr}$), which are scaled by a factor of 0.98 (Alecu et al., 2010). Intrinsic reaction coordinate (IRC) calculations are carried out to verify that each transition state is connected to the desired reactant and product (Fukui, 1981). The single point energy (SPE) calculations are performed at the M06-2X/ma-TZVP level of theory based on the M06-2X/6-311+G(2df,2p) optimized geometries. Moreover, the basis set superposition error (BSSE) is performed by using the counterpoise method proposed by Boys and Bernardi (1970) to evaluate the stability of the pre-reactive complex (RC). Herein, the Gibbs free energy ($G$) is defined as the sum of SPE and Gibbs correction ($G = E + G_{corr}$). Electronic energy ($\Delta E^{\neq}$) and Gibbs free energy ($\Delta G^{\neq}$) barriers are defined as the difference in energy between a TS and a RC ($\Delta E^{\neq} = E_{TS} - E_{RC}$ and $\Delta G^{\neq} = G_{TS} - G_{RC}$). Reaction Gibbs free energy ($\Delta G$) is defined as the difference in energy between a P and a R ($\Delta G = G_P - G_R$).

To further assess the reliability of the selected M06-2X/ma-TZVP method for SPE calculations, the single point energies of all stationary points involved in the initiation reactions of distinct SCIs with HCOOH are recalculated at the high-accuracy CCSD(T)/6-311+G(2df,2p) and QCISD(T)/6-311+G(2df,2p) levels of theory. The calculated results are summarized in Table S1. This table shows that the $\Delta E^{\neq}$ and $\Delta G^{\neq}$ obtained using the QCISD(T) method are in excellent agreement with those obtained using the CCSD(T) approach. It is therefore that the energies obtained using the CCSD(T) method are used as the benchmark for comparison. The mean absolute deviations (MAD) of $\Delta E^{\neq}$ and $\Delta G^{\neq}$ between the CCSD(T) and M06-2X methods are 0.43 and 0.41 kcal mol$^{-1}$, respectively; the largest deviations of $\Delta E^{\neq}$ and $\Delta G^{\neq}$ are 1.0 and 1.1 kcal mol$^{-1}$, respectively. These results reveal that the energies obtained using the M06-2X method are close to those obtained using the CCSD(T) approach. Therefore, the M06-2X/ma-TZVP method is suitable to investigate the SCIs

oligomerization reactions. In the following discussion, the energies are applied in terms of Gibbs free energy to describe the reaction mechanism unless otherwise stated. All electronic structure calculations are carried out by using Gaussian 09 program (Frisch et al., 2009). The Multiwfn program and Visual molecular dynamics (VMD) are utilized to analysis and visualize the molecular orbitals of the relevant species (Lu and Chen, 2012).

## 2.2 Kinetics calculations

The rate coefficients for the barrierless reactions are determined by employing the inverse Laplace transformation (ILT) method. The ILT calculations are performed with the MESMER 6.0 program (Glowacki et al., 2012). In the ILT treatment, the rotational constants, vibrational frequencies, molecular weights, energies and other input parameters are obtained from the M06-2X/6-311+G(2df,2p) or M06-2X/ma-TZVP methods. For the barrierless reaction of 1,4 O-H insertion of SCIs into HCOOH, SCIs and HCOOH are assigned as the deficient and excess reactants, respectively. The concentration of HCOOH is given a value of $5.0 \times 10^{10}$ molecules $cm^{-3}$ in the simulation, which is taken from the typical concentration of HCOOH in the tropical forest environments (Vereecken et al., 2012). $N_2$ is applied as the buffer gas. A single exponential down model is employed to simulate the collision transfer ($<\Delta E>_{down} = 200$ $cm^{-1}$). The collisional Lennard-Jones parameters are estimated with the empirical formula described by Gilbert and Smith (1990).

The rate coefficients for the bimolecular reactions with the tight transition states are calculated by using the canonical transition state theory (CTST) along with one-dimensional asymmetric Eckart tunneling correction (Truhlar et al., 1996; Eckart, 1930). The CTST/Eckart calculations are performed with the KiSThelP 2019 program (Canneaux et al., 2013). As shown in Fig. 1, the entrance pathway Entry2 of $R_1R_2COO$ reaction with HCOOH consists of two steps: (i) an intermediate IMent2 is formed via a barrierless process; (ii) then, it rearranges to the product Pent2 through a tight transition state TSent2. The whole reaction process can be described as Eq. (1):

$$\text{R}_1\text{R}_2\text{COO} + \text{HCOOH} \underset{k_{-1}}{\overset{k_1}{\rightleftharpoons}} \text{IMent2} \xrightarrow{k_2} \text{Pent2} \tag{1}$$

Assuming the rapid equilibrium is established between the IMent2 and reactants. According to the steady-state approximation (SSA), the total rate coefficient is approximately expressed as Eq. (2) (Zhang et al., 2012):

$$k_{\text{tot}} = \frac{k_1}{k_{-1} + k_2} k_2 \approx \frac{k_1}{k_{-1}} k_2 = K_{\text{eq}} k_2 \tag{2}$$

The equilibrium constant $K_{\text{eq}}$ is written as Eq. (3):

$$K_{\text{eq}} = \sigma \frac{Q_{\text{IM}}(T)}{Q_{\text{R1}}(T)Q_{\text{R2}}(T)} \exp\left(\frac{G_{\text{R}} - G_{\text{IM}}}{RT}\right) \tag{3}$$

where $\sigma$ refers to reaction symmetry number, $Q_{\text{IM}}(T)$, $Q_{\text{R1}}(T)$ and $Q_{\text{R2}}(T)$ denote the partition functions of intermediate, reactants R1 and R2, which are equal to the multiplication of translational, rotational, vibrational and electronic partition functions ($Q = Q_{\text{rot}}Q_{\text{vib}}Q_{\text{trans}}Q_{\text{elec}}$) (Mendes et al., 2014), $T$ is the temperature in Kelvin, $R$ is the ideal gas constant, $G_{\text{R}}$ and $G_{\text{IM}}$ are the total Gibbs free energies of reactant and intermediate, respectively.

## 3. Results and discussion

## 3.1 Initiation reactions of distinct SCIs with HCOOH

The reaction with HCOOH is one of the dominant loss processes for SCIs and is expected to trigger the formation of SOA in the atmosphere (Chhantyal-Pun et al., 2018; Cabezas and Endo, 2020; Zhao et al., 2018; Zhou et al., 2019). The potential energy surface (PES) of distinct SCIs ($CH_2OO$, *syn*-, *anti*-$CH_3CHOO$ and $(CH_3)_2COO$) reactions with HCOOH is drawn in Fig. 1. The geometries of all stationary points are displayed in Fig. S1. The relative free energy of each stationary point and free energy barrier ($\Delta G^{\neq}$) of each elementary reaction are summarized in Table 1. As shown in Fig. 1, the bimolecular reaction of distinct SCIs with HCOOH proceeds via four possible pathways, namely (1) 1,4 O-H insertion (Entry 1), (2) 1,2 O-H insertion (Entry 2), (3) C-H insertion (Entry 3), and (4) C=O cycloaddition (Entry 4). For Entry 1, the addition reaction of $CH_2OO$ with HCOOH proceeds through the 1,4 O-H insertion of $CH_2OO$

into HCOOH to form a hydroperoxide ester HC(O)O-CH$_2$OO-H with a exoergicity of
37.6 kcal·mol$^{-1}$. The formation of HC(O)O-CH$_2$OO-H is obtained through a concerted
process of O$_2$-H$_2$ bond breaking in the HCOOH and O$_4$-H$_2$ and C$_2$-O$_1$ bonds forming.
Despite an attempt by various methods, the corresponding transition state is still not
located in the effort of optimization. To further validate the barrierless process of 1,4
O-H insertion reaction, a relaxed scan over the O$_4$-H$_2$ and C$_2$-O$_1$ bonds is performed at
the M06-2X/6-311+G(2df,2p) level of theory. The scans start from the optimized
structure of the adduct product HC(O)O-CH$_2$OO-H, and the O$_4$-H$_2$ and C$_2$-O$_1$ bond
length are then increased in steps of 0.10 Å. The relaxed scan energy profiles are
presented in Fig. S2. As seen in Fig. S2a, the relative energy of the minimum energy
path from reactant to product decreases monotonically when the bond length of O$_4$-H$_2$
and C$_2$-O$_1$ bonds decreases, suggesting that the transition state is not exist in the 1,4 O-
H insertion reaction of CH$_2$OO with HCOOH. Similar conclusion is also obtained from
the relaxed scan energy profiles for the HCOOH + *anti*-CH$_3$CHOO, HCOOH + *syn*-
CH$_3$CHOO and HCOOH + (CH$_3$)$_2$COO (Fig. S2b-d) reactions that 1,4 O-H insertion
reactions are barrierless. This conclusion is further supported by the analogous reaction
systems that 1,4 O-H insertion reactions of carbonyl oxides with carboxylic acids are a
barrierless process including concerted hydrogen atom transfer and new C-O bond
formation (Chhantyal-Pun et al., 2017; Long et al., 2009; Vereecken, 2017; Cabezas
and Endo, 2019; Lin et al., 2019).

The exothermicity of 1,4 O-H insertion reactions of distinct SCIs with HCOOH is

assessed by the reaction enthalpy ($\Delta_r H^{\circ}_{298}$), which is defined as the difference between
the enthalpies of formation ($\Delta_f H^{\circ}_{298}$) of the products and reactants
($\Delta_r H^{\circ}_{298} = \sum_{products} \Delta_f H^{\circ}_{298} - \sum_{reactants} \Delta_f H^{\circ}_{298}$). To the best of our knowledge, there are no literature
values available on the enthalpies of formation of carbonyl oxides and hydroperoxide
esters except the simplest carbonyl oxide CH$_2$OO. Therefore, the isodesmic reaction
method is adopted to obtain the enthalpies of formation, and the results are listed in
Table S2. An isodesmic reaction is a hypothetical reaction, in which the type of
chemical bonds in the reactants is the similar as that of chemical bonds in the products.
The following isodesmic reaction is constructed because the experimental values of $H_2$,
$CH_4$ and $H_2O$ are available ( $\Delta_f H^{\circ}_{298}$ ($H_2$) = 0.00 kcal·mol$^{-1}$; $\Delta_f H^{\circ}_{298}$ ($CH_4$) = -17.82
kcal·mol$^{-1}$; $\Delta_f H^{\circ}_{298}$ ($H_2O$) = -57.79 kcal·mol$^{-1}$).

$SCIs + nH_2 \rightarrow CH_4 + mH_2O$                               (4)

As seen in Table S2, the enthalpy of formation of $CH_2OO$ is calculated to be 23.23
kcal·mol$^{-1}$, which is in good agreement with the available literature values (Chen et al.,
2016; Karton et al., 2013). This result implies that the theoretical method employed
herein is reasonable to predict the thermochemical parameters. The enthalpies of
formation of carbonyl oxides and hydroperoxide esters significantly decrease with
increasing the number of methyl groups. Notably, the decreased values in the enthalpies
of formation of carbonyl oxides are greater than those of hydroperoxide esters under
the condition of the same number of methyl groups. For example, the enthalpy of
formation of *anti*-$CH_3CHOO$ decreases by 12.95 kcal·mol$^{-1}$ compared to the enthalpy
of formation of $CH_2OO$, and the enthalpy of formation of Pent1b decreases by 12.12
kcal·mol$^{-1}$ compared to the enthalpy of formation of Pent1a. The reaction enthalpies
decrease in the order of -44.69 ($CH_2OO$ + HCOOH → Pent1a) < -43.86 (*anti*-
$CH_3CHOO$ + HCOOH → Pent1b) < -38.13 (*syn*-$CH_3CHOO$ + HCOOH → Pent1c) < -
37.12 kcal·mol$^{-1}$ (($CH_3$)$_2$COO + HCOOH → Pent1d), indicating that the reaction
enthalpies are highly dependent on the number and location of methyl groups. The trend
in reaction enthalpies is consistent with the trend in the enthalpies of formation of
carbonyl oxides. The reason might be attributed to the decreased values in the enthalpies
of formation of carbonyl oxides greater than those of hydroperoxide esters under the
condition of the same number of methyl groups.
For Entry 2, each addition reaction starts with the formation of a pre-reactive
hydrogen bonded complex IMent2 in the entrance channel. Then it immediately
converts into product Pent2 through the 1,2 O-H insertion transition state. The
formation of Pent2 is obtained via a concerted process of $O_2$-$H_2$ bond rupture in the
HCOOH and $O_4$-$H_2$ and $C_2$-$O_2$ bonds forming. The reaction barrier $\Delta G^{\neq}$ increases in
the order of 10.0 ($CH_2OO$) < 13.0 (*anti*-$CH_3CHOO$) < 14.6 (*syn*-$CH_3CHOO$) $\approx$ 14.4
(($CH_3$)$_2$$COO$) kcal·mol$^{-1}$, suggesting that the parent $CH_2OO$ + HCOOH reaction is
favored kinetically. Compared with the barrier of the parent system, the barrier
increases by 3.0 kcal·mol$^{-1}$ when a methyl substitution occurs at the $R_1$ position, and
the barrier increases by ~ 5 kcal·mol$^{-1}$ when a methyl group is introduced at the $R_2$
position and dimethyl substituent. The aforementioned result implies that the methyl-
substituted $CH_2OO$ hinders the 1,2 O-H insertion of carbonyl oxides into formic acid.
Notably, the exothermicity decreases significantly as the number of methyl group is
increased. The products Pent1 and Pent2 formed from Entry 1 and 2 are two
conformations that differ in the orientation of the –C(O)H moiety over the –OOH group.
The calculated result shows that Pent1 is more stable than Pent2 in energy due to the
existence of intramolecular hydrogen bond between hydrogen atom of –OOH group
and carbonyl oxygen atom.
For Entry 3, the addition reaction begins with the formation of a pre-reactive
complex IMent3 in the entrance channel, and then it surmounts a barrier to reaction.
However, the barriers of C-H insertion reactions are significantly high (21.8-27.6
kcal·mol$^{-1}$), such that they are of less importance in the atmosphere. The high reaction
barriers might be attributed to the large bond dissociation energy (BDE) of C-H bond
in the formic acid. For Entry 4, the addition reaction proceeds through a cyclization
process of $C_2$-$O_1$ and $O_4$-$C_1$ bond forming to produce a five-membered ring compound
Pent4. The barrier of C=O cycloaddition reaction in the $CH_2OO$ + HCOOH reaction is
5.8 kcal·mol$^{-1}$, which is lower than that of the corresponding channels in Entry 2 and
Entry 3 by 4.2 and 16.0 kcal·mol$^{-1}$, respectively. The result reveals that the C=O
cycloaddition reaction is feasible kinetically. A similar conclusion is also obtained from
the reactions of HCOOH with *syn*-, *anti*-$CH_3CHOO$ and ($CH_3$)$_2$$COO$ that the C=O
cycloaddition reactions are favored over 1,2 O-H and C-H insertion reactions.
The rate coefficients of each elementary pathway included in the initiation
reactions of distinct SCIs with HCOOH are calculated in the temperature range of 273-
400 K, as listed in Table S3-S6. As shown in Table S3, the total rate coefficients $k_{\text{tot-}}$
$_{\text{CH2OO}}$ of $CH_2OO$ reaction with HCOOH are in excess of $1.0 \times 10^{-10}$ $cm^3$ molecule$^{-1}$ s$^{-1}$,
and they exhibit a slightly negative temperature dependence in the temperature range
studied. $k_{\text{tot-CH2OO}}$ is estimated to be $1.4 \times 10^{-10}$ $cm^3$ molecule$^{-1}$ s$^{-1}$ at 298 K, which is in
good agreement with the experimental values reported by Welz et al. (2014) ([$1.1 \pm 0.1$]
$\times 10^{-10}$), Chung et al. (2019) ([$1.4 \pm 0.3$] $\times 10^{-10}$), and Peltola et al. (2020) ([$1.0 \pm 0.03$]
$\times 10^{-10}$). $k(\text{TS}_{\text{ent1}})$ is approximately equal to $k_{\text{tot-CH2OO}}$ in the whole temperature range,
and it decreases in the range of $1.7 \times 10^{-10}$ (273 K) to $1.2 \times 10^{-10}$ (400 K) $cm^3$ molecule$^-$
$^1$ s$^{-1}$ with increasing temperature. $k(\text{TS}_{\text{ent1}})$ is several orders of magnitude greater than
$k(\text{TS}_{\text{ent2}})$, $k(\text{TS}_{\text{ent3}})$ and $k(\text{TS}_{\text{ent4}})$ over the temperature range from 273 to 400 K. The
result again shows that the barrierless 1,4 O-H insertion reaction is predominant. The
calculated $K_{\text{eq-ent2}}$, $k_{\text{2-ent2}}$, and $k(\text{TS}_{\text{ent2}})$ ($k(\text{TS}_{\text{ent2}}) = K_{\text{eq-ent2}} \times k_{\text{2-ent2}}$) in Entry 2 are listed
in Table S7. This table shows that $K_{\text{eq-ent2}}$ significantly decreases with increasing
temperature, and $k_{\text{2-ent2}}$ increases as the temperature is increased. However, the
decreased value in $K_{\text{eq-ent2}}$ is greater than the increased value in $k_{\text{2-ent2}}$ under the same
temperature range. For example, $K_{\text{eq-ent2}}$ deceases by a factor of 6.3 and $k_{\text{2-ent2}}$ increases
by a factor of 2.9 at 298 K compared with the values of $K_{\text{eq-ent2}}$ and $k_{\text{2-ent2}}$ at 273 K. It is
therefore that $k(\text{TS}_{\text{ent2}})$ decreases with the temperature increasing. Similar conclusion
is also obtained from the results of the rate coefficients in Entry 4 that $k(\text{TS}_{\text{ent4}})$ exhibits
a negative temperature dependence in the temperature range studied (Table S8). The
aforementioned results imply that $k(\text{TS}_{\text{ent2}})$ and $k(\text{TS}_{\text{ent4}})$ are mediated by the pre-
reactive complexes IMent2 and IMent4 in the Entry 2 and 4. It should be noted that
although the barrier of Entry 2 is 4.2 kcal·mol$^{-1}$ higher than that of Entry 4, $k(\text{TS}_{\text{ent2}})$ is
merely about 1-2 fold smaller than $k(\text{TS}_{\text{ent4}})$. The reason is ascribed to the fact that the
C=O cycloaddition reaction is entropically unfavorable (Vereecken, 2017).
Equivalent to the case of $CH_2OO$ reaction with HCOOH, the rate coefficient of
each elementary pathway involved in the *anti*-$CH_3CHOO$ + HCOOH reaction also
decreases with the temperature increasing (Table S4). This table shows that Entry 1 is
kinetically favored over Entry 2, 3 and 4, and Entry 2 is competitive with Entry 4 in the

range 273-400 K. Similar conclusion is also obtained from the results of the rate coefficients for the reactions of *syn*-CH$_3$CHOO and (CH$_3$)$_2$COO with HCOOH that Entry 1 is the dominant pathway (Table S5-S6). It deserves mentioning that the competition of Entry 2 is significantly greater than that of Entry 4 in the *syn*-CH$_3$CHOO + HCOOH and (CH$_3$)$_2$COO + HCOOH systems. Based on the above discussions, it can be concluded that the relative importance of different pathways is highly dependent on the number and location of methyl substituents in the carbonyl oxides. Notably, the rate coefficient of each elementary pathway included in the *anti*-CH$_3$CHOO + HCOOH reaction is several orders of magnitude greater than that of the corresponding channel involved in the other SCIs + HCOOH systems. It is because that *anti*-CH$_3$CHOO is substantially more reactive toward HCOOH than other SCIs. Similar phenomenon has also observed from the reactivity of *anti*-CH$_3$CHOO toward water and SO$_2$ (Taatjes et al., 2013; Long et al., 2016; Huang et al., 2015; Cabezas and Endo, 2018). At ambient temperature, the total rate coefficients of HCOOH reactions with *anti*-CH$_3$CHOO, *syn*-CH$_3$CHOO and (CH$_3$)$_2$COO are estimated to be 5.9, 2.7 and 4.8 × 10$^{-10}$ cm$^3$ molecule$^{-1}$ s$^{-1}$, respectively, which are consistent with the prior experimental measurements of 5 ± 3, 2.5 ± 0.3 and 4.5 × 10$^{-10}$ cm$^3$ molecule$^{-1}$ s$^{-1}$ (Welz et al., 2014; Chung et al., 2019; Sipilä et al., 2014).

In summary, the barrierless 1,4 O-H insertion reaction is the dominant pathway in the initiation reactions of distinct SCIs with HCOOH. This conclusion is consistent with the recent experimental results derived from the reactions of formic acid with methacrolein oxide (MACR-OO) and methyl vinyl ketone oxide (MVK-OO) that the 1,4-addition mechanism is energetically favorable (Vansco et al., 2021; Caravan et al., 2020). Therefore, in the present study, the adduct products Pent1 formed form the barrierless 1,4 O-H insertion of carbonyl oxides into HCOOH are selected as the model compounds to investigate the oligomerization reaction mechanisms of carbonyl oxides reactions with hydroperoxide esters.

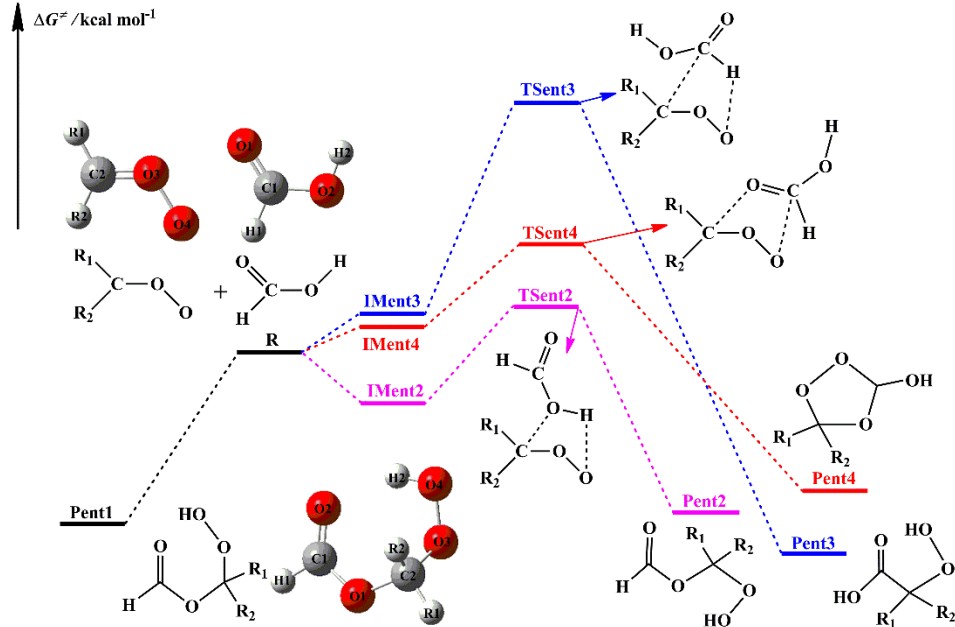

**Figure 1.** Schematic PES for the possible entrance pathways of the initiation reactions of HCOOH with various SCIs (black, pink, blue, and red lines represent 1,4 O-H insertion, 1,2 O-H insertion, C-H insertion, and C=O cycloaddition reactions, respectively)

**Table 1** Relative free energies of stationary points and free-energy barriers ($\Delta G^{\neq}$) at 298 K in kcal mol$^{-1}$ for the various SCIs ($R_1R_2COO$, $R_1$, $R_2$=H, $CH_3$) reactions with HCOOH calculated at the M06-2X/ma-TZVP//M06-2X/6-311+G(2df,2p) level of theory

| Entry | R1 | R2 | IMent | TSent | Pent | $\Delta G^{\neq}$ |
|-------|------|------|-------|-------|-------|-------|
| 1 | H | H | – | – | -37.6 | – |
|   | CH$_3$ | H | – | – | -34.0 | – |
|   | H | CH$_3$ | – | – | -29.8 | – |
|   | CH$_3$ | CH$_3$ | – | – | -25.6 | – |
| 2 | H | H | -3.1 | 6.9 | -37.3 | 10.0 |
|   | CH$_3$ | H | -11.0 | 2.0 | -33.7 | 13.0 |
|   | H | CH$_3$ | -6.6 | 8.0 | -29.1 | 14.6 |
|   | CH$_3$ | CH$_3$ | -8.8 | 5.6 | -24.9 | 14.4 |
| 3 | H | H | 3.4 | 25.2 | -46.9 | 21.8 |
|   | CH$_3$ | H | 1.8 | 24.0 | -41.5 | 22.2 |
|   | H | CH$_3$ | 3.0 | 30.6 | -37.6 | 27.6 |
|   | CH$_3$ | CH$_3$ | 1.9 | 29.5 | -33.0 | 27.6 |
| 4 | H | H | 3.4 | 9.2 | -31.7 | 5.8 |
|   | CH$_3$ | H | 2.2 | 7.8 | -29.4 | 5.6 |
|   | H | CH$_3$ | 3.5 | 14.6 | -25.3 | 11.1 |
|   | CH$_3$ | CH$_3$ | 3.0 | 13.2 | -22.9 | 10.2 |

**3.2. The reactions of distinct SCIs with their respective hydroperoxide**

**esters**

The formed hydroperoxide ester has two possible unimolecular decay pathways. The first is the direct O-O bond rupture resulting in the formation of oxylmethylformate and OH radicals (Vereecken, 2017). The second is the –OH fragment binding to adjacent hydrogen atom leading to the formation of anhydride and $H_2O$ (Aplincourt and Ruiz-López, 2000; Neeb et al., 1998). However, the barriers of these two unimolecular reactions are extremely high, such that they are of less importance in the atmosphere. The formed hydroperoxide ester possess –OOH and –OC(O)H groups, both of them can serve as the reactive moieties to react with carbonyl oxides giving rise to the formation of oligomers. In the present study, we mainly consider two types of pathways: (a) –OOH insertion and (b) –CH insertion, while the C=O cycloaddition reaction is not taken into account because it is entropically unfavorable (Vereecken, 2017; Lin, et al., 2019). The aforementioned reactions are discussed in detail in the following subsections.

### 3.2.1 The reactions of 2CH₂OO with Pent1a

The simplest carbonyl oxide, $CH_2OO$, originates from the reaction of all terminal alkenes with ozone (ozonolysis) in the atmosphere (Lin and Chao, 2017). The reaction with HCOOH is expected to be one of the dominant loss processes for $CH_2OO$, and the main product is Pent1a (also called as HPMF) (Welz et al., 2014; Cabezas and Endo, 2019). A schematic PES for the addition reaction $2CH_2OO$ + Pent1a is drawn in Fig. 2, and the optimized geometries of all stationary points are displayed in Fig. S3. As seen in Fig. 2, the successive insertion of $CH_2OO$ into Pent1a eventually leads to the formation of oligomers P2a and P2b composed of $CH_2OO$ as the repeat unit. These oligomerization reactions are strongly exothermic and spontaneous ($> 83$ kcal·mol$^{-1}$), implying that they are feasible thermodynamically.

The addition reaction $2CH_2OO$ + Pent1a initially proceeds through two possible pathways, namely (1) –OOH insertion reaction R1a, and (2) –CH insertion reaction R1b. For the –OOH insertion reaction R1a, the pre-reactive intermediate IM1a with a seven-membered ring structure is formed in the entrance channel, which is stabilized by the hydrogen bond interactions between the $H_4$ atom of Pent1a and the $O_6$ atom of $CH_2OO$

($D_{(O6-H4)}$ = 1.706 Å), and between the $H_6$ atom of $CH_2OO$ and the $O_3$ atom of Pent1a
($D_{(O3-H6)}$ = 2.115 Å). Then IM1a converts into P1a ($C_3H_6O_6$, HC(O)O–$(CH_2OO)_2$–H)
via a concerted process of $O_4$-$H_4$ bond breaking in the Pent1a and $O_4$-$C_3$ and $H_4$-$O_6$
bonds forming with a barrier of 8.1 kcal·mol$^{-1}$. For the –CH insertion reaction R1b, the
pre-reactive intermediate IM1b with a seven-membered ring structure is formed in the
entrance channel, which is stabilized by the van der Waals (vdW) interactions between
the $O_3$ atom of Pent1a and the $C_3$ atom of $CH_2OO$ ($D_{(O3-C3)}$ = 2.602 Å), and between
the $O_6$ atom of $CH_2OO$ and the $C_1$ atom of Pent1a ($D_{(O6-C1)}$ = 2.608 Å). Due to the
absence of hydrogen bond in IM1b, the energy of IM1b is lower than that of IM1a by
3.0 kcal·mol$^{-1}$. IM1b transforms into P1b ($C_3H_6O_6$, $HO_2CH_2OC(O)CH_2OOH$) via a
concerted process of $C_1$-$H_1$ bond breaking in the Pent1a and $C_1$-$C_3$ and $H_1$-$O_6$ bonds
forming with a barrier of 21.5 kcal·mol$^{-1}$. By comparing the barriers of R1a and R1b, it
can be concluded that the –OOH insertion reaction is favored over the –CH insertion
reaction. The high reaction barrier of R1b is attributed to the large bond dissociation
energy (BDE) of C-H bond in the Pent1a. To further insight into the reaction mechanism
of R1a, the natural bond orbital (NBO) analysis of the donor-accepter orbitals involved
in the TS1a is performed using the M06-2X wave function. The possible donor-accepter
interactions are estimated by using the second order perturbation theory. As illustrated
in Fig. S4, the strong interactions are identified as the interaction of the lone pair orbital
of $O_6$ atom and the antibonding orbital of $O_4$-$H_4$ bond, and the interaction of the lone
pair orbital of $O_4$ atom and the antibonding orbital of $C_3$-$O_5$ bond.
Similarly, the addition reaction $CH_2OO$ + P1a proceeds through the formation of
the pre-reactive intermediates IM2a and IM2b in the entrance channel, which are
stabilized by a hydrogen bond between the terminal oxygen atom of $CH_2OO$ and the
reacting hydrogen atom of P1a, and a van der Waals (vdW) interaction between the
central carbon atom of $CH_2OO$ and the carbonyl oxygen atom of P1a. The relative
energies of IM2a and IM2b with respect to the separate reactants P1a and $CH_2OO$ are
-1.2 and 3.2 kcal·mol$^{-1}$, respectively, below the energies of the initial reactants $2CH_2OO$
and Pent1a are 41.6 and 37.2 kcal·mol$^{-1}$, respectively. Then they immediately transform
into the respective products P2a and P2b through the –OOH and –CH insertion
transition states TS2a and TS2b with the barriers of 10.1 and 21.6 kcal·mol$^{-1}$. This result
again shows that the –OOH insertion reaction is favored kinetically. It deserves
mentioning that the barrier of –OOH insertion reaction increases as the number of
CH$_2$OO is increased. From the viewpoint of the geometrical parameters of TS2a and
TS2b, the breaking O-H and C-H bonds are elongated by 14.8% and 20.6%,
respectively, with respect to the equilibrium structures of IM2a and IM2b, while the
forming C-O and C-C bond length are 2.013 and 2.264 Å, respectively. The result
reveals that TS2a and TS2b are structurally reactant-like, which are consistent with the
Hammond's hypothesis that the earlier transition states are generally exothermic
(Hammond, 1955).

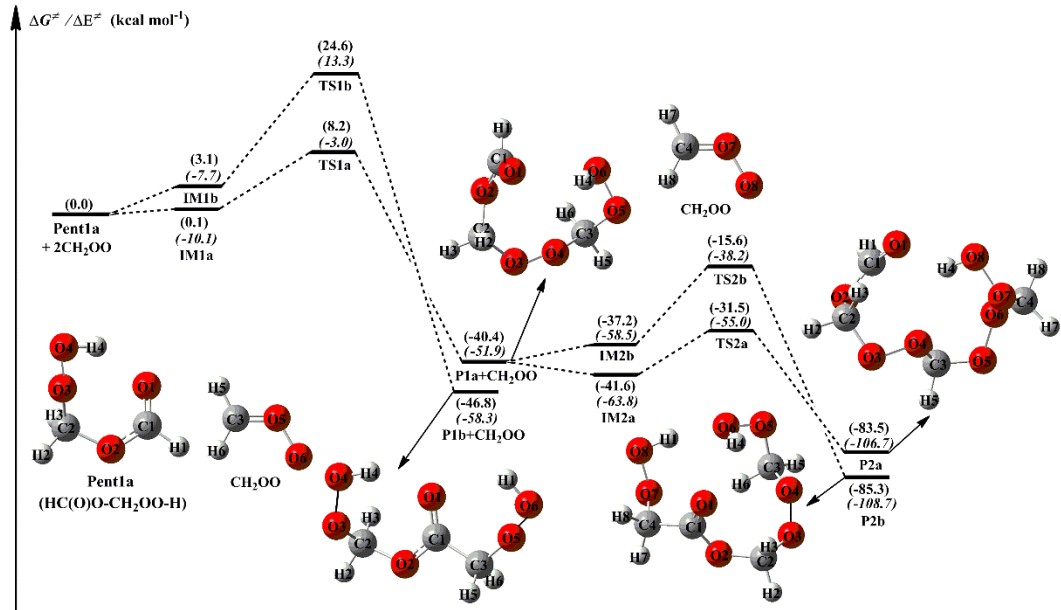

**Figure 2.** PES ($\Delta G$ and $\Delta E$, in italics) for the 2CH$_2$OO + Pent1a reaction at the M06-2X/ma-
TZVP//M06-2X/6-311+G(2df,2p) level of theory
### 3.2.2 The reactions of *anti*-CH$_3$CHOO with Pent1b
The methyl-substituted CH$_2$OO has two conformers, *syn*- and *anti*-CH$_3$CHOO that
distinguish by the orientation of methyl group relative to the terminal oxygen (Taatjes
et al., 2013). *syn*-CH$_3$CHOO is more stable than *anti*-CH$_3$CHOO in energy due to the
existence of intramolecular hydrogen bond (Long et al., 2016). The activation enthalpy
of the interconversion between *syn*-CH$_3$CHOO and *anti*-CH$_3$CHOO is up to 38.5
kcal·mol$^{-1}$, implying that they can treat as independent species in the atmosphere (Long
et al., 2016; Yin and Takahashi, 2017). A schematic PES for the addition reaction 2*anti*-
CH$_3$CHOO + Pent1b is presented in Fig. 3, and the optimized geometries of all
stationary points are shown in Fig. S5. As shown in Fig. 3, the addition reaction 2*anti*-
CH$_3$CHOO + Pent1b proceeds through successive insertion of *anti*-CH$_3$CHOO into
Pent1b leading to the formation of oligomers P4a and P4b that contain *anti*-CH$_3$CHOO
as chain unit. The first *anti*-CH$_3$CHOO addition reaction begins with the formation of
IM3a and IM3b in the entrance channel, which lie -2.2 and 2.4 kcal·mol$^{-1}$ respectively,
with respect to the separate reactants. Then the IM3a and IM3b transform into P3a and
P3b via –OOH and –CH insertion transition states TS3a and TS3b with the barriers of
5.6 and 20.3 kcal·mol$^{-1}$. This result shows that the –OOH insertion reaction is more
favorable than the –CH insertion pathway. Compared with the barriers of R1a and R1b
in the 2CH$_2$OO + Pent1a reaction, the barriers of R3a and R3b decrease by 2.5 and 1.2
kcal·mol$^{-1}$ when a methyl group is introduced at the *anti*-position. The result reveals
that the reactivity of *anti*-CH$_3$CHOO is substantially higher than that of CH$_2$OO. This
conclusion is further supported by the findings of other studies, which have reported
that *anti*-CH$_3$CHOO is more reactive toward H$_2$O, SO$_2$, and H$_2$O$_2$ than CH$_2$OO (Chen
et al., 2017; Taatjes et al., 2013; Huang et al., 2015). Similarly, the secondary *anti*-
CH$_3$CHOO addition reaction starts with the formation of IM4a and IM4b in the entrance
channel with the 0.1 and 3.7 kcal·mol$^{-1}$ stability, followed by conversion to the final
products P4a and P4b through the –OOH and –CH insertion reactions R4a and R4b.
The transition states TS4a and TS4b lie 7.0 and 21.0 kcal·mol$^{-1}$, respectively, above the
energies of the respective intermediates IM4a and IM4b. This result again shows that
the –OOH insertion reaction is the most favorable channel, and the barrier increases as
the number of *anti*-CH$_3$CHOO is increased.

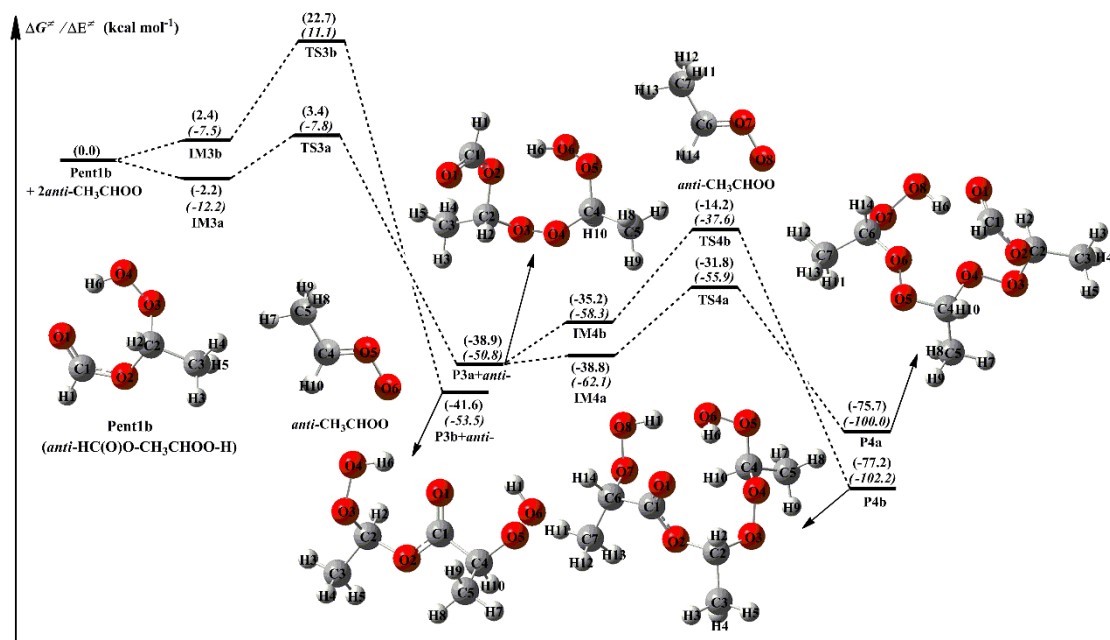

**Figure 3.** PES ($\Delta G$ and $\Delta E$, in italics) for the 2*anti*-CH$_3$CHOO + Pent1b reaction at the M06-2X/ma-TZVP//M06-2X/6-311+G(2df,2p) level of theory

### 3.2.3 The reactions of *syn*-CH$_3$CHOO with Pent1c

Equivalent to the 2*anti*-CH$_3$CHOO + Pent1b reaction, the addition reaction 2*syn*-CH$_3$CHOO + Pent1c has similar transformation pathways, and is thus briefly discussed in the present study. From Fig. 4, it can be seen that the addition reaction 2*syn*-CH$_3$CHOO + Pent1c undergoes via successive insertion of *syn*-CH$_3$CHOO into Pent1c to form P6a and P6b that involve *syn*-CH$_3$CHOO as the repeating unit. The most favorable pathway is that the breakage of O$_4$-H$_6$ bond in the –OOH group of Pent1c occurs simultaneously with the insertion of first *syn*-CH$_3$CHOO into Pent1c to form P5a, followed by the insertion of secondary *syn*-CH$_3$CHOO into P5a to produce P6a. The barriers of these two –OOH insertion reactions R5a and R6a are 13.8 and 11.8 kcal·mol$^{-1}$, respectively, which are higher than those of R3a and R4a in the 2*anti*-CH$_3$CHOO + Pent1b system by 8.2 and 4.8 kcal·mol$^{-1}$, respectively. The result reveals that the reactivity of *syn*-CH$_3$CHOO is substantially lower than that of *anti*-CH$_3$CHOO. Notably, the barrier of the favorable –OOH insertion pathway decreases with increasing the number of *syn*-CH$_3$CHOO in the 2*syn*-CH$_3$CHOO + Pent1c reaction, which is contrary to the case of the 2CH$_2$OO + Pent1a and 2*anti*-CH$_3$CHOO + Pent1b reactions.

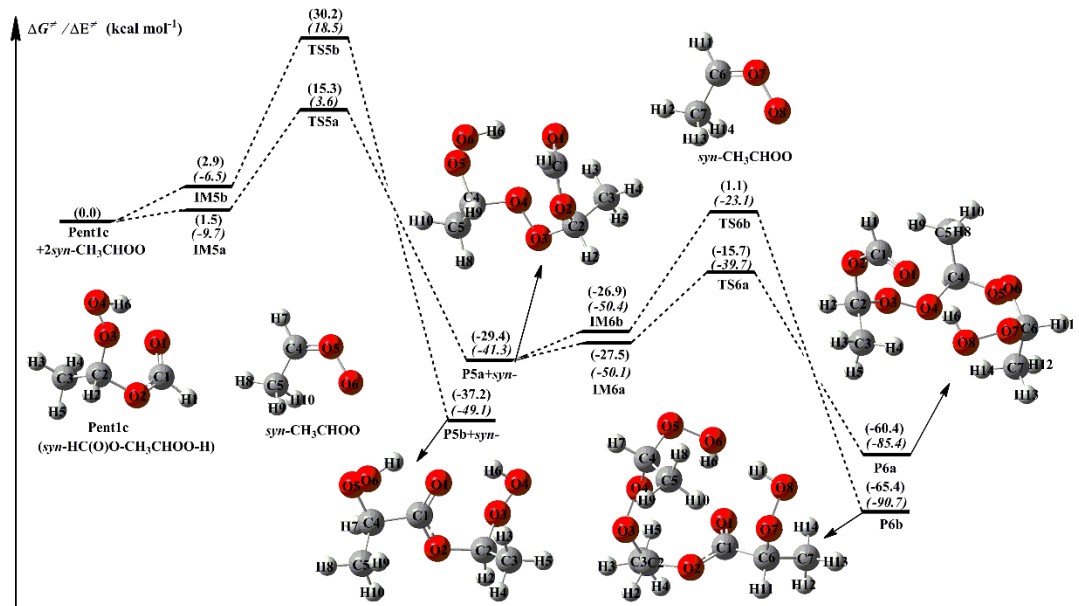

**Figure 4.** PES ($\Delta G$ and $\Delta E$, in italics) for the 2$syn$-CH$_3$CHOO + Pent1c reaction at the M06-2X/ma-TZVP//M06-2X/6-311+G(2df,2p) level of theory

### 3.2.4 The reactions of 2(CH$_3$)$_2$COO with Pent1d

The dimethyl-substituted Criegee intermediate, (CH$_3$)$_2$COO, is generated from the ozonolysis of 2,3-dimethyl-2-butene in the atmosphere (Lester and Klippenstein, 2018; Drozd et al., 2017; Long et al., 2018). The bimolecular reaction of (CH$_3$)$_2$COO with water is not fast enough ($k < 1.5 \times 10^{-16}$ cm$^3$ molecule$^{-1}$ s$^{-1}$), while the reaction of (CH$_3$)$_2$COO with HCOOH has a near gas kinetic limit rate ($k = 5.4 \times 10^{-10}$ cm$^3$ molecule$^{-1}$ s$^{-1}$) (Huang et al., 2015). The result implies that a fraction of (CH$_3$)$_2$COO may survive under high humidity environments and react with HCOOH leading to the formation of hydroperoxide ester Pent1d. A schematic PES for the addition reaction 2(CH$_3$)$_2$COO + Pent1d is plotted in Fig. 5, and the optimized geometries of all stationary points are shown in Fig. S7.

As seen in Fig. 5, the addition reaction 2(CH$_3$)$_2$COO + Pent1d starts with the formation of complexes IM7a and IM7b, which lie 1.9 and 2.4 kcal·mol$^{-1}$, respectively, above the energies of the separate reactants. Then they subsequently transform into products P7a and P7b through the –OOH and –CH insertion transition states TS7a and TS7b with the barriers of 12.2 and 26.4 kcal·mol$^{-1}$. This result again shows that the –OOH insertion reaction is favored over the –CH insertion pathway. A similar conclusion is also obtained from the secondary (CH$_3$)$_2$COO addition reaction that the –OOH

insertion reaction is the dominant pathway. It is of interest to compare the barriers of –OOH insertion reactions in the $(CH_3)_2COO$ + Pent1d system with those of the analogous reactions in other SCIs + Pent1 reactions. It can be found that the barriers decrease in the order of *syn*-$CH_3CHOO$ > $(CH_3)_2COO$ > $CH_2OO$ > *anti*-$CH_3CHOO$ in the first-step SCIs addition reaction, while they become $(CH_3)_2COO$ > *syn*-$CH_3CHOO$ > $CH_2OO$ > *anti*-$CH_3CHOO$ in the second-step SCI addition pathway. The result shows that the reactivity of SCIs is significantly affected by the number and location of methyl substituents. A similar conclusion is also obtained from the thermodynamic parameters that the exothermicity of –OOH insertion reactions significantly decreases with increasing the number of methyl substituents, and the exothermicity of *anti*-methyl substituted carbonyl oxide is obviously higher than that of *syn*-methyl substituted carbonyl oxide.

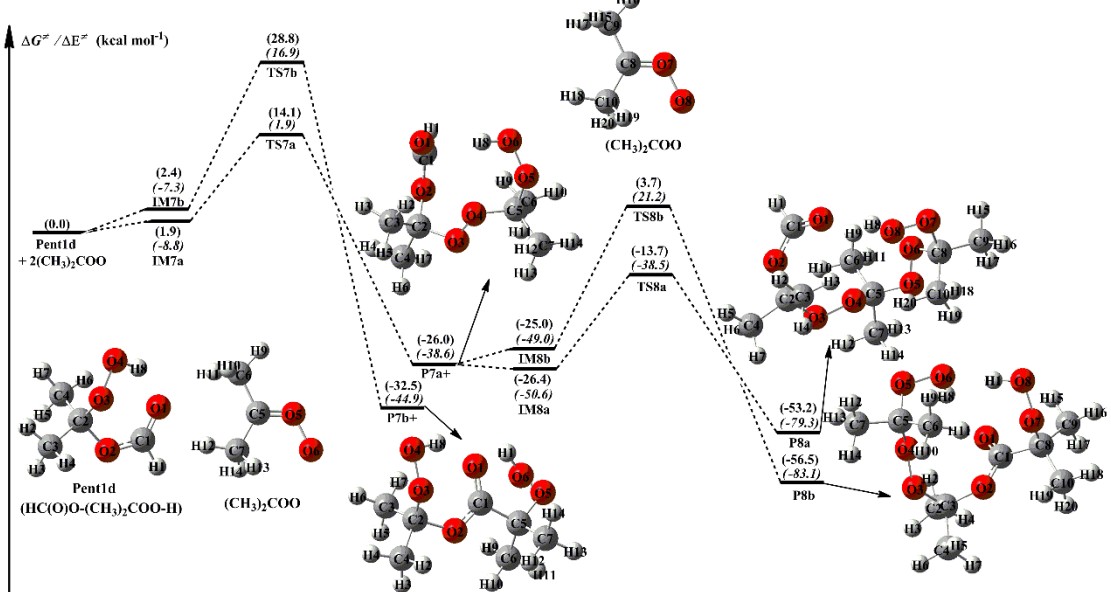

**Figure 5.** PES ($\Delta G$ and $\Delta E$, in italics) for the $2(CH_3)_2COO$ + Pent1d reaction at the M06-2X/ma-TZVP//M06-2X/6-311+G(2df,2p) level of theory.

## 3.3 The reactions of distinct SCIs with Pent1a and implications in atmospheric chemistry

To further elucidate the effect of the number and location of methyl substituents on the reactivity of carbonyl oxides toward hydroperoxide esters, Pent1a (also called as HPMF) is selected as the model compound since it is the simplest hydroperoxide ester

formed from the barrierless reaction of 1,4 O-H insertion of $CH_2OO$ into HCOOH. As mentioned above, –OOH insertion reaction in the oligomerization reactions is the most favorable pathway. Therefore, this type of reaction is merely considered in the reactions of distinct SCIs with Pent1a. The corresponding PES and the optimized geometries of all stationary points are displayed in Figs. 6 and S8, respectively. As seen in Fig. 6, each pathway starts with the formation of a pre-reactive intermediate, and then it overcomes a modest barrier to reaction. The barrier of the reaction of $CH_2OO$ with Pent1a is calculated to be 8.1 kcal·mol$^{-1}$, which is higher than that of the *anti*-$CH_3CHOO$ + Pent1a reaction by 2.5 kcal·mol$^{-1}$. The reason of low barrier can be explained by the NPA atomic charges, as presented in Fig. S9. As seen in Fig. S9, the charges of the central carbon atom $C_1$ and the terminal oxygen atom $O_1$ of $CH_2OO$ are 0.186e and -0.459e, respectively, indicating that $CH_2OO$ is indeed a zwitterion. The $C_1$ atom charge becomes more positive (0.393e), while the $O_1$ atom charge becomes more negative (-0.497e) when a methyl substituent occurs at the *anti*-position. This result suggests that the *anti*-methyl substituent enhances the characteristic of carbonyl oxides zwitterion and reduces the reaction barriers. Compared with the barrier of the $CH_2OO$ + Pent1a reaction, the barriers increase by about 3.0 kcal·mol$^{-1}$ when a methyl group is introduced at the *syn*-position and dimethyl substituent. Although *syn*-methyl and dimethyl substituent promote the raise of carbonyl oxides zwitterion, the steric hindrance effect and intramolecular hydrogen bond are obviously dominant for *syn*-$CH_3CHOO$ and $(CH_3)_2COO$, that are not thus conducive to the nucleophilic attack of hydroperoxide esters. It is worth noting that the exothermicity of distinct SCIs reactions with Pent1a obviously decreases as the number of methyl group is increased, and the exothermicity of *anti*-methyl substituent is higher than that of *syn*-methyl substituent.

The rate coefficients of distinct SCIs reactions with Pent1a are calculated in the temperature range of 273-400 K as summarized in Table S9. This table shows that the rate coefficients $k_{R1a}$ of the $CH_2OO$ + Pent1a reaction (R1a) decrease in the range of $5.0 \times 10^{-11}$ (273 K) to $5.0 \times 10^{-12}$ cm$^3$ molecule$^{-1}$ s$^{-1}$ (400 K) with increasing temperature. A similar phenomenon is also observed from the rate coefficients of Pent1a reactions

with *anti*-CH$_3$CHOO (R9), *syn*-CH$_3$CHOO (R10), and (CH$_3$)$_2$COO (R11) that they
exhibit a slightly negative temperature dependence. $k_{R9}$ is several orders of magnitude
greater than $k_{R1a}$, $k_{R10}$ and $k_{R11}$ in the whole temperature range, suggesting that the
bimolecular reaction *anti*-CH$_3$CHOO + Pent1a (R9) is favored kinetically. Compared
with the rate coefficients of R1a, the rate coefficients increase by about one order of
magnitude when a methyl substituent occurs at the *anti*-position, whereas the rate
coefficients decrease by 1 to 2 orders of magnitude when a methyl group is introduced
at the *syn*-position. It should be noted that although the barrier of R10 is nearly identical
to that of R11, $k_{R10}$ is 1 to 2 orders of magnitude lower than $k_{R11}$ in the entire temperature
range. This is probably because the rate coefficients are mediated by pre-reactive
intermediates that IM11 is more stable than IM10 in energy.

It is of interest to assess whether the reactions of distinct SCIs with HPMF can

compete well with the losses to reactions with trace species (e.g., H$_2$O, HCOOH and
SO$_2$), because it is well known that the reactions with trace species are expected to be
the dominant chemical sinks for SCIs in the atmosphere (Taatjes et al., 2013; Long et
al., 2016). The reported concentrations of coreactant, the rate coefficients $k$, and the
effective pseudo-first-order rate constants ($k_{eff} = k$[coreactant]) for distinct SCI
reactions with H$_2$O, HCOOH, SO$_2$, and HPMF are summarized in Table 2. As seen in
Table 2, the rate coefficient of a particular SCI reaction with trace species is strongly
dependent on its structure. The methyl group substitution may alter the rate coefficient
by several to tens of times. The atmospheric concentrations of H$_2$O, HCOOH and SO$_2$
in the tropical forest environments are measured to be 3.9-6.1 $\times$ 10$^{17}$, 5.0-10 $\times$ 10$^{10}$, and
1.7-9.0 $\times$ 10$^{10}$ molecules cm$^{-3}$, respectively (Vereecken, 2012). For the reactions of
CH$_2$OO with H$_2$O, HCOOH, and SO$_2$, the experimental rate coefficients are determined
to be $< 1.5 \times 10^{-15}$, $[1.1 \pm 0.1] \times 10^{-10}$, and $[3.9 \pm 0.7] \times 10^{-11}$ cm$^3$ molecule$^{-1}$ s$^{-1}$,
respectively (Welz et al., 2012 and 2014; Chao et al., 2015), which translate into
$k_{eff(CH2OO+H2O)}$, $k_{eff(CH2OO+HCOOH)}$ and $k_{eff(CH2OO+SO2)}$ of 5.9-9.2 $\times$ 10$^2$, 5.5-11, and 0.7-3.5
s$^{-1}$, respectively. The result reveals that the reaction of CH$_2$OO with H$_2$O is the most
important bimolecular reaction. $k_{eff(CH2OO+HCOOH)}$ is greater by a factor of 3-8 than
$k_{eff(CH2OO+SO2)}$, indicating that the reaction of $CH_2OO$ with HCOOH is favored over
reaction with $SO_2$. Similar conclusion is also obtained from the results of $k_{eff}$ for the
reactions of *anti*-$CH_3CHOO$, *syn*-$CH_3CHOO$ and $(CH_3)_2COO$ with $H_2O$, HCOOH and
$SO_2$ that SCIs reactions with $H_2O$ are faster than with HCOOH, which, in turn, are
faster than with $SO_2$.

According to the results shown in the Table 2, the room temperature rate

coefficient for the reaction of $CH_2OO$ with HPMF is calculated to be $2.7 \times 10^{-11}$ $cm^3$
molecule$^{-1}$ s$^{-1}$. However, to the best of our knowledge, the atmospheric concentration
of HPMF has not been reported up to now. If we assume that the concentration of HPMF
is the same as that of HCOOH, $k_{eff(CH2OO+HPMF)}$ is estimated to be 1.4-2.7 s$^{-1}$, which is
significantly lower than $k_{eff(CH2OO+H2O)}$ and $k_{eff(CH2OO+HCOOH)}$. $k_{eff(CH2OO+HPMF)}$ is nearly
identical to $k_{eff(CH2OO+SO2)}$, indicating that the $CH_2OO$ + HPMF reaction is competitive
with the $CH_2OO$ + $SO_2$ system. Previous model-measurement studies have estimated
the surface-level SCIs concentrations in the range of $1.0 \times 10^4$ to $1.0 \times 10^5$ molecules
cm$^{-3}$ (Khan et al., 2018; Novelli et al., 2017). If we assume that the concentration of
HPMF is equal to that of SCIs, $k_{eff(CH2OO+HPMF)}$ is calculated to be $2.7\text{-}27 \times 10^{-7}$ s$^{-1}$,
which is several orders of magnitude lower than $k_{eff(CH2OO+H2O)}$, $k_{eff(CH2OO+HCOOH)}$ and
$k_{eff(CH2OO+SO2)}$. This result indicates that the reaction of $CH_2OO$ with HPMF is of less
importance. Similar conclusion is also obtained from the reactions of *anti*-$CH_3CHOO$,
*syn*-$CH_3CHOO$ and $(CH_3)_2COO$ with HPMF.

To further evaluate the relative importance of the complex SCIs reactions with

coreactant, the bimolecular reactions of methyl vinyl ketone oxide (MVK-OO) with
$H_2O$, HCOOH, $SO_2$, and HPMF are considered. MVK-OO, formed with 21 to 23%
yield from the ozonolysis of isoprene, is a four carbon, asymmetric, resonance-
stabilized Criegee intermediate (Barber et al., 2018). MVK-OO has four conformers,
*syn-trans-*, *syn-cis-*, *anti-trans-*, and *anti-cis-* as shown in Fig. S10. Herein, *syn* and *anti*
refer to the orientation of the –$CH_3$ group relative to the terminal oxygen of MVK-OO,
whereas *cis* and *trans* refer to the orientation of the $C_8=C_9$ bond relative to the $C_1=O_2$
bond. According to the results shown in the Fig. S10, the lowest-energy conformer is

*syn-trans*-MVK-OO, which is lower than *syn-cis*-, *anti-trans*-, and *anti-cis*-MVK-OO by 1.42, 2.43 and 2.69 kcal·mol$^{-1}$, respectively. Therefore, the lowest-energy conformer *syn-trans*-MVK-OO is selected as the model compound to study its bimolecular reactions. As shown in Table 2, the rate coefficient of $H_2O$ reaction with *syn-trans*-MVK-OO is lower than with other SCIs by 2 to 3 orders of magnitude. The reason is likely to be that the existence of methyl and vinyl groups hinders the occurrence of bimolecular reaction with water vapour. Consequently, a fraction of *syn-trans*-MVK-OO may survive in the presence of water vapour and react with other species. $k_{eff(MVK-OO+H2O)}$ is nearly identical to $k_{eff(MVK-OO+HCOOH)}$, which is greater than $k_{eff(MVK-OO+SO2)}$ and $k_{eff(MVK-OO+HPMF)}$ when the concentration of HPMF is the same as that of HCOOH. $k_{eff(MVK-OO+H2O)}$ and $k_{eff(MVK-OO+HCOOH)}$ are greater than $k_{eff(MVK-OO+SO2)}$, which, in turn, are greater than $k_{eff(MVK-OO+HPMF)}$ when the concentration of HPMF is equal to that of SCIs. Based on the above discussions, it can be concluded that the relative importance of carbonyl oxides reactions with hydroperoxide esters is significantly dependent on the concentrations of hydroperoxide esters. These reactions may play a certain role in the formation of organic new particle in some regions where low concentration of water vapour and high concentration of hydroperoxide esters occur.

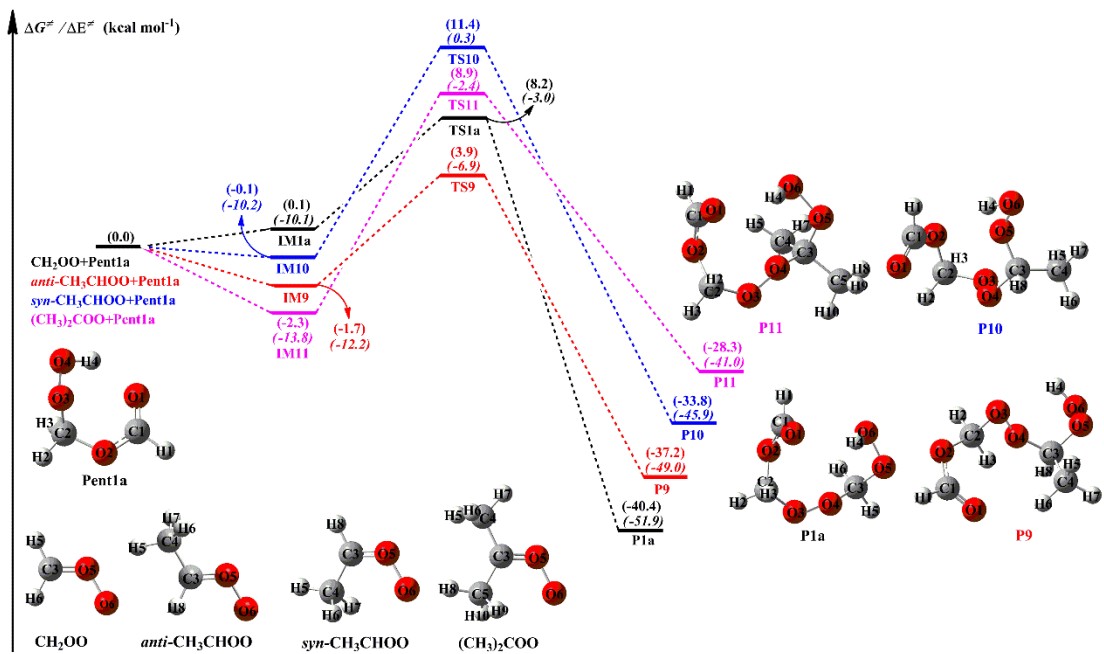

**Figure 6.** PES ($\Delta G$ and $\Delta E$, in italics) for the distinct SCIs + Pent1a reactions at the M06-2X/ma-TZVP//M06-2X/6-311+G(2df,2p) level of theory

**Table 2** The reported concentrations of coreactant, the rate coefficients $k$, and the effective pseudo-
first-order rate constants ($k_{eff} = k$[coreactant]) for distinct SCI reactions with HPMF, $H_2O$, HCOOH
and $SO_2$ at the tropical forest environments

| SCIs | Coreactant | [Coreactant] (molecules $cm^{-3}$) | $k$ ($cm^3$ molecule$^{-1}$ s$^{-1}$) | $k_{eff}$ (s$^{-1}$) | Reference |
|---|---|---|---|---|---|
| CH$_2$OO | $H_2O$ | $3.9\text{-}6.1 \times 10^{17}$ | $< 1.5 \times 10^{-15}$ | $5.9\text{-}9.2 \times 10^2$ | Chao et al., (2015) |
| | HCOOH | $5.0\text{-}10.0 \times 10^{10}$ | $[1.1 \pm 0.1] \times 10^{-10}$ | 5.5-11 | Welz et al., (2014)] |
| | $SO_2$ | $1.7\text{-}9.0 \times 10^{10}$ | $[3.9 \pm 0.7] \times 10^{-11}$ | 0.7-3.5 | Welz et al., (2012) |
| | HPMF | - | $2.7 \times 10^{-11}$ | - | This work |
| anti-CH$_3$CHOO | $H_2O$ | $3.9\text{-}6.1 \times 10^{17}$ | $[1.0 \pm 0.4] \times 10^{-14}$ | $3.9\text{-}6.1 \times 10^3$ | Taatjes et al., (2013) |
| | HCOOH | $5.0\text{-}10.0 \times 10^{10}$ | $[5 \pm 3] \times 10^{-10}$ | 25.0-50.0 | Welz et al., (2014) |
| | $SO_2$ | $1.7\text{-}9.0 \times 10^{10}$ | $[6.7 \pm 1.0] \times 10^{-11}$ | 1.1-6.0 | Taatjes et al., (2013) |
| | HPMF | - | $3.3 \times 10^{-10}$ | - | This work |
| syn-CH$_3$CHOO | $H_2O$ | $3.9\text{-}6.1 \times 10^{17}$ | $< 4.0 \times 10^{-15}$ | $1.6\text{-}2.4 \times 10^3$ | Taatjes et al., (2013) |
| | HCOOH | $5.0\text{-}10.0 \times 10^{10}$ | $[2.5 \pm 0.3] \times 10^{-10}$ | 12.5-25.0 | Welz et al., (2014) |
| | $SO_2$ | $1.7\text{-}9.0 \times 10^{10}$ | $[2.4 \pm 0.3] \times 10^{-11}$ | 0.4-2.2 | Taatjes et al., (2013) |
| | HPMF | - | $1.7 \times 10^{-13}$ | - | This work |
| (CH$_3$)$_2$COO | $H_2O$ | $3.9\text{-}6.1 \times 10^{17}$ | $< 1.5 \times 10^{-16}$ | 58.5-91.5 | Huang et al., (2015) |
| | HCOOH | $5.0\text{-}10.0 \times 10^{10}$ | $4.5 \times 10^{-10}$ | 22.5-45.0 | Sipilä et al., (2014) |
| | $SO_2$ | $1.7\text{-}9.0 \times 10^{10}$ | $1.3 \times 10^{-10}$ | 2.2-11.7 | Huang et al., (2015) |
| | HPMF | - | $2.2 \times 10^{-11}$ | - | This work |
| syn-trans-MVK-OO | $H_2O$ | $3.9\text{-}6.1 \times 10^{17}$ | $< 4.0 \times 10^{-17}$ | 15.6-24.4 | Caravan et al., (2020) |
| | HCOOH | $5.0\text{-}10.0 \times 10^{10}$ | $[3.0 \pm 0.1] \times 10^{-10}$ | 15.0-30.0 | Caravan et al., (2020) |
| | $SO_2$ | $1.7\text{-}9.0 \times 10^{10}$ | $[4.2 \pm 0.6] \times 10^{-11}$ | 0.7-3.8 | Caravan et al., (2020) |

| | | | | |
|---|---|---|---|---|
| HPMF | - | $3.0 \times 10^{-11}$ | - | This work |

### 3.4 Vapour pressure and volatility of adduct products

The assessment of Barley and McFiggans (2010) and O'Meara et al. (2014) found that the combination of boiling point estimation from Nannoolal et al. (2004) and vapour pressure estimation from Nannoolal et al. (2008) gives the lowest mean bias error of vapour pressure for atmospherically relevant compounds. Therefore, the saturated vapour pressure ($P^0$) of adduct products at room temperature is estimated by using the Nannoolal-Nannoolal method, and the results are listed in Table S10. From Table S10, it can be seen that the $P^0$ of adduct products involved in the successive reactions of $CH_2OO$ with HCOOH increases first and then decreases with increasing the number of $CH_2OO$. The $P^0$ of the adduct product $HC(O)O(CH_2OO)_3H$ is maximum when the number of $CH_2OO$ is equal to three. The $P^0$ of adduct products included in the successive reactions of anti-$CH_3CHOO$ with HCOOH decreases significantly as the number of anti-$CH_3CHOO$ is increased. Similar phenomenon is also observed from the successive reactions of syn-$CH_3CHOO$ and $(CH_3)_2COO$ with HCOOH. Notably, the $P^0$ of adduct products decreases obviously when the size of SCIs increases. For example, the $P^0$ of the adduct product $HC(O)O(CH_2OO)_3H$ in the $nCH_2OO$ + HCOOH reaction is estimated to be $4.43 \times 10^{-3}$ atm, which is greater than those of the corresponding adduct products in the nanti-$CH_3CHOO$ + HCOOH ($7.12 \times 10^{-4}$), nsyn-$CH_3CHOO$ + HCOOH ($7.12 \times 10^{-4}$), and $n(CH_3)_2COO$ + HCOOH ($1.27 \times 10^{-4}$) reactions by 6.22, 6.22 and 34.88 times, respectively.

A classify scheme of various organic compounds is based on their volatility, as presented by Donahue et al. (2012) The volatility of organic compounds is described by their effective saturation concentration. The saturated concentrations ($c^0$) of adduct products formed from the successive reactions of SCIs with HCOOH are predicted by using the SIMPOL.1 method proposed by Pankow and Asher (2008), and the results are listed in Table S10. As shown in Table S10, the $c^0$ of adduct products involved in the $nCH_2OO$ + HCOOH reaction decreases with increasing the number of $CH_2OO$. According to the Volatility Basis Set (VBS) of organic compounds (Donahue et al.,

2012), these adduct products belong to VOC ($c^0 > 3 \times 10^6$ ug/m$^3$). Similarly, the $c^0$ of
adduct products included in the n*anti*-CH$_3$CHOO + HCOOH, n*syn*-CH$_3$CHOO +
HCOOH, and n(CH$_3$)$_2$COO + HCOOH reactions decreases when the number of SCIs
increases. It deserves mentioning that the adduct products in the n*anti*-CH$_3$CHOO +
HCOOH and n*syn*-CH$_3$CHOO + HCOOH reactions belong to intermediate volatility
organic compounds (IVOC, $300 < c^0 < 3 \times 10^6$ ug/m$^3$) when the number of SCIs is equal
to five. However, the adduct products in the n(CH$_3$)$_2$COO + HCOOH reaction become
IVOC when the number of (CH$_3$)$_2$COO is greater than or equal to two. Based on the
above discussions, it can be concluded that the volatility of adduct products is
significantly affected by the number and size of SCIs in the successive reaction of SCIs
with HCOOH.

## 4. Conclusions

The oligomerization reaction mechanism and kinetics of Criegee intermediates
reactions with their respective hydroperoxide esters as well as HPMF are investigated
using quantum chemical calculations and kinetics modeling methods. The main
conclusion is summarized as follows.
(a) For the initiation reactions of distinct SCIs with HCOOH, the barrierless 1,4
O-H insertion reaction leading to the formation of hydroperoxide esters is the most
favorable pathway. The exothermicity of distinct SCIs reactions with HCOOH
decreases when the number of methyl groups increases, and the exothermicity of the
*anti*-CH$_3$CHOO + HCOOH reaction is higher than that of the *syn*-CH$_3$CHOO +
HCOOH system.
(b) The addition reactions of SCIs with hydroperoxide esters proceed through
successive insertion of SCIs into hydroperoxide ester to form oligomers that involve
SCIs as the repeating unit. These oligomerization reactions are strongly exthermoic and
spontaneous. The exothermicity of oligomerization reactions significantly decreases
when the number of methyl substituents increases, and the exothermicity of *anti*-methyl
substituted carbonyl oxides is obviously higher than that of *syn*-methyl substituted
carbonyl oxides.
(c) The –OOH insertion reaction is favored over the –CH insertion pathway in the
SCIs oligomerization reactions, and the barrier heights increase with increasing the
number of SCIs added to the oligomer except *syn*-$CH_3CHOO$. The barrier of –OOH
insertion pathway shows a dramatic decrease when a methyl substituent occurs at the
*anti*-position, while it reveals a significant increase when a methyl group is introduced
at the *syn*-position and dimethyl substituent.
(d) Compared with the barrier of $CH_2OO$ reaction with HPMF (8.1 kcal·mol$^{-1}$),
the barrier decreases by 2.5 kcal·mol$^{-1}$ when a methyl substituent occurs at the *anti*-
position, while the barrier increases by about 3.0 kcal·mol$^{-1}$ when a methyl group is
introduced at the *syn*-position and dimethyl substituent. The rate coefficients increase
by about one order of magnitude when a methyl substituent occurs at the *anti*-position,
whereas the rate coefficients decrease by 1 to 2 orders of magnitude when a methyl
group is introduced at the *syn*-position compared to the rate coefficients of the $CH_2OO$
+ HPMF reaction.

## Data availability

The data are accessible by contacting the corresponding author
(huangyu@ieecas.cn).

## Supplement

The following information is provided in the Supplement: The electronic energy
($\Delta E^{\neq}$) and Gibbs free energy ($\Delta G^{\neq}$) barriers for the initiation reactions of distinct SCIs
with HCOOH predicted at different levels; Enthalpies of formation for the various
carbonyl oxides and hydroperoxide esters; Rate coefficients of initiation reactions of
distinct SCIs with HCOOH; Rate coefficients of distinct SCIs reactions with HPMF;
Predicted saturated vapour pressure ($P^0$) and saturated concentrations ($c^0$) for the adduct
products; Relaxed scan energy profiles for varying the C-O and O-H bonds in the 1,4-
insertion reactions of distinct SCIs with HCOOH; Natural bond orbital (NBO) analysis
of the donor-acceptor orbitals; The NPA charges of different atoms in the distinct SCIs;
Optimized geometries of all the stationary points;

## Author contribution

LC designed the study. LC and YH wrote the paper. LC performed theoretical
calculation. YX, ZJ, and WW analyzed the data. All authors reviewed and commented
on the paper.

## Competing interests

The authors declare that they have no conflict of interest.

## Acknowledgments

This work was supported by the National Natural Science Foundation of China
(grant Nos. 42175134, 41805107, and 22002080). It was also partially supported as
Strategic Priority Research Program of the Chinese Academy of Sciences, China (grant
Nos. XDA23010300 and XDA23010000), and CAS "Light of West China" Program
(XAB2019B01).

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
