# Peer review of "Oligomer formation from the gas-phase reactions of Criegee intermediates with hydroperoxide esters: mechanism and kinetics"

_Atmospheric Chemistry and Physics, 2022_

## Author Comment (AC1)

Prof. Yu Huang
State Key Lab of Loess and Quaternary Geology
Institute of Earth Environment, Chinese Academy
of Sciences, Xi'an, 710061, China
Tel./Fax: (86) 29-62336261
E-mail: huangyu@ieecas.cn

Aug. 25, 2022

Dear Prof. Kourtchev,

**Revision for Manuscript ACP-2022-376**

We thank you very much for giving us the opportunity to revise our manuscript. We highly appreciate the reviewers for their comments and suggestions on the manuscript entitled "**Oligomer formation from the gas-phase reactions of Criegee intermediates with hydroperoxide esters: mechanism and kinetics**". We have made revisions of our manuscript carefully according to the comments and suggestions of reviewers. The revised contents are marked in blue color. The response letter to reviewers is attached at the end of this cover letter.

We hope that the revised manuscript can meet the requirement of Atmospheric Chemistry & Physics. Any further modifications or revisions, please do not hesitate to contact us.

Look forward to hearing from you as soon as possible.

Best regards,

Yu Huang

**Comments of reviewer #1**

1. The authors should explain their variational TST calculations for barrierless reactions (p.7) in more detail, particularly since they consistently predict higher CI + HCOOH rate constants than experiment (p.10-11).

   **Response:** In the original manuscript, the rate coefficients for the barrierless reactions are calculated by employing the variational transition state theory (VTST), and the rate coefficients for the bimolecular reactions with the tight transition states are computed by using the canonical transition state theory (CTST) along with one-dimensional asymmetric Eckart tunneling correction. For the initiation reactions of distinct stabilized Criegee intermediates (SCIs) with HCOOH, there are four possible pathways, namely (1) 1,4 O-H insertion (Entry 1), (2) 1,2 O-H insertion (Entry 2), (3) C-H insertion (Entry 3), and (4) C=O cycloaddition (Entry 4), in which Entry 1 is barrierless and Entry 2-4 have the tight transition states. The total rate coefficient for the reaction of SCIs with HCOOH is equal to the sum of the rate coefficient of each pathway. For the barrierless 1,4 O-H insertion reaction, the VTST is approximated with a Morse potential function, $V(R) = D_e\{1\text{-exp}[-\beta(R\text{-}R_e)]\}^2$, along with an anisotropy potential function to stand for the minimum energy path, which is used to calculate the rate coefficients (Raghunath et al., 2017). Here, $D_e$ is the bond energy excluding the zero-point energy, $R$ is the reaction coordinate, and $R_e$ is the equilibrium value of $R$. It is assumed that the stretching potential in an anisotropy potential is used in conjunction with a potential form of $V_{\text{anisotropy}} = V_0[1\text{-cos}^2(\theta_1–\theta_{1e}) \times \cos^2(\theta_2–\theta_{2e})]$ (Raghunath et al., 2017). Here, $V_0$ is the stretching potential, which stands for by a Morse potential, $\theta_1$ and $\theta_{1e}$ represent the rotational angle between fragment 1 and the reference axis and the equilibrium bond angle of fragment 1, $\theta_2$ and $\theta_{2e}$ stand for the rotational angle between fragment 2 and the reference axis and the equilibrium bond angle of fragment 2. The association curve for the reaction of 1,4 O-H insertion of SCIs into HCOOH is computed at the M06-2X/6-311+G(2df,2p) level of theory to cover a range from 0.97 to 1.97 Å at step size 0.1 Å for O-H bond and from 1.44 to 2.44 Å at step size 0.1 Å for C-O bond, while other structural parameters are fully optimized. The computed potential energies are fitted to the Morse potential function. However, the calculated rate coefficients for the reactions of SCIs with HCOOH are higher than the prior experimental measurements. The reason is ascribed to the fact that the approximation of VTST using a Morse potential function in conjunction with an anisotropy potential function is unsuitable to predict the rate coefficients for the barrierless 1,4 O-H insertion

reaction.

In the revised manuscript, the rate coefficients for the barrierless reactions are computed by employing the inverse Laplace transformation (ILT) method, and the rate coefficients for the bimolecular reactions with the tight transition states are calculated by utilizing CTST in conjunction with Eckart tunneling correction. The ILT and CTST/Eckart calculations are performed by using the MESMER 6.0 and KiSThelP 2019 programs, respectively (Glowacki et al., 2012; Canneaux et al., 2013). In the ILT treatment, the rotational constants, vibrational frequencies, molecular weights, energies and other input parameters are obtained from the M06-2X/6-311+G(2df,2p) or M06-2X/ma-TZVP methods. For the barrierless reaction of 1,4 O-H insertion of SCIs into HCOOH, SCIs and HCOOH are assigned as the deficient and excess reactants, respectively. The concentration of HCOOH is given a value of $5.0 \times 10^{10}$ molecules $cm^{-3}$ in the simulation, which is taken from the typical concentration of HCOOH in the tropical forest environments (Vereecken, 2012). $N_2$ is applied as the buffer gas. A single exponential down model is employed to simulate the collision transfer ($<\Delta E>_{down}$ = 200 $cm^{-1}$). The collisional Lennard-Jones parameters are estimated with the empirical formula described by Gilbert and Smith (1990).

The rate coefficients of each elementary pathway included in the initiation reactions of distinct SCIs with HCOOH are calculated in the temperature range of 273-400 K, as listed in Table S3-S6. As shown in Table S3, the total rate coefficients $k_{tot-CH2OO}$ of $CH_2OO$ reaction with HCOOH are in excess of $1.0 \times 10^{-10}$ $cm^3$ molecule$^{-1}$ s$^{-1}$, and they exhibit a slightly negative temperature dependence in the temperature range studied. $k_{tot-CH2OO}$ is estimated to be $1.4 \times 10^{-10}$ $cm^3$ molecule$^{-1}$ s$^{-1}$ at 298 K, which is in good agreement with the experimental values reported by Welz et al. (2014) ([$1.1 \pm 0.1$] $\times 10^{-10}$), Chung et al. (2019) ([$1.4 \pm 0.3$] $\times 10^{-10}$), and Peltola et al. (2020) ([$1.0 \pm 0.03$] $\times 10^{-10}$). $k(TS_{ent1})$ is approximately equal to $k_{tot-CH2OO}$ in the whole temperature range, and it decreases in the range of $1.7 \times 10^{-10}$ (273 K) to $1.2 \times 10^{-10}$ (400 K) $cm^3$ molecule$^{-1}$ s$^{-1}$ with increasing temperature. $k(TS_{ent1})$ is several orders of magnitude greater than $k(TS_{ent2})$, $k(TS_{ent3})$ and $k(TS_{ent4})$ over the temperature range from 273 to 400 K. The result again shows that the barrierless 1,4 O-H insertion reaction is predominant. Similar conclusion is also obtained from the results of the rate coefficients for the reactions of HCOOH with *anti*-$CH_3CHOO$, *syn*-$CH_3CHOO$ and $(CH_3)_2COO$ (Table S4-S6). At ambient temperature, the total rate coefficients of HCOOH reactions with *anti*-$CH_3CHOO$, *syn*-$CH_3CHOO$ and $(CH_3)_2COO$ are estimated to be 5.9, 2.7 and 4.8 $\times$ $10^{-10}$ $cm^3$ molecule$^{-1}$ s$^{-1}$,

respectively, which are consistent with the prior experimental measurements of $5 \pm 3$, $2.5 \pm 0.3$ and $4.5 \times 10^{-10}$ $cm^3$ $molecule^{-1}$ $s^{-1}$ (Welz et al., 2014; Chung et al., 2019; Sipilä et al., 2014).

**Table S3** Rate coefficients ($cm^3$ $molecule^{-1}$ $s^{-1}$) of each elementary pathway involved in the initiation reaction of $CH_2OO$ with HCOOH computed at different temperatures

| T/K | $k$ ($TS_{ent1}$) | $k$ ($TS_{ent2}$) | $k$ ($TS_{ent3}$) | $k$ ($TS_{ent4}$) | $k_{tot\text{-}CH2OO}$ |
|---|---|---|---|---|---|
| 273 | $1.7 \times 10^{-10}$ | $3.6 \times 10^{-12}$ | $1.0 \times 10^{-22}$ | $3.6 \times 10^{-12}$ | $1.8 \times 10^{-10}$ |
| 280 | $1.6 \times 10^{-10}$ | $2.9 \times 10^{-12}$ | $1.2 \times 10^{-22}$ | $3.1 \times 10^{-12}$ | $1.7 \times 10^{-10}$ |
| 298 | $1.4 \times 10^{-10}$ | $1.9 \times 10^{-12}$ | $2.2 \times 10^{-22}$ | $2.3 \times 10^{-12}$ | $1.4 \times 10^{-10}$ |
| 300 | $1.4 \times 10^{-10}$ | $1.8 \times 10^{-12}$ | $2.4 \times 10^{-22}$ | $2.2 \times 10^{-12}$ | $1.4 \times 10^{-10}$ |
| 320 | $1.3 \times 10^{-10}$ | $1.2 \times 10^{-12}$ | $4.9 \times 10^{-22}$ | $1.6 \times 10^{-12}$ | $1.3 \times 10^{-10}$ |
| 340 | $1.3 \times 10^{-10}$ | $8.2 \times 10^{-13}$ | $1.0 \times 10^{-21}$ | $1.3 \times 10^{-12}$ | $1.3 \times 10^{-10}$ |
| 360 | $1.2 \times 10^{-10}$ | $5.9 \times 10^{-13}$ | $2.2 \times 10^{-21}$ | $1.0 \times 10^{-12}$ | $1.2 \times 10^{-10}$ |
| 380 | $1.2 \times 10^{-10}$ | $4.5 \times 10^{-13}$ | $4.5 \times 10^{-21}$ | $8.2 \times 10^{-13}$ | $1.2 \times 10^{-10}$ |
| 400 | $1.2 \times 10^{-10}$ | $3.5 \times 10^{-13}$ | $9.0 \times 10^{-21}$ | $6.9 \times 10^{-13}$ | $1.2 \times 10^{-10}$ |

**Table S4** Rate coefficients ($cm^3$ $molecule^{-1}$ $s^{-1}$) of each elementary pathway involved in the initiation reaction of *anti*-$CH_3CHOO$ with HCOOH computed at different temperatures

| T/K | $k$ ($TS_{ent1}$-*anti*) | $k$ ($TS_{ent2}$-*anti*) | $k$ ($TS_{ent3}$-*anti*) | $k$ ($TS_{ent4}$-*anti*) | $k_{tot\text{-}anti}$ |
|---|---|---|---|---|---|
| 273 | $5.9 \times 10^{-10}$ | $4.2 \times 10^{-11}$ | $5.5 \times 10^{-22}$ | $6.1 \times 10^{-11}$ | $6.9 \times 10^{-10}$ |
| 280 | $5.7 \times 10^{-10}$ | $3.8 \times 10^{-11}$ | $6.7 \times 10^{-22}$ | $4.9 \times 10^{-11}$ | $6.6 \times 10^{-10}$ |
| 298 | $5.4 \times 10^{-10}$ | $2.3 \times 10^{-11}$ | $1.2 \times 10^{-21}$ | $3.0 \times 10^{-11}$ | $5.9 \times 10^{-10}$ |
| 300 | $5.3 \times 10^{-10}$ | $2.0 \times 10^{-11}$ | $1.3 \times 10^{-21}$ | $2.8 \times 10^{-11}$ | $5.8 \times 10^{-10}$ |
| 320 | $5.0 \times 10^{-10}$ | $1.5 \times 10^{-11}$ | $2.6 \times 10^{-21}$ | $1.7 \times 10^{-11}$ | $5.3 \times 10^{-10}$ |
| 340 | $4.7 \times 10^{-10}$ | $9.4 \times 10^{-12}$ | $5.4 \times 10^{-21}$ | $1.1 \times 10^{-11}$ | $4.9 \times 10^{-10}$ |
| 360 | $4.5 \times 10^{-10}$ | $7.0 \times 10^{-12}$ | $1.1 \times 10^{-20}$ | $7.8 \times 10^{-12}$ | $4.7 \times 10^{-10}$ |
| 380 | $4.4 \times 10^{-10}$ | $3.6 \times 10^{-12}$ | $2.1 \times 10^{-20}$ | $5.6 \times 10^{-12}$ | $4.5 \times 10^{-10}$ |
| 400 | $4.3 \times 10^{-10}$ | $2.0 \times 10^{-12}$ | $4.0 \times 10^{-20}$ | $4.2 \times 10^{-12}$ | $4.4 \times 10^{-10}$ |

**Table S5** Rate coefficients ($cm^3$ $molecule^{-1}$ $s^{-1}$) of each elementary pathway involved in the initiation reaction of *syn*-$CH_3CHOO$ with HCOOH computed at different temperatures

| T/K | $k$ ($TS_{ent1}$-*syn*) | $k$ ($TS_{ent2}$-*syn*) | $k$ ($TS_{ent3}$-*syn*) | $k$ ($TS_{ent4}$-*syn*) | $k_{tot\text{-}syn}$ |
|---|---|---|---|---|---|

| | | | | | |
|---|---|---|---|---|---|
| 273 | $3.1 \times 10^{-10}$ | $9.5 \times 10^{-13}$ | $4.6 \times 10^{-27}$ | $7.5 \times 10^{-16}$ | $3.1 \times 10^{-10}$ |
| 280 | $2.8 \times 10^{-10}$ | $8.0 \times 10^{-13}$ | $7.1 \times 10^{-27}$ | $6.4 \times 10^{-16}$ | $2.8 \times 10^{-10}$ |
| 298 | $2.7 \times 10^{-10}$ | $5.4 \times 10^{-13}$ | $8.9 \times 10^{-26}$ | $5.5 \times 10^{-16}$ | $2.7 \times 10^{-10}$ |
| 300 | $2.7 \times 10^{-10}$ | $5.2 \times 10^{-13}$ | $9.9 \times 10^{-26}$ | $4.6 \times 10^{-16}$ | $2.7 \times 10^{-10}$ |
| 320 | $2.5 \times 10^{-10}$ | $3.6 \times 10^{-13}$ | $3.0 \times 10^{-25}$ | $3.8 \times 10^{-16}$ | $2.5 \times 10^{-10}$ |
| 340 | $2.5 \times 10^{-10}$ | $2.6 \times 10^{-13}$ | $9.1 \times 10^{-25}$ | $3.1 \times 10^{-16}$ | $2.5 \times 10^{-10}$ |
| 360 | $2.3 \times 10^{-10}$ | $2.0 \times 10^{-13}$ | $2.6 \times 10^{-24}$ | $3.0 \times 10^{-16}$ | $2.3 \times 10^{-10}$ |
| 380 | $2.2 \times 10^{-10}$ | $1.5 \times 10^{-13}$ | $7.2 \times 10^{-24}$ | $2.4 \times 10^{-16}$ | $2.2 \times 10^{-10}$ |
| 400 | $2.2 \times 10^{-10}$ | $1.2 \times 10^{-13}$ | $1.8 \times 10^{-23}$ | $2.2 \times 10^{-16}$ | $2.2 \times 10^{-10}$ |

**Table S6** Rate coefficients ($cm^3$ molecule$^{-1}$ s$^{-1}$) of each elementary pathway involved in the initiation reaction of $(CH_3)_2OO$ with HCOOH computed at different temperatures

| T/K | $k$ (TS$_{ent1}$-*dim*) | $k$ (TS$_{ent2}$-*dim*) | $k$ (TS$_{ent3}$-*dim*) | $k$ (TS$_{ent4}$-*dim*) | $k_{tot\text{-}dim}$ |
|---|---|---|---|---|---|
| 273 | $5.3 \times 10^{-10}$ | $6.8 \times 10^{-12}$ | $1.4 \times 10^{-26}$ | $4.4 \times 10^{-15}$ | $5.4 \times 10^{-10}$ |
| 280 | $5.1 \times 10^{-10}$ | $5.2 \times 10^{-12}$ | $2.2 \times 10^{-26}$ | $4.2 \times 10^{-15}$ | $5.2 \times 10^{-10}$ |
| 298 | $4.8 \times 10^{-10}$ | $2.8 \times 10^{-12}$ | $8.0 \times 10^{-26}$ | $4.0 \times 10^{-15}$ | $4.8 \times 10^{-10}$ |
| 300 | $4.7 \times 10^{-10}$ | $2.6 \times 10^{-12}$ | $9.2 \times 10^{-26}$ | $3.9 \times 10^{-15}$ | $4.7 \times 10^{-10}$ |
| 320 | $4.5 \times 10^{-10}$ | $1.4 \times 10^{-12}$ | $3.6 \times 10^{-25}$ | $3.7 \times 10^{-15}$ | $4.5 \times 10^{-10}$ |
| 340 | $4.2 \times 10^{-10}$ | $8.6 \times 10^{-13}$ | $1.3 \times 10^{-24}$ | $3.6 \times 10^{-15}$ | $4.2 \times 10^{-10}$ |
| 360 | $3.9 \times 10^{-10}$ | $5.5 \times 10^{-13}$ | $4.5 \times 10^{-24}$ | $3.5 \times 10^{-15}$ | $3.9 \times 10^{-10}$ |
| 380 | $3.7 \times 10^{-10}$ | $3.7 \times 10^{-13}$ | $1.4 \times 10^{-23}$ | $3.4 \times 10^{-15}$ | $3.7 \times 10^{-10}$ |
| 400 | $3.7 \times 10^{-10}$ | $2.6 \times 10^{-13}$ | $3.9 \times 10^{-23}$ | $3.4 \times 10^{-15}$ | $3.7 \times 10^{-10}$ |

Corresponding descriptions have been added in the page 7 line 173-190, page 11 line 303-315, page 12 line 330-338 and page 13 line 346-351 of the revised manuscript:

*The rate coefficients for the barrierless reactions are determined by employing the inverse Laplace transformation (ILT) method. The ILT calculations are performed with the MESMER 6.0 program (Glowacki et al., 2012). In the ILT treatment, the rotational constants, vibrational frequencies, molecular weights, energies and other input parameters are obtained from the M06-2X/6-311+G(2df,2p) or M06-2X/ma-TZVP methods. For the barrierless reaction of 1,4 O-H insertion of SCIs into HCOOH, SCIs and HCOOH are assigned as the deficient and excess reactants,*

respectively. The concentration of HCOOH is given a value of $5.0 \times 10^{10}$ molecules $cm^{-3}$ in the simulation, which is taken from the typical concentration of HCOOH in the tropical forest environments (Vereecken et al., 2012). $N_2$ is applied as the buffer gas. A single exponential down model is employed to simulate the collision transfer ($<\Delta E>_{down}$ = 200 $cm^{-1}$). The collisional Lennard-Jones parameters are estimated with the empirical formula described by Gilbert and Smith (1990).

The rate coefficients for the bimolecular reactions with the tight transition states are calculated by using the canonical transition state theory (CTST) along with one-dimensional asymmetric Eckart tunneling correction (Truhlar et al., 1996; Eckart, 1930). The CTST/Eckart calculations are performed with the KiSThelP 2019 program (Canneaux et al., 2013).

The rate coefficients of each elementary pathway included in the initiation reactions of distinct SCIs with HCOOH are calculated in the temperature range of 273-400 K, as listed in Table S3-S6. As shown in Table S3, the total rate coefficients $k_{tot\text{-}CH2OO}$ of $CH_2OO$ reaction with HCOOH are in excess of $1.0 \times 10^{-10}$ $cm^3$ $molecule^{-1}$ $s^{-1}$, and they exhibit a slightly negative temperature dependence in the temperature range studied. $k_{tot\text{-}CH2OO}$ is estimated to be $1.4 \times 10^{-10}$ $cm^3$ $molecule^{-1}$ $s^{-1}$ at 298 K, which is in good agreement with the experimental values reported by Welz et al. (2014) ([1.1 ± 0.1] $\times 10^{-10}$), Chung et al. (2019) ([1.4 ± 0.3] $\times 10^{-10}$), and Peltola et al. (2020) ([1.0 ± 0.03] $\times 10^{-10}$). $k(TS_{ent1})$ is approximately equal to $k_{tot\text{-}CH2OO}$ in the whole temperature range, and it decreases in the range of $1.7 \times 10^{-10}$ (273 K) to $1.2 \times 10^{-10}$ (400 K) $cm^3$ $molecule^{-1}$ $s^{-1}$ with increasing temperature. $k(TS_{ent1})$ is several orders of magnitude greater than $k(TS_{ent2})$, $k(TS_{ent3})$ and $k(TS_{ent4})$ over the temperature range from 273 to 400 K. The result again shows that the barrierless 1,4 O-H insertion reaction is predominant.

Equivalent to the case of $CH_2OO$ reaction with HCOOH, the rate coefficient of each elementary pathway involved in the anti-$CH_3CHOO$ + HCOOH reaction also decreases with the temperature increasing (Table S4). This table shows that Entry 1 is kinetically favored over Entry 2, 3 and 4, and Entry 2 is competitive with Entry 4 in the range 273-400 K. Similar conclusion is also obtained from the results of the rate coefficients for the reactions of syn-$CH_3CHOO$ and $(CH_3)_2COO$ with HCOOH that Entry 1 is the dominant pathway (Table S5-S6). It deserves mentioning that the competition of Entry 2 is significantly greater than that of Entry 4 in the syn-$CH_3CHOO$ + HCOOH and $(CH_3)_2COO$ + HCOOH systems. At ambient temperature, the total rate

*coefficients of HCOOH reactions with anti-CH₃CHOO, syn-CH₃CHOO and (CH₃)₂COO are estimated to be 5.9, 2.7 and 4.8 × 10⁻¹⁰ cm³ molecule⁻¹ s⁻¹, respectively, which are consistent with the prior experimental measurements of 5 ± 3, 2.5 ± 0.3 and 4.5 × 10⁻¹⁰ cm³ molecule⁻¹ s⁻¹ (Welz et al., 2014; Chung et al., 2019; Sipilä et al., 2014).*

2. The trend in exothermicity with substitution pattern (p.8-9) should be explained.

**Response:** Based on the Reviewer's suggestion, the relevance explanations on the trend in exothermicity have been added in the revised manuscript. The exothermicity of 1,4 O-H insertion reactions of distinct SCIs with HCOOH is assessed by the reaction enthalpy ($\Delta_r H^{\circ}_{298}$), which is defined as the difference between the enthalpies of formation ($\Delta_f H^{\circ}_{298}$) of the products and reactants ($\Delta_r H^{\circ}_{298} = \sum_{products} \Delta_f H^{\circ}_{298} - \sum_{reactants} \Delta_f H^{\circ}_{298}$). To the best of our knowledge, there are no literature values available on the enthalpies of formation of carbonyl oxides and hydroperoxide esters except the simplest carbonyl oxide $CH_2OO$. Therefore, the isodesmic reaction method is adopted to obtain the enthalpies of formation, and the results are listed in Table S2. An isodesmic reaction is a hypothetical reaction, in which the type of chemical bonds in the reactants is the similar as that of chemical bonds in the products. The following isodesmic reaction is constructed because the experimental values of $H_2$, $CH_4$ and $H_2O$ are available ($\Delta_f H^{\circ}_{298}$ ($H_2$) = 0.00 kcal·mol⁻¹; $\Delta_f H^{\circ}_{298}$ ($CH_4$) = -17.82 kcal·mol⁻¹; $\Delta_f H^{\circ}_{298}$ ($H_2O$) = -57.79 kcal·mol⁻¹).

$$SCIs + nH_2 \rightarrow CH_4 + mH_2O \qquad (4)$$

As seen in Table S2, the enthalpy of formation of $CH_2OO$ is calculated to be 23.23 kcal·mol⁻¹, which is in good agreement with the available literature values (Karton et al., 2013; Chen et al., 2016). This result implies that the theoretical method employed herein is reasonable to predict the thermochemical parameters. The enthalpies of formation of carbonyl oxides and hydroperoxide esters significantly decrease with increasing the number of methyl groups. Notably, the decreased values in the enthalpies of formation of carbonyl oxides are greater than those of hydroperoxide esters under the condition of the same number of methyl groups. For example, the enthalpy of formation of *anti*-CH₃CHOO decreases by 12.95 kcal·mol⁻¹ compared to the enthalpy of formation of $CH_2OO$, and the enthalpy of formation of Pent1b decreases by 12.12 kcal·mol⁻¹ compared to the

enthalpy of formation of Pent1a. The reaction enthalpies decrease in the order of -44.69 ($CH_2OO$ + HCOOH → Pent1a) < -43.86 (*anti*-$CH_3CHOO$ + HCOOH → Pent1b) < -38.13 (*syn*-$CH_3CHOO$ + HCOOH → Pent1c) < -37.12 kcal·mol$^{-1}$ (($CH_3)_2COO$ + HCOOH → Pent1d), indicating that the reaction enthalpies are highly dependent on the number and location of methyl groups. The trend in reaction enthalpies is consistent with the trend in the enthalpies of formation of carbonyl oxides. The reason might be attributed to the decreased values in the enthalpies of formation of carbonyl oxides greater than those of hydroperoxide esters under the condition of the same number of methyl groups.

**Table S2** Enthalpies of formation ($\Delta_f H^o_{298}$) for the various carbonyl oxides and hydroperoxide esters computed at the CCSD(T)//M06-2X/6-311+G(2df,2p) level of theory

| Species | Cal (kcal·mol$^{-1}$) | Refs. (kcal·mol$^{-1}$) |
|---|---|---|
| $CH_2OO$ | 23.23 | 22.92[a]
 24.59[b] |
| *anti*-$CH_3OO$ | 10.28 | |
| *syn*-$CH_3CHOO$ | 6.73 | |
| $(CH_3)_2COO$ | -6.77 | |
| HCOOH | | -90.62 (exp) |
| HC(O)OCH$_2$OOH (Pent1a) | -112.08 | |
| HC(O)OCH(CH$_3$)OOH (Pent1b) | -124.20 | |
| HC(O)OCH(CH$_3$)OOH (Pent1c) | -122.02 | |
| HC(O)OC(CH$_3$)$_2$OOH (Pent1d) | -134.51 | |

Exp is taken from NIST Chemistry Webbook
[a] the value is obtained at the G4 level of theory (Chen et al., 2016)
[b] the value is obtained at the W3-F12 level of theory (Karton et al., 2013)

Corresponding descriptions have been added in the page 9 line 240-247 and page 10 line 248-271 of the revised manuscript:

*The exothermicity of 1,4 O-H insertion reactions of distinct SCIs with HCOOH is assessed by the reaction enthalpy ($\Delta_r H^o_{298}$), which is defined as the difference between the enthalpies of formation ($\Delta_f H^o_{298}$) of the products and reactants ($\Delta_r H^o_{298} = \sum_{products} \Delta_f H^o_{298} - \sum_{reactants} \Delta_f H^o_{298}$). To the best of our knowledge, there are no literature values available on the enthalpies of formation of carbonyl oxides and hydroperoxide esters except the simplest carbonyl oxide $CH_2OO$. Therefore, the isodesmic reaction method is adopted to obtain the enthalpies of formation, and the results are listed*

*in Table S2. An isodesmic reaction is a hypothetical reaction, in which the type of chemical bonds in the reactants is the similar as that of chemical bonds in the products. The following isodesmic reaction is constructed because the experimental values of $H_2$, $CH_4$ and $H_2O$ are available ($\Delta_f H^o_{298}$ ($H_2$) = 0.00 kcal·mol$^{-1}$;  $\Delta_f H^o_{298}$ ($CH_4$) = -17.82 kcal·mol$^{-1}$;  $\Delta_f H^o_{298}$ ($H_2O$) = -57.79 kcal·mol$^{-1}$).*

$$SCIs + nH_2 \;\rightarrow\; CH_4 + mH_2O \tag{4}$$

*As seen in Table S2, the enthalpy of formation of $CH_2OO$ is calculated to be 23.23 kcal·mol$^{-1}$, which is in good agreement with the available literature values (Chen et al., 2016; Karton et al., 2013). This result implies that the theoretical method employed herein is reasonable to predict the thermochemical parameters. The enthalpies of formation of carbonyl oxides and hydroperoxide esters significantly decrease with increasing the number of methyl groups. Notably, the decreased values in the enthalpies of formation of carbonyl oxides are greater than those of hydroperoxide esters under the condition of the same number of methyl groups. For example, the enthalpy of formation of anti-$CH_3CHOO$ decreases by 12.95 kcal·mol$^{-1}$ compared to the enthalpy of formation of $CH_2OO$, and the enthalpy of formation of Pent1b decreases by 12.12 kcal·mol$^{-1}$ compared to the enthalpy of formation of Pent1a. The reaction enthalpies decrease in the order of -44.69 ($CH_2OO$ + HCOOH $\rightarrow$ Pent1a) < -43.86 (anti-$CH_3CHOO$ + HCOOH $\rightarrow$ Pent1b) < -38.13 (syn-$CH_3CHOO$ + HCOOH $\rightarrow$ Pent1c) < -37.12 kcal·mol$^{-1}$ (($CH_3$)$_2$COO + HCOOH $\rightarrow$ Pent1d), indicating that the reaction enthalpies are highly dependent on the number and location of methyl groups. The trend in reaction enthalpies is consistent with the trend in the enthalpies of formation of carbonyl oxides. The reason might be attributed to the decreased values in the enthalpies of formation of carbonyl oxides greater than those of hydroperoxide esters under the condition of the same number of methyl groups.*

3. The analysis of possible bimolecular CI reactions (p.21) should be extended to the three substituted CIs.

**Response:** Kalinowski et al. has comfirmed that the central CO bond of carbonyl oxides is a double bond, while the terminal OO bond is a single bond (Kalinowski et al., 2014). It is therefore that the maximum degree of substitution of carbonyl oxides is two. To further evaluate the relative importance of the complex SCIs reactions with coreactant, the bimolecular reactions of methyl vinyl ketone

oxide (MVK-OO) with $H_2O$, HCOOH, $SO_2$ and HPMF have been considered in the revised manuscript. MVK-OO, formed with 21 to 23% yield from the ozonolysis of isoprene, is a four carbon, asymmetric, resonance-stabilized Criegee intermediate (Barber et al., 2018). MVK-OO has four conformers, *syn-trans-*, *syn-cis-*, *anti-trans-*, and *anti-cis-* as shown in Fig. S10. Herein, *syn* and *anti* refer to the orientation of the -$CH_3$ group relative to the terminal oxygen of MVK-OO, whereas *cis* and *trans* refer to the orientation of the $C_8=C_9$ bond relative to the $C_1=O_2$ bond. According to the results shown in the Fig. S10, the lowest-energy conformer is *syn-trans*-MVK-OO, which is lower than *syn-cis-*, *anti-trans-*, and *anti-cis*-MVK-OO by 1.42, 2.43 and 2.69 kcal·mol$^{-1}$, respectively. Therefore, the lowest-energy conformer *syn-trans*-MVK-OO is selected as the model compound to study its bimolecular reactions. As shown in Table 2, the rate coefficient of $H_2O$ reaction with *syn-trans*-MVK-OO is lower than with other SCIs by 2 to 3 orders of magnitude. The reason is likely to be that the existence of methyl and vinyl groups hinders the occurrence of bimolecular reaction with water vapour. Consequently, a fraction of *syn-trans*-MVK-OO may survive in the presence of water vapour and react with other species. $k_{eff(MVK-OO+H2O)}$ is nearly identical to $k_{eff(MVK-OO+HCOOH)}$, which is greater than $k_{eff(MVK-OO+SO2)}$ and $k_{eff(MVK-OO+HPMF)}$ when the concentration of HPMF is the same as that of HCOOH. $k_{eff(MVK-OO+H2O)}$ and $k_{eff(MVK-OO+HCOOH)}$ are greater than $k_{eff(MVK-OO+SO2)}$, which, in turn, are greater than $k_{eff(MVK-OO+HPMF)}$ when the concentration of HPMF is equal to that of SCIs. Based on the above discussions, it can be concluded that the relative importance of carbonyl oxides reactions with hydroperoxide esters is significantly dependent on the concentrations of hydroperoxide esters. These reactions may play a certain role in the formation of organic new particle in some regions where low concentration of water vapour and high concentration of hydroperoxide esters occur.

| | trans | cis |
|---|---|---|
| syn |  |  |
| | 0.00 | 1.42 |

[Figure]

| anti | 2.43 | 2.69 |

**Figure S10.** The optimized geometries and relative energies (kcal·mol$^{-1}$) computed for the four conformers of MVK-oxide. Geometries are optimized at the M06-2X/6-311+g(2df,2p) level of theory. Single point energies are calculated at the CCSD(T)/6-311+g(2df,2p) level of theory.

Corresponding descriptions have been added in the page 24 line 611-619 and page 25 line 620-636 of the revised manuscript:

*To further evaluate the relative importance of the complex SCIs reactions with coreactant, the bimolecular reactions of methyl vinyl ketone oxide (MVK-OO) with $H_2O$, HCOOH, $SO_2$, and HPMF are considered. MVK-OO, formed with 21 to 23% yield from the ozonolysis of isoprene, is a four carbon, asymmetric, resonance-stabilized Criegee intermediate (Barber et al., 2018). MVK-OO has four conformers, syn-trans-, syn-cis-, anti-trans-, and anti-cis- as shown in Fig. S10. Herein, syn and anti refer to the orientation of the $–CH_3$ group relative to the terminal oxygen of MVK-OO, whereas cis and trans refer to the orientation of the $C_8=C_9$ bond relative to the $C_1=O_2$ bond. According to the results shown in the Fig. S10, the lowest-energy conformer is syn-trans-MVK-OO, which is lower than syn-cis-, anti-trans-, and anti-cis-MVK-OO by 1.42, 2.43 and 2.69 kcal·mol$^{-1}$, respectively. Therefore, the lowest-energy conformer syn-trans-MVK-OO is selected as the model compound to study its bimolecular reactions. As shown in Table 2, the rate coefficient of $H_2O$ reaction with syn-trans-MVK-OO is lower than with other SCIs by 2 to 3 orders of magnitude. The reason is likely to be that the existence of methyl and vinyl groups hinders the occurrence of bimolecular reaction with water vapour. Consequently, a fraction of syn-trans-MVK-OO may survive in the presence of water vapour and react with other species. $k_{eff(MVK-OO+H2O)}$ is nearly identical to $k_{eff(MVK-OO+HCOOH)}$, which is greater than $k_{eff(MVK-OO+SO2)}$ and $k_{eff(MVK-OO+HPMF)}$ when the concentration of HPMF is the same as that of HCOOH. $k_{eff(MVK-OO+H2O)}$ and $k_{eff(MVK-OO+HCOOH)}$ are greater than $k_{eff(MVK-OO+SO2)}$, which, in turn, are greater than $k_{eff(MVK-OO+HPMF)}$ when the concentration of HPMF is equal to that of SCIs. Based on the above discussions, it can be concluded*

*that the relative importance of carbonyl oxides reactions with hydroperoxide esters is significantly dependent on the concentrations of hydroperoxide esters. These reactions may play a certain role in the formation of organic new particle in some regions where low concentration of water vapour and high concentration of hydroperoxide esters occur.*

4. Since the CI is clearly the limiting reactant in the CI + HCOOH reaction, the atmospheric concentration of HPMF (and the other hydroperoxy esters) is much better estimated to be the CI concentration. (This, of course, will greatly lower the predicted pseudo-first-order rate constants for the CI + HPMF reaction.)

**Response:** Based on the Reviewer's suggestion, the relevance explanations on the predicted pseudo-first-order rate constants have been added in the revised manuscript. It is of interest to assess whether the reactions of distinct SCIs with HPMF can compete well with the losses to reactions with trace species (e.g., $H_2O$, HCOOH and $SO_2$), because it is well known that the reactions with trace species are expected to be the dominant chemical sinks for SCIs in the atmosphere (Taatjes et al., 2013; Long et al., 2016). The reported concentrations of coreactant, the rate coefficients $k$, and the effective pseudo-first-order rate constants ($k_{eff} = k$[coreactant]) for distinct SCI reactions with $H_2O$, HCOOH, $SO_2$ and HPMF are summarized in Table 2. As seen in Table 2, the rate coefficient of a particular SCI reaction with trace species is strongly dependent on its structure. The methyl group substitution may alter the rate coefficient by several to tens of times. The atmospheric concentrations of $H_2O$, HCOOH and $SO_2$ in the tropical forest environments are measured to be 3.9-6.1 $\times$ $10^{17}$, 5.0-10 $\times$ $10^{10}$, and 1.7-9.0 $\times$ $10^{10}$ molecules $cm^{-3}$, respectively (Vereecken, 2012). For the reactions of $CH_2OO$ with $H_2O$, HCOOH, and $SO_2$, the experimental rate coefficients are determined to be < 1.5 $\times$ $10^{-15}$, [1.1 $\pm$ 0.1] $\times$ $10^{-10}$, and [3.9 $\pm$ 0.7] $\times$ $10^{-11}$ $cm^3$ molecule$^{-1}$ s$^{-1}$, respectively (Welz et al., 2012 and 2014; Chao et al., 2015), which translate into $k_{eff(CH2OO+H2O)}$, $k_{eff(CH2OO+HCOOH)}$ and $k_{eff(CH2OO+SO2)}$ of 5.9-9.2 $\times$ $10^2$, 5.5-11, and 0.7-3.5 s$^{-1}$, respectively. The result reveals that the reaction of $CH_2OO$ with $H_2O$ is the most important bimolecular reaction. $k_{eff(CH2OO+HCOOH)}$ is greater by a factor of 3-8 than $k_{eff(CH2OO+SO2)}$, indicating that the reaction of $CH_2OO$ with HCOOH is favored over reaction with $SO_2$. Similar conclusion is also obtained from the results of $k_{eff}$ for the reactions of *anti*-$CH_3CHOO$, *syn*-$CH_3CHOO$ and $(CH_3)_2COO$ with $H_2O$, HCOOH and $SO_2$ that SCIs reactions with $H_2O$ are faster than with HCOOH, which, in turn, are

faster than with $SO_2$.

[revised manuscript text omitted]

5. Since a big motivation for the computations is the potential for CI + hydroperoxy ester reactions to lead to SOA, there should be some specific discussion, perhaps buttressed by rough calculations, of how many cycles of CI addition are required before a given adduct is expected to have low

volatility. The approach of Chhantyal-Pun et al. (ACS Earth Space Chem. 2018, 2, 8, 833-842) is an example of the approach the authors should take.

**Response:** Based on the Reviewer's suggestion, the vapour pressure and volatility of adduct products formed from the successive reactions of SCIs with hydroperoxide esters have been added in the revised manuscript. The assessment of Barley and McFiggans (2010) and O'Meara et al. (2014) found that the combination of boiling point estimation from Nannoolal et al. (2004) and vapour pressure estimation from Nannoolal et al. (2008) gives the lowest mean bias error of vapour pressure for atmospherically relevant compounds. Therefore, the saturated vapour pressure ($P^0$) of adduct products at room temperature is estimated by using the Nannoolal-Nannoolal method, and the results are listed in Table S10.

From Table S10, it can be seen that the $P^0$ of adduct products involved in the successive reactions of $CH_2OO$ with HCOOH increases first and then decreases with increasing the number of $CH_2OO$. The $P^0$ of the adduct product $HC(O)O(CH_2OO)_3H$ is maximum when the number of $CH_2OO$ is equal to three. The $P^0$ of adduct products included in the successive reactions of *anti*-$CH_3CHOO$ with HCOOH decreases significantly as the number of *anti*-$CH_3CHOO$ is increased. Similar phenomenon is also observed from the successive reactions of *syn*-$CH_3CHOO$ and $(CH_3)_2COO$ with HCOOH. Notably, the $P^0$ of adduct products decreases obviously when the size of SCIs increases. For example, the $P^0$ of the adduct product $HC(O)O(CH_2OO)_3H$ in the $nCH_2OO$ + HCOOH reaction is estimated to be $4.43 \times 10^{-3}$ atm, which is greater than those of the corresponding adduct products in the n*anti*-$CH_3CHOO$ + HCOOH ($7.12 \times 10^{-4}$), n*syn*-$CH_3CHOO$ + HCOOH ($7.12 \times 10^{-4}$), and $n(CH_3)_2COO$ + HCOOH ($1.27 \times 10^{-4}$) reactions by 6.22, 6.22 and 34.88 times, respectively.

A classify scheme of various organic compounds is based on their volatility, as presented by Donahue et al. (2012) The volatility of organic compounds is described by their effective saturation concentration. The saturated concentrations ($c^0$) of adduct products formed from the successive reactions of SCIs with HCOOH are predicted by using the SIMPOL.1 method proposed by Pankow and Asher (2008), and the results are listed in Table S10. As shown in Table S10, the $c^0$ of adduct products involved in the $nCH_2OO$ + HCOOH reaction decreases with increasing the number of $CH_2OO$. According to the Volatility Basis Set (VBS) of organic compounds (Donahue et al., 2012), these adduct products belong to volatile organic compounds (VOC, $c^0 > 3 \times 10^6$ ug/m$^3$). Similarly,

the $c^0$ of adduct products included in the n$anti$-CH$_3$CHOO + HCOOH, n$syn$-CH$_3$CHOO + HCOOH, and n(CH$_3$)$_2$COO + HCOOH reactions decreases when the number of SCIs increases. It deserves mentioning that the adduct products in the n$anti$-CH$_3$CHOO + HCOOH and n$syn$-CH$_3$CHOO + HCOOH reactions belong to intermediate volatility organic compounds (IVOC, $300 < c^0 < 3 \times 10^6$ ug/m$^3$) when the number of SCIs is equal to five. However, the adduct products in the n(CH$_3$)$_2$COO + HCOOH reaction become IVOC when the number of (CH$_3$)$_2$COO is greater than or equal to two. Based on the above discussions, it can be concluded that the volatility of adduct products is significantly affected by the number and size of SCIs in the successive reaction of SCIs with HCOOH.

**Table S10** Predicted saturated vapour pressure ($P^0$) and saturated concentrations ($c^0$) for the adduct products of the successive reactions of SCIs with HCOOH

| | formula | $P^0$ (atm) | $c^0$ (ug/m$^3$) |
|---|---|---|---|
| n CH$_2$OO + HCOOH | | | |
| n = 1 | HC(O)OCH$_2$OOH | $2.12 \times 10^{-3}$ | $7.86 \times 10^7$ |
| n = 2 | HC(O)O(CH$_2$OO)$_2$H | $3.80 \times 10^{-3}$ | $3.99 \times 10^7$ |
| n = 3 | HC(O)O(CH$_2$OO)$_3$H | $4.43 \times 10^{-3}$ | $3.91 \times 10^7$ |
| n = 4 | HC(O)O(CH$_2$OO)$_4$H | $4.21 \times 10^{-3}$ | $3.29 \times 10^7$ |
| n = 5 | HC(O)O(CH$_2$OO)$_5$H | $3.59 \times 10^{-3}$ | $2.12 \times 10^7$ |
| n    $anti$-CH$_3$CHOO    + HCOOH | | | |
| n = 1 | HC(O)OCH(CH$_3$)OOH | $1.25 \times 10^{-3}$ | $8.32 \times 10^6$ |
| n = 2 | HC(O)O(CH(CH$_3$)OO)$_2$H | $1.13 \times 10^{-3}$ | $7.57 \times 10^6$ |
| n = 3 | HC(O)O(CH(CH$_3$)OO)$_3$H | $7.12 \times 10^{-4}$ | $6.49 \times 10^6$ |
| n = 4 | HC(O)O(CH(CH$_3$)OO)$_4$H | $3.90 \times 10^{-4}$ | $4.50 \times 10^6$ |
| n = 5 | HC(O)O(CH(CH$_3$)OO)$_5$H | $2.01 \times 10^{-4}$ | $2.81 \times 10^6$ |
| n    $syn$-CH$_3$CHOO    + HCOOH | | | |
| n = 1 | HC(O)OCH(CH$_3$)OOH | $1.25 \times 10^{-3}$ | $8.32 \times 10^6$ |
| n = 2 | HC(O)O(CH(CH$_3$)OO)$_2$H | $1.13 \times 10^{-3}$ | $7.57 \times 10^6$ |
| n = 3 | HC(O)O(CH(CH$_3$)OO)$_3$H | $7.12 \times 10^{-4}$ | $6.49 \times 10^6$ |
| n = 4 | HC(O)O(CH(CH$_3$)OO)$_4$H | $3.90 \times 10^{-4}$ | $4.50 \times 10^6$ |
| n = 5 | HC(O)O(CH(CH$_3$)OO)$_5$H | $2.01 \times 10^{-4}$ | $2.81 \times 10^6$ |
| n (CH$_3$)$_2$COO + HCOOH | | | |
| n = 1 | HC(O)OC(CH$_3$)$_2$OOH | $7.23 \times 10^{-4}$ | $3.50 \times 10^6$ |
| n = 2 | HC(O)O(C(CH$_3$)$_2$OO)$_2$H | $3.50 \times 10^{-4}$ | $2.74 \times 10^6$ |
| n = 3 | HC(O)O(C(CH$_3$)$_2$OO)$_3$H | $1.27 \times 10^{-4}$ | $1.38 \times 10^6$ |
| n = 4 | HC(O)O(C(CH$_3$)$_2$OO)$_4$H | $4.27 \times 10^{-5}$ | $5.90 \times 10^5$ |
| n = 5 | HC(O)O(C(CH$_3$)$_2$OO)$_5$H | $1.40 \times 10^{-5}$ | $2.36 \times 10^5$ |

Corresponding descriptions have been added in the page 27 line 644-671 and page 28 line 672-682 of the revised manuscript:

*The assessment of Barley and McFiggans (2010) and O'Meara et al. (2014) found that the combination of boiling point estimation from Nannoolal et al. (2004) and vapour pressure estimation from Nannoolal et al. (2008) gives the lowest mean bias error of vapour pressure for atmospherically relevant compounds. Therefore, the saturated vapour pressure ($P^0$) of adduct products at room temperature is estimated by using the Nannoolal-Nannoolal method, and the results are listed in Table S10. From Table S10, it can be seen that the $P^0$ of adduct products involved in the successive reactions of $CH_2OO$ with HCOOH increases first and then decreases with increasing the number of $CH_2OO$. The $P^0$ of the adduct product $HC(O)O(CH_2OO)_3H$ is maximum when the number of $CH_2OO$ is equal to three. The $P^0$ of adduct products included in the successive reactions of anti-$CH_3CHOO$ with HCOOH decreases significantly as the number of anti-$CH_3CHOO$ is increased. Similar phenomenon is also observed from the successive reactions of syn-$CH_3CHOO$ and $(CH_3)_2COO$ with HCOOH. Notably, the $P^0$ of adduct products decreases obviously when the size of SCIs increases. For example, the $P^0$ of the adduct product $HC(O)O(CH_2OO)_3H$ in the $nCH_2OO$ + HCOOH reaction is estimated to be $4.43 \times 10^{-3}$ atm, which is greater than those of the corresponding adduct products in the nanti-$CH_3CHOO$ + HCOOH ($7.12 \times 10^{-4}$), nsyn-$CH_3CHOO$ + HCOOH ($7.12 \times 10^{-4}$), and $n(CH_3)_2COO$ + HCOOH ($1.27 \times 10^{-4}$) reactions by 6.22, 6.22 and 34.88 times, respectively.*

*A classify scheme of various organic compounds is based on their volatility, as presented by Donahue et al. (2012) The volatility of organic compounds is described by their effective saturation concentration. The saturated concentrations ($c^0$) of adduct products formed from the successive reactions of SCIs with HCOOH are predicted by using the SIMPOL.1 method proposed by Pankow and Asher (2008), and the results are listed in Table S10. As shown in Table S10, the $c^0$ of adduct products involved in the $nCH_2OO$ + HCOOH reaction decreases with increasing the number of $CH_2OO$. According to the Volatility Basis Set (VBS) of organic compounds (Donahue et al., 2012), these adduct products belong to VOC ($c^0 > 3 \times 10^6$ ug/m$^3$). Similarly, the $c^0$ of adduct products included in the nanti-$CH_3CHOO$ + HCOOH, nsyn-$CH_3CHOO$ + HCOOH, and $n(CH_3)_2COO$ + HCOOH reactions decreases when the number of SCIs increases. It deserves mentioning that the adduct products in the nanti-$CH_3CHOO$ + HCOOH and nsyn-$CH_3CHOO$ + HCOOH reactions*

*belong to intermediate volatility organic compounds (IVOC, $300 < c^0 < 3 \times 10^6$ ug/m$^3$) when the number of SCIs is equal to five. However, the adduct products in the n(CH$_3$)$_2$COO + HCOOH reaction become IVOC when the number of (CH$_3$)$_2$COO is greater than or equal to two. Based on the above discussions, it can be concluded that the volatility of adduct products is significantly affected by the number and size of SCIs in the successive reaction of SCIs with HCOOH.*

6. On p. 6, line 145: "saddle point" should be "minimum".

   **Response:** The word "saddle point" has been replaced by "minimum" in the revised manuscript.

7. On p. 6, line 162: "precision" should be "accuracy".

   **Response:** The word "precision" has been replaced by "accuracy" in the revised manuscript.

8. On p. 7, line 182: "decomposes" should be "rearranges".

   **Response:** The word "decomposes" has been replaced by "rearranges" in the revised manuscript.

9. On p. 14, lines 341-342, use a non-breaking hyphen.

   **Response:** A non-breaking hyphen has been used in the revised manuscript.

10. On p. 15, line 372, "intermolecular" should be "intramolecular".

   **Response:** The word "intermolecular" has been replaced by "intramolecular" in the revised manuscript.

11. On p. 17, lines 413-414, use a non-breaking hyphen.

   **Response:** A non-breaking hyphen has been used in the revised manuscript.

**References**

[revised manuscript text omitted]

---

## Author Comment (AC2)

Prof. Yu Huang
State Key Lab of Loess and Quaternary Geology
Institute of Earth Environment, Chinese Academy
of Sciences, Xi'an, 710061, China
Tel./Fax: (86) 29-62336261
E-mail: huangyu@ieecas.cn

Aug. 25, 2022

Dear Prof. Kourtchev,

**Revision for Manuscript ACP-2022-376**

We thank you very much for giving us the opportunity to revise our manuscript. We highly appreciate the reviewers for their comments and suggestions on the manuscript entitled "**Oligomer formation from the gas-phase reactions of Criegee intermediates with hydroperoxide esters: mechanism and kinetics**". We have made revisions of our manuscript carefully according to the comments and suggestions of reviewers. The revised contents are marked in blue color. The response letter to reviewers is attached at the end of this cover letter.

We hope that the revised manuscript can meet the requirement of Atmospheric Chemistry & Physics. Any further modifications or revisions, please do not hesitate to contact us.

Look forward to hearing from you as soon as possible.

Best regards,

Yu Huang

**Comments of reviewer #2**

1. However, a deeper discussion is required for the data in this paper. For example, in lines 263-266 "At room temperature, $k_{tot}$ is estimated to be $3.6 \times 10^{-10}$ cm$^3$ molecule$^{-1}$ s$^{-1}$, which is greater by a factor of ~3 than that reported by Welz et al. (2014) ($[1.1 \pm 0.1] \times 10\text{-}10$ cm$^3$ molecule$^{-1}$ s$^{-1}$), Chung et al. (2019) ($[1.4 \pm 0.3] \times 10^{-10}$ cm$^3$ molecule$^{-1}$ s$^{-1}$), and Peltola et al. (2020) ($[1.0 \pm 0.03] \times 10^{-10}$ cm$^3$ molecule$^{-1}$ s$^{-1}$)". What is the reason for the difference of the k value about three times?

**Response:** In the original manuscript, the rate coefficients for the barrierless reactions are calculated by employing the variational transition state theory (VTST), and the rate coefficients for the bimolecular reactions with the tight transition states are computed by using the canonical transition state theory (CTST) along with one-dimensional asymmetric Eckart tunneling correction. For the initiation reactions of distinct stabilized Criegee intermediates (SCIs) with HCOOH, there are four possible pathways, namely (1) 1,4 O-H insertion (Entry 1), (2) 1,2 O-H insertion (Entry 2), (3) C-H insertion (Entry 3), and (4) C=O cycloaddition (Entry 4), in which Entry 1 is barrierless and Entry 2-4 have the tight transition states. The total rate coefficient for the reaction of SCIs with HCOOH is equal to the sum of the rate coefficient of each pathway. For the barrierless 1,4 O-H insertion reaction, the VTST is approximated with a Morse potential function, $V(R) = D_e\{1\text{-exp}[-\beta(R\text{-}R_e)]\}^2$, along with an anisotropy potential function to stand for the minimum energy path, which is used to calculate the rate coefficients (Raghunath et al., 2017). Here, $D_e$ is the bond energy excluding the zero-point energy, $R$ is the reaction coordinate, and $R_e$ is the equilibrium value of $R$. It is assumed that the stretching potential in an anisotropy potential is used in conjunction with a potential form of $V_{anisotropy} = V_0[1\text{-}\cos^2(\theta_1-\theta_{1e}) \times \cos^2(\theta_2-\theta_{2e})]$ (Raghunath et al., 2017). Here, $V_0$ is the stretching potential, which stands for by a Morse potential, $\theta_1$ and $\theta_{1e}$ represent the rotational angle between fragment 1 and the reference axis and the equilibrium bond angle of fragment 1, $\theta_2$ and $\theta_{2e}$ stand for the rotational angle between fragment 2 and the reference axis and the equilibrium bond angle of fragment 2. The association curve for the reaction of 1,4 O-H insertion of SCIs into HCOOH is computed at the M06-2X/6-311+G(2df,2p) level of theory to cover a range from 0.97 to 1.97 Å at step size 0.1 Å for O-H bond and from 1.44 to 2.44 Å at step size 0.1 Å for C-O bond, while other structural parameters are fully optimized. The computed potential energies are fitted to the Morse potential function. However, the calculated rate coefficients for the reactions of SCIs with HCOOH are higher than the prior experimental measurements. The reason is ascribed to the fact

that the approximation of VTST using a Morse potential function in conjunction with an anisotropy potential function is unsuitable to predict the rate coefficients for the barrierless 1,4 O-H insertion reaction.

In the revised manuscript, the rate coefficients for the barrierless reactions are computed by employing the inverse Laplace transformation (ILT) method, and the rate coefficients for the bimolecular reactions with the tight transition states are calculated by utilizing CTST in conjunction with Eckart tunneling correction. The ILT and CTST/Eckart calculations are performed by using the MESMER 6.0 and KiSThelP 2019 programs, respectively (Glowacki et al., 2012; Canneaux et al., 2013). In the ILT treatment, the rotational constants, vibrational frequencies, molecular weights, energies and other input parameters are obtained from the M06-2X/6-311+G(2df,2p) or M06-2X/ma-TZVP methods. For the barrierless reaction of 1,4 O-H insertion of SCIs into HCOOH, SCIs and HCOOH are assigned as the deficient and excess reactants, respectively. The concentration of HCOOH is given a value of $5.0 \times 10^{10}$ molecules $cm^{-3}$ in the simulation, which is taken from the typical concentration of HCOOH in the tropical forest environments (Vereecken, 2012). $N_2$ is applied as the buffer gas. A single exponential down model is employed to simulate the collision transfer ($<\Delta E>_{down} = 200$ $cm^{-1}$). The collisional Lennard-Jones parameters are estimated with the empirical formula described by Gilbert and Smith (1990).

The rate coefficients of each elementary pathway included in the initiation reactions of distinct SCIs with HCOOH are calculated in the temperature range of 273-400 K, as listed in Table S3-S6. As shown in Table S3, the total rate coefficients $k_{tot-CH2OO}$ of $CH_2OO$ reaction with HCOOH are in excess of $1.0 \times 10^{-10}$ $cm^3$ $molecule^{-1}$ $s^{-1}$, and they exhibit a slightly negative temperature dependence in the temperature range studied. $k_{tot-CH2OO}$ is estimated to be $1.4 \times 10^{-10}$ $cm^3$ $molecule^{-1}$ $s^{-1}$ at 298 K, which is in good agreement with the experimental values reported by Welz et al. (2014) ([$1.1 \pm 0.1$] $\times 10^{-10}$), Chung et al. (2019) ([$1.4 \pm 0.3$] $\times 10^{-10}$), and Peltola et al. (2020) ([$1.0 \pm 0.03$] $\times 10^{-10}$). $k(TS_{ent1})$ is approximately equal to $k_{tot-CH2OO}$ in the whole temperature range, and it decreases in the range of $1.7 \times 10^{-10}$ (273 K) to $1.2 \times 10^{-10}$ (400 K) $cm^3$ $molecule^{-1}$ $s^{-1}$ with increasing temperature. $k(TS_{ent1})$ is several orders of magnitude greater than $k(TS_{ent2})$, $k(TS_{ent3})$ and $k(TS_{ent4})$ over the temperature range from 273 to 400 K. The result again shows that the barrierless 1,4 O-H insertion reaction is predominant. Similar conclusion is also obtained from the results of the rate coefficients for the reactions of HCOOH with *anti*-$CH_3CHOO$, *syn*-$CH_3CHOO$ and $(CH_3)_2COO$ (Table S4-S6).

At ambient temperature, the total rate coefficients of HCOOH reactions with *anti*-CH$_3$CHOO, *syn*-CH$_3$CHOO and (CH$_3$)$_2$COO are estimated to be 5.9, 2.7 and 4.8 × 10$^{-10}$ cm$^3$ molecule$^{-1}$ s$^{-1}$, respectively, which are consistent with the prior experimental measurements of 5 ± 3, 2.5 ± 0.3 and 4.5 × 10$^{-10}$ cm$^3$ molecule$^{-1}$ s$^{-1}$ (Welz et al., 2014; Chung et al., 2019; Sipilä et al., 2014).

**Table S3** Rate coefficients (cm$^3$ molecule$^{-1}$ s$^{-1}$) of each elementary pathway involved in the initiation reaction of CH$_2$OO with HCOOH computed at different temperatures

| T/K | $k$ (TS$_{ent1}$) | $k$ (TS$_{ent2}$) | $k$ (TS$_{ent3}$) | $k$ (TS$_{ent4}$) | $k_{tot\text{-}CH2OO}$ |
|---|---|---|---|---|---|
| 273 | 1.7 × 10$^{-10}$ | 3.6 × 10$^{-12}$ | 1.0 × 10$^{-22}$ | 3.6 × 10$^{-12}$ | 1.8 × 10$^{-10}$ |
| 280 | 1.6 × 10$^{-10}$ | 2.9 × 10$^{-12}$ | 1.2 × 10$^{-22}$ | 3.1 × 10$^{-12}$ | 1.7 × 10$^{-10}$ |
| 298 | 1.4 × 10$^{-10}$ | 1.9 × 10$^{-12}$ | 2.2 × 10$^{-22}$ | 2.3 × 10$^{-12}$ | 1.4 × 10$^{-10}$ |
| 300 | 1.4 × 10$^{-10}$ | 1.8 × 10$^{-12}$ | 2.4 × 10$^{-22}$ | 2.2 × 10$^{-12}$ | 1.4 × 10$^{-10}$ |
| 320 | 1.3 × 10$^{-10}$ | 1.2 × 10$^{-12}$ | 4.9 × 10$^{-22}$ | 1.6 × 10$^{-12}$ | 1.3 × 10$^{-10}$ |
| 340 | 1.3 × 10$^{-10}$ | 8.2 × 10$^{-13}$ | 1.0 × 10$^{-21}$ | 1.3 × 10$^{-12}$ | 1.3 × 10$^{-10}$ |
| 360 | 1.2 × 10$^{-10}$ | 5.9 × 10$^{-13}$ | 2.2 × 10$^{-21}$ | 1.0 × 10$^{-12}$ | 1.2 × 10$^{-10}$ |
| 380 | 1.2 × 10$^{-10}$ | 4.5 × 10$^{-13}$ | 4.5 × 10$^{-21}$ | 8.2 × 10$^{-13}$ | 1.2 × 10$^{-10}$ |
| 400 | 1.2 × 10$^{-10}$ | 3.5 × 10$^{-13}$ | 9.0 × 10$^{-21}$ | 6.9 × 10$^{-13}$ | 1.2 × 10$^{-10}$ |

**Table S4** Rate coefficients (cm$^3$ molecule$^{-1}$ s$^{-1}$) of each elementary pathway involved in the initiation reaction of *anti*-CH$_3$CHOO with HCOOH computed at different temperatures

| T/K | $k$ (TS$_{ent1}$-*anti*) | $k$ (TS$_{ent2}$-*anti*) | $k$ (TS$_{ent3}$-*anti*) | $k$ (TS$_{ent4}$-*anti*) | $k_{tot\text{-}anti}$ |
|---|---|---|---|---|---|
| 273 | 5.9 × 10$^{-10}$ | 4.2 × 10$^{-11}$ | 5.5 × 10$^{-22}$ | 6.1 × 10$^{-11}$ | 6.9 × 10$^{-10}$ |
| 280 | 5.7 × 10$^{-10}$ | 3.8 × 10$^{-11}$ | 6.7 × 10$^{-22}$ | 4.9 × 10$^{-11}$ | 6.6 × 10$^{-10}$ |
| 298 | 5.4 × 10$^{-10}$ | 2.3 × 10$^{-11}$ | 1.2 × 10$^{-21}$ | 3.0 × 10$^{-11}$ | 5.9 × 10$^{-10}$ |
| 300 | 5.3 × 10$^{-10}$ | 2.0 × 10$^{-11}$ | 1.3 × 10$^{-21}$ | 2.8 × 10$^{-11}$ | 5.8 × 10$^{-10}$ |
| 320 | 5.0 × 10$^{-10}$ | 1.5 × 10$^{-11}$ | 2.6 × 10$^{-21}$ | 1.7 × 10$^{-11}$ | 5.3 × 10$^{-10}$ |
| 340 | 4.7 × 10$^{-10}$ | 9.4 × 10$^{-12}$ | 5.4 × 10$^{-21}$ | 1.1 × 10$^{-11}$ | 4.9 × 10$^{-10}$ |
| 360 | 4.5 × 10$^{-10}$ | 7.0 × 10$^{-12}$ | 1.1 × 10$^{-20}$ | 7.8 × 10$^{-12}$ | 4.7 × 10$^{-10}$ |
| 380 | 4.4 × 10$^{-10}$ | 3.6 × 10$^{-12}$ | 2.1 × 10$^{-20}$ | 5.6 × 10$^{-12}$ | 4.5 × 10$^{-10}$ |
| 400 | 4.3 × 10$^{-10}$ | 2.0 × 10$^{-12}$ | 4.0 × 10$^{-20}$ | 4.2 × 10$^{-12}$ | 4.4 × 10$^{-10}$ |

**Table S5** Rate coefficients (cm$^3$ molecule$^{-1}$ s$^{-1}$) of each elementary pathway involved in the initiation reaction of *syn*-CH$_3$CHOO with HCOOH computed at different temperatures

| T/K | $k$ (TS$_{ent1}$-*syn*) | $k$ (TS$_{ent2}$-*syn*) | $k$ (TS$_{ent3}$-*syn*) | $k$ (TS$_{ent4}$-*syn*) | $k_{\text{tot-}syn}$ |
|---|---|---|---|---|---|
| 273 | $3.1 \times 10^{-10}$ | $9.5 \times 10^{-13}$ | $4.6 \times 10^{-27}$ | $7.5 \times 10^{-16}$ | $3.1 \times 10^{-10}$ |
| 280 | $2.8 \times 10^{-10}$ | $8.0 \times 10^{-13}$ | $7.1 \times 10^{-27}$ | $6.4 \times 10^{-16}$ | $2.8 \times 10^{-10}$ |
| 298 | $2.7 \times 10^{-10}$ | $5.4 \times 10^{-13}$ | $8.9 \times 10^{-26}$ | $5.5 \times 10^{-16}$ | $2.7 \times 10^{-10}$ |
| 300 | $2.7 \times 10^{-10}$ | $5.2 \times 10^{-13}$ | $9.9 \times 10^{-26}$ | $4.6 \times 10^{-16}$ | $2.7 \times 10^{-10}$ |
| 320 | $2.5 \times 10^{-10}$ | $3.6 \times 10^{-13}$ | $3.0 \times 10^{-25}$ | $3.8 \times 10^{-16}$ | $2.5 \times 10^{-10}$ |
| 340 | $2.5 \times 10^{-10}$ | $2.6 \times 10^{-13}$ | $9.1 \times 10^{-25}$ | $3.1 \times 10^{-16}$ | $2.5 \times 10^{-10}$ |
| 360 | $2.3 \times 10^{-10}$ | $2.0 \times 10^{-13}$ | $2.6 \times 10^{-24}$ | $3.0 \times 10^{-16}$ | $2.3 \times 10^{-10}$ |
| 380 | $2.2 \times 10^{-10}$ | $1.5 \times 10^{-13}$ | $7.2 \times 10^{-24}$ | $2.4 \times 10^{-16}$ | $2.2 \times 10^{-10}$ |
| 400 | $2.2 \times 10^{-10}$ | $1.2 \times 10^{-13}$ | $1.8 \times 10^{-23}$ | $2.2 \times 10^{-16}$ | $2.2 \times 10^{-10}$ |

**Table S6** Rate coefficients ($cm^3$ molecule$^{-1}$ s$^{-1}$) of each elementary pathway involved in the initiation reaction of $(CH_3)_2OO$ with HCOOH computed at different temperatures

| T/K | $k$ (TS$_{ent1}$-*dim*) | $k$ (TS$_{ent2}$-*dim*) | $k$ (TS$_{ent3}$-*dim*) | $k$ (TS$_{ent4}$-*dim*) | $k_{\text{tot-}dim}$ |
|---|---|---|---|---|---|
| 273 | $5.3 \times 10^{-10}$ | $6.8 \times 10^{-12}$ | $1.4 \times 10^{-26}$ | $4.4 \times 10^{-15}$ | $5.4 \times 10^{-10}$ |
| 280 | $5.1 \times 10^{-10}$ | $5.2 \times 10^{-12}$ | $2.2 \times 10^{-26}$ | $4.2 \times 10^{-15}$ | $5.2 \times 10^{-10}$ |
| 298 | $4.8 \times 10^{-10}$ | $2.8 \times 10^{-12}$ | $8.0 \times 10^{-26}$ | $4.0 \times 10^{-15}$ | $4.8 \times 10^{-10}$ |
| 300 | $4.7 \times 10^{-10}$ | $2.6 \times 10^{-12}$ | $9.2 \times 10^{-26}$ | $3.9 \times 10^{-15}$ | $4.7 \times 10^{-10}$ |
| 320 | $4.5 \times 10^{-10}$ | $1.4 \times 10^{-12}$ | $3.6 \times 10^{-25}$ | $3.7 \times 10^{-15}$ | $4.5 \times 10^{-10}$ |
| 340 | $4.2 \times 10^{-10}$ | $8.6 \times 10^{-13}$ | $1.3 \times 10^{-24}$ | $3.6 \times 10^{-15}$ | $4.2 \times 10^{-10}$ |
| 360 | $3.9 \times 10^{-10}$ | $5.5 \times 10^{-13}$ | $4.5 \times 10^{-24}$ | $3.5 \times 10^{-15}$ | $3.9 \times 10^{-10}$ |
| 380 | $3.7 \times 10^{-10}$ | $3.7 \times 10^{-13}$ | $1.4 \times 10^{-23}$ | $3.4 \times 10^{-15}$ | $3.7 \times 10^{-10}$ |
| 400 | $3.7 \times 10^{-10}$ | $2.6 \times 10^{-13}$ | $3.9 \times 10^{-23}$ | $3.4 \times 10^{-15}$ | $3.7 \times 10^{-10}$ |

Corresponding descriptions have been added in the page 7 line 173-190, page 11 line 303-315, page 12 line 330-338 and page 13 line 346-351 of the revised manuscript:

*The rate coefficients for the barrierless reactions are determined by employing the inverse Laplace transformation (ILT) method. The ILT calculations are performed with the MESMER 6.0 program (Glowacki et al., 2012). In the ILT treatment, the rotational constants, vibrational frequencies, molecular weights, energies and other input parameters are obtained from the M06-2X/6-311+G(2df,2p) or M06-2X/ma-TZVP methods. For the barrierless reaction of 1,4 O-H*

*insertion of SCIs into HCOOH, SCIs and HCOOH are assigned as the deficient and excess reactants, respectively. The concentration of HCOOH is given a value of $5.0 \times 10^{10}$ molecules $cm^{-3}$ in the simulation, which is taken from the typical concentration of HCOOH in the tropical forest environments (Vereecken et al., 2012). $N_2$ is applied as the buffer gas. A single exponential down model is employed to simulate the collision transfer ($<\Delta E>_{down} = 200\ cm^{-1}$). The collisional Lennard-Jones parameters are estimated with the empirical formula described by Gilbert and Smith (1990).*

*The rate coefficients for the bimolecular reactions with the tight transition states are calculated by using the canonical transition state theory (CTST) along with one-dimensional asymmetric Eckart tunneling correction (Truhlar et al., 1996; Eckart, 1930). The CTST/Eckart calculations are performed with the KiSThelP 2019 program (Canneaux et al., 2013).*

*The rate coefficients of each elementary pathway included in the initiation reactions of distinct SCIs with HCOOH are calculated in the temperature range of 273-400 K, as listed in Table S3-S6. As shown in Table S3, the total rate coefficients $k_{tot\text{-}CH2OO}$ of $CH_2OO$ reaction with HCOOH are in excess of $1.0 \times 10^{-10}\ cm^3\ molecule^{-1}\ s^{-1}$, and they exhibit a slightly negative temperature dependence in the temperature range studied. $k_{tot\text{-}CH2OO}$ is estimated to be $1.4 \times 10^{-10}\ cm^3\ molecule^{-1}\ s^{-1}$ at 298 K, which is in good agreement with the experimental values reported by Welz et al. (2014) ($[1.1 \pm 0.1] \times 10^{-10}$), Chung et al. (2019) ($[1.4 \pm 0.3] \times 10^{-10}$), and Peltola et al. (2020) ($[1.0 \pm 0.03] \times 10^{-10}$). $k(TS_{ent1})$ is approximately equal to $k_{tot\text{-}CH2OO}$ in the whole temperature range, and it decreases in the range of $1.7 \times 10^{-10}$ (273 K) to $1.2 \times 10^{-10}$ (400 K) $cm^3\ molecule^{-1}\ s^{-1}$ with increasing temperature. $k(TS_{ent1})$ is several orders of magnitude greater than $k(TS_{ent2})$, $k(TS_{ent3})$ and $k(TS_{ent4})$ over the temperature range from 273 to 400 K. The result again shows that the barrierless 1,4 O-H insertion reaction is predominant.*

*Equivalent to the case of $CH_2OO$ reaction with HCOOH, the rate coefficient of each elementary pathway involved in the anti-$CH_3CHOO$ + HCOOH reaction also decreases with the temperature increasing (Table S4). This table shows that Entry 1 is kinetically favored over Entry 2, 3 and 4, and Entry 2 is competitive with Entry 4 in the range 273-400 K. Similar conclusion is also obtained from the results of the rate coefficients for the reactions of syn-$CH_3CHOO$ and $(CH_3)_2COO$ with HCOOH that Entry 1 is the dominant pathway (Table S5-S6). It deserves mentioning that the competition of Entry 2 is significantly greater than that of Entry 4 in the syn-*

*CH₃CHOO + HCOOH and (CH₃)₂COO + HCOOH systems. At ambient temperature, the total rate coefficients of HCOOH reactions with anti-CH₃CHOO, syn-CH₃CHOO and (CH₃)₂COO are estimated to be 5.9, 2.7 and 4.8 × 10⁻¹⁰ cm³ molecule⁻¹ s⁻¹, respectively, which are consistent with the prior experimental measurements of 5 ± 3, 2.5 ± 0.3 and 4.5 × 10⁻¹⁰ cm³ molecule⁻¹ s⁻¹ (Welz et al., 2014; Chung et al., 2019; Sipilä et al., 2014).*

2. Furthermore, this paper should also exhibit some extended discussions about atmospheric implications of these reactions and their products. For example, what is the role of the formed oligomers on the atmosphere? It follows in the requirements of ACP journal "The journal scope is focused on studies with important implications for our understanding of the state and behavior of the atmosphere. Articles with a local focus must clearly explain how the results extend and compare with current knowledge".

**Response:** Based on the Reviewer's suggestion, the atmospheric implication of the reactions of SCIs with hydroperoxide esters and the role of the formed oligomers have been added in the revised manuscript. It is well known that the reactions with trace species (e.g., $H_2O$, HCOOH and $SO_2$) are expected to be the dominant chemical sinks for SCIs in the atmosphere (Taatjes et al., 2013; Long et al., 2016). The relative importance of distinct SCIs reactions with hydroperoxide esters and trace species is taken into account. In the present study, the hydroperoxymethyl formate (
[revised manuscript text omitted]

| | formula | $P^0$ (atm) | $c^0$ (ug/m$^3$) |
|---|---|---|---|
| n CH$_2$OO + HCOOH | | | |
| n = 1 | HC(O)OCH$_2$OOH | 2.12 × 10$^{-3}$ | 7.86 × 10$^7$ |
| n = 2 | HC(O)O(CH$_2$OO)$_2$H | 3.80 × 10$^{-3}$ | 3.99 × 10$^7$ |
| n = 3 | HC(O)O(CH$_2$OO)$_3$H | 4.43 × 10$^{-3}$ | 3.91 × 10$^7$ |

| | | | |
|---|---|---|---|
| n = 4 | HC(O)O(CH$_2$OO)$_4$H | $4.21 \times 10^{-3}$ | $3.29 \times 10^7$ |
| n = 5 | HC(O)O(CH$_2$OO)$_5$H | $3.59 \times 10^{-3}$ | $2.12 \times 10^7$ |
| n  *anti*-CH$_3$CHOO  + HCOOH | | | |
| n = 1 | HC(O)OCH(CH$_3$)OOH | $1.25 \times 10^{-3}$ | $8.32 \times 10^6$ |
| n = 2 | HC(O)O(CH(CH$_3$)OO)$_2$H | $1.13 \times 10^{-3}$ | $7.57 \times 10^6$ |
| n = 3 | HC(O)O(CH(CH$_3$)OO)$_3$H | $7.12 \times 10^{-4}$ | $6.49 \times 10^6$ |
| n = 4 | HC(O)O(CH(CH$_3$)OO)$_4$H | $3.90 \times 10^{-4}$ | $4.50 \times 10^6$ |
| n = 5 | HC(O)O(CH(CH$_3$)OO)$_5$H | $2.01 \times 10^{-4}$ | $2.81 \times 10^6$ |
| n  *syn*-CH$_3$CHOO  + HCOOH | | | |
| n = 1 | HC(O)OCH(CH$_3$)OOH | $1.25 \times 10^{-3}$ | $8.32 \times 10^6$ |
| n = 2 | HC(O)O(CH(CH$_3$)OO)$_2$H | $1.13 \times 10^{-3}$ | $7.57 \times 10^6$ |
| n = 3 | HC(O)O(CH(CH$_3$)OO)$_3$H | $7.12 \times 10^{-4}$ | $6.49 \times 10^6$ |
| n = 4 | HC(O)O(CH(CH$_3$)OO)$_4$H | $3.90 \times 10^{-4}$ | $4.50 \times 10^6$ |
| n = 5 | HC(O)O(CH(CH$_3$)OO)$_5$H | $2.01 \times 10^{-4}$ | $2.81 \times 10^6$ |
| n (CH$_3$)$_2$COO + HCOOH | | | |
| n = 1 | HC(O)OC(CH$_3$)$_2$OOH | $7.23 \times 10^{-4}$ | $3.50 \times 10^6$ |
| n = 2 | HC(O)O(C(CH$_3$)$_2$OO)$_2$H | $3.50 \times 10^{-4}$ | $2.74 \times 10^6$ |
| n = 3 | HC(O)O(C(CH$_3$)$_2$OO)$_3$H | $1.27 \times 10^{-4}$ | $1.38 \times 10^6$ |
| n = 4 | HC(O)O(C(CH$_3$)$_2$OO)$_4$H | $4.27 \times 10^{-5}$ | $5.90 \times 10^5$ |
| n = 5 | HC(O)O(C(CH$_3$)$_2$OO)$_5$H | $1.40 \times 10^{-5}$ | $2.36 \times 10^5$ |

Corresponding descriptions have been added in the page 23 line 573-590, page 24 line 591-610, page 27 line 645-671 and page 28 line 672-682 of the revised manuscript:

[revised manuscript text omitted]

3. Hence, as a quick assessment, some deeper and extended discussions should be required and strengthened, such as the nature of the reactions, the detailed atmospheric implications, if this paper

is published in the ACP journal.

**Response:** Based on the Reviewer's suggestion, the deeper discussions on the nature of the reactions of SCIs with hydroperoxide esters have been added in the revised manuscript. A schematic potential energy surface (
[revised manuscript text omitted]

---

## Author Comment (AC3)

Prof. Yu Huang
State Key Lab of Loess and Quaternary Geology
Institute of Earth Environment, Chinese Academy
of Sciences, Xi'an, 710061, China
Tel./Fax: (86) 29-62336261
E-mail: huangyu@ieecas.cn

Aug. 25, 2022

Dear Prof. Kourtchev,

**Revision for Manuscript ACP-2022-376**

We thank you very much for giving us the opportunity to revise our manuscript. We highly appreciate the reviewers for their comments and suggestions on the manuscript entitled "**Oligomer formation from the gas-phase reactions of Criegee intermediates with hydroperoxide esters: mechanism and kinetics**". We have made revisions of our manuscript carefully according to the comments and suggestions of reviewers. The revised contents are marked in blue color. The response letter to reviewers is attached at the end of this cover letter.

We hope that the revised manuscript can meet the requirement of Atmospheric Chemistry & Physics. Any further modifications or revisions, please do not hesitate to contact us.

Look forward to hearing from you as soon as possible.

Best regards,

Yu Huang

**Comments of reviewer #3**

1. For Entry 1 of the initiation reaction, how is it validated that 1,4 O-H insertion is barrierless? Is there a multi-point potential energy surface showing that no barrier is found along the reaction coordinate?

   **Response:** Based on the Reviewer's suggestion, the relevance descriptions on the barrierless 1,4 O-H insertion reactions have been added in the revised manuscript. The potential energy surface (PES) of the initiation reactions of distinct stabilized Criegee intermediates (SCIs) ($CH_2OO$, *syn-*, *anti-*$CH_3CHOO$ and $(CH_3)_2COO$) with HCOOH is drawn in Fig. 1. As shown in Fig. 1, the bimolecular reaction of distinct SCIs with HCOOH proceeds via four possible pathways, namely (1) 1,4 O-H insertion (Entry 1), (2) 1,2 O-H insertion (Entry 2), (3) C-H insertion (Entry 3), and (4) C=O cycloaddition (Entry 4). For Entry 1, the addition reaction of $CH_2OO$ with HCOOH proceeds through the 1,4 O-H insertion of $CH_2OO$ into HCOOH to form a hydroperoxide ester HC(O)O-$CH_2OO$-H with a exoergicity of 37.6 kcal·$mol^{-1}$. The formation of HC(O)O-$CH_2OO$-H is obtained through a concerted process of $O_2$-$H_2$ bond breaking in the HCOOH and $O_4$-$H_2$ and $C_2$-$O_1$ bonds forming. Despite an attempt by various methods, the corresponding transition state is still not located in the effort of optimization. To further validate the barrierless process of 1,4 O-H insertion reaction, a relaxed scan over the $O_4$-$H_2$ and $C_2$-$O_1$ bonds is performed at the M06-2X/6-311+G(2df,2p) level of theory. The scans start from the optimized structure of the adduct product HC(O)O-$CH_2OO$-H, and the $O_4$-$H_2$ and $C_2$-$O_1$ bond length are then increased in steps of 0.10 Å. The relaxed scan energy profiles are presented in Fig. S2. As seen in Fig. S2a, the relative energy of the minimum energy path from reactant to product decreases monotonically when the bond length of $O_4$-$H_2$ and $C_2$-$O_1$ bonds decreases, suggesting that the transition state is not exist in the 1,4 O-H insertion reaction of $CH_2OO$ with HCOOH. Similar conclusion is also obtained from the relaxed scan energy profiles for the HCOOH + *anti-*$CH_3CHOO$, HCOOH + *syn-*$CH_3CHOO$ and HCOOH + $(CH_3)_2COO$ (Fig. S2b-d) reactions that 1,4 O-H insertion reactions are barrierless. This conclusion is further supported by the analogous reaction systems that 1,4 O-H insertion reactions of carbonyl oxides with carboxylic acids are a barrierless process including concerted hydrogen atom transfer and new bond formation (Long et al., 2009; Vereecken, 2017; Cabezas and Endo, 2019; Lin et al., 2019; Chhantyal-Pun et al., 2017).

[Figure]

**Figure 1.** Schematic PES for the possible entrance pathways of the initiation reactions of HCOOH with various SCIs (black, pink, blue, and red lines represent 1,4 O-H insertion, 1,2 O-H insertion, C-H insertion, and C=O cycloaddition reactions, respectively)

[Figure]

[Figure]

**Figure S2**. Relaxed scan energy profiles calculated using the M06-2X/6-311+G(2df,2p) method for varying the C-O and O-H bonds in the 1,4-insertion reactions $CH_2OO$ + HCOOH (a), *anti*-$CH_3CHOO$ + HCOOH (b), *syn*-$CH_3CHOO$ + HCOOH (c) and $(CH_3)_2COO$ + HCOOH (d) (the black solid line represents the minimum energy path)

Corresponding descriptions have been added in the page 8 line 215-239 of the revised manuscript:

*The potential energy surface (PES) of distinct SCIs ($CH_2OO$, syn-, anti-$CH_3CHOO$ and $(CH_3)_2COO$) reactions with HCOOH is drawn in Fig. 1. As shown in Fig. 1, the bimolecular reaction of distinct SCIs with HCOOH proceeds via four possible pathways, namely (1) 1,4 O-H insertion (Entry 1), (2) 1,2 O-H insertion (Entry 2), (3) C-H insertion (Entry 3), and (4) C=O cycloaddition (Entry 4). For Entry 1, the addition reaction of $CH_2OO$ with HCOOH proceeds through the 1,4 O-H insertion of $CH_2OO$ into HCOOH to form a hydroperoxide ester HC(O)O-$CH_2OO$-H with a exoergicity of 37.6 kcal·mol^{-1}. The formation of HC(O)O-$CH_2OO$-H is obtained through a concerted process of $O_2$-$H_2$ bond breaking in the HCOOH and $O_4$-$H_2$ and $C_2$-$O_1$ bonds forming. Despite an attempt by various methods, the corresponding transition state is still not located in the effort of optimization. To further validate the barrierless process of 1,4 O-H insertion reaction, a relaxed scan over the $O_4$-$H_2$ and $C_2$-$O_1$ bonds is performed at the M06-2X/6-311+G(2df,2p) level of theory. The scans start from the optimized structure of the adduct product HC(O)O-$CH_2OO$-H, and the $O_4$-$H_2$ and $C_2$-$O_1$ bond length are then increased in steps of 0.10 Å. The relaxed scan energy profiles are presented in Fig. S2. As seen in Fig. S2a, the relative energy of the minimum energy path from reactant to product decreases monotonically when the bond length of $O_4$-$H_2$ and $C_2$-$O_1$ bonds decreases, suggesting that the transition state is not exist in the 1,4 O-H*

*insertion reaction of CH₂OO with HCOOH. Similar conclusion is also obtained from the relaxed scan energy profiles for the HCOOH + anti-CH₃CHOO, HCOOH + syn-CH₃CHOO and HCOOH + (CH₃)₂COO (Fig. S2b-d) reactions that 1,4 O-H insertion reactions are barrierless. This conclusion is further supported by the analogous reaction systems that 1,4 O-H insertion reactions of carbonyl oxides with carboxylic acids are a barrierless process including concerted hydrogen atom transfer and new C-O bond formation (Chhantyal-Pun et al., 2017; Long et al., 2009; Vereecken, 2017; Cabezas and Endo, 2019; Lin et al., 2019).*

2. The calculated $k_{tot}$ in this study is greater by a factor of ~3 than several previous studies. Since this is related to one of the major conclusions of the paper, the authors should carefully validate this result. For example, what could be the reason they underestimate the value? Which value can have a better interpretation of the experimental or atmospheric data?

**Response:** In the original manuscript, the rate coefficients for the barrierless reactions are calculated by employing the variational transition state theory (VTST), and the rate coefficients for the bimolecular reactions with the tight transition states are computed by using the canonical transition state theory (CTST) along with one-dimensional asymmetric Eckart tunneling correction. For the initiation reactions of distinct SCIs with HCOOH, there are four possible pathways, namely (1) 1,4 O-H insertion (Entry 1), (2) 1,2 O-H insertion (Entry 2), (3) C-H insertion (Entry 3), and (4) C=O cycloaddition (Entry 4), in which Entry 1 is barrierless and Entry 2-4 have the tight transition states. The total rate coefficient for the reaction of SCIs with HCOOH is equal to the sum of the rate coefficient of each pathway. For the barrierless 1,4 O-H insertion reaction, the VTST is approximated with a Morse potential function, $V(R) = D_e\{1\text{-}exp[-\beta(R\text{-}R_e)]\}^2$, along with an anisotropy potential function to stand for the minimum energy path, which is used to calculate the rate coefficients (Raghunath et al., 2017). Here, $D_e$ is the bond energy excluding the zero-point energy, $R$ is the reaction coordinate, and $R_e$ is the equilibrium value of $R$. It is assumed that the stretching potential in an anisotropy potential is used in conjunction with a potential form of $V_{anisotropy} = V_0[1\text{-}cos^2(\theta_1\text{-}\theta_{1e}) \times cos^2(\theta_2\text{-}\theta_{2e})]$ (Raghunath et al., 2017). Here, $V_0$ is the stretching potential, which stands for by a Morse potential, $\theta_1$ and $\theta_{1e}$ represent the rotational angle between fragment 1 and the reference axis and the equilibrium bond angle of fragment 1, $\theta_2$ and $\theta_{2e}$ stand for the rotational angle between fragment 2 and the reference axis and the equilibrium bond angle of

fragment 2. The association curve for the reaction of 1,4 O-H insertion of SCIs into HCOOH is computed at the M06-2X/6-311+G(2df,2p) level of theory to cover a range from 0.97 to 1.97 Å at step size 0.1 Å for O-H bond and from 1.44 to 2.44 Å at step size 0.1 Å for C-O bond, while other structural parameters are fully optimized. The computed potential energies are fitted to the Morse potential function. However, the calculated rate coefficients for the reactions of SCIs with HCOOH are higher than the prior experimental measurements. The reason is ascribed to the fact that the approximation of VTST using a Morse potential function in conjunction with an anisotropy potential function is unsuitable to predict the rate coefficients for the barrierless 1,4 O-H insertion reaction.

In the revised manuscript, the rate coefficients for the barrierless reactions are computed by employing the inverse Laplace transformation (ILT) method, and the rate coefficients for the bimolecular reactions with the tight transition states are calculated by utilizing CTST in conjunction with Eckart tunneling correction. The ILT and CTST/Eckart calculations are performed by using the MESMER 6.0 and KiSThelP 2019 programs, respectively (Glowacki et al., 2012; Canneaux et al., 2013). In the ILT treatment, the rotational constants, vibrational frequencies, molecular weights, energies and other input parameters are obtained from the M06-2X/6-311+G(2df,2p) or M06-2X/ma-TZVP methods. For the barrierless reaction of 1,4 O-H insertion of SCIs into HCOOH, SCIs and HCOOH are assigned as the deficient and excess reactants, respectively. The concentration of HCOOH is given a value of $5.0 \times 10^{10}$ molecules cm$^{-3}$ in the simulation, which is taken from the typical concentration of HCOOH in the tropical forest environments (Vereecken et al., 2012). $N_2$ is applied as the buffer gas. A single exponential down model is employed to simulate the collision transfer ($<\Delta E>_{down} = 200$ cm$^{-1}$). The collisional Lennard-Jones parameters are estimated with the empirical formula described by Gilbert and Smith (1990).

The rate coefficients of each elementary pathway included in the initiation reactions of distinct SCIs with HCOOH are calculated in the temperature range of 273-400 K, as listed in Table S3-S6. As shown in Table S3, the total rate coefficients $k_{tot\text{-}CH2OO}$ of CH$_2$OO reaction with HCOOH are in excess of $1.0 \times 10^{-10}$ cm$^3$ molecule$^{-1}$ s$^{-1}$, and they exhibit a slightly negative temperature dependence in the temperature range studied. $k_{tot\text{-}CH2OO}$ is estimated to be $1.4 \times 10^{-10}$ cm$^3$ molecule$^{-1}$ s$^{-1}$ at 298 K, which is in good agreement with the experimental values reported by Welz et al. (2014) ([$1.1 \pm 0.1$] $\times 10^{-10}$), Chung et al. (2019) ([$1.4 \pm 0.3$] $\times 10^{-10}$), and Peltola et al. (2020) ([$1.0 \pm 0.03$] $\times 10^{-10}$). $k(\text{TS}_{ent1})$ is approximately equal to $k_{tot\text{-}CH2OO}$ in the whole temperature range, and it decreases in the

range of $1.7 \times 10^{-10}$ (273 K) to $1.2 \times 10^{-10}$ (400 K) cm$^3$ molecule$^{-1}$ s$^{-1}$ with increasing temperature. $k(\text{TS}_{\text{ent1}})$ is several orders of magnitude greater than $k(\text{TS}_{\text{ent2}})$, $k(\text{TS}_{\text{ent3}})$ and $k(\text{TS}_{\text{ent4}})$ over the temperature range from 273 to 400 K. The result again shows that the barrierless 1,4 O-H insertion reaction is predominant. Similar conclusion is also obtained from the results of the rate coefficients for the reactions of HCOOH with *anti*-CH$_3$CHOO, *syn*-CH$_3$CHOO and (CH$_3$)$_2$COO (Table S4-S6). At ambient temperature, the total rate coefficients of HCOOH reactions with *anti*-CH$_3$CHOO, *syn*-CH$_3$CHOO and (CH$_3$)$_2$COO are estimated to be 5.9, 2.7 and $4.8 \times 10^{-10}$ cm$^3$ molecule$^{-1}$ s$^{-1}$, respectively, which are consistent with the prior experimental measurements of $5 \pm 3$, $2.5 \pm 0.3$ and $4.5 \times 10^{-10}$ cm$^3$ molecule$^{-1}$ s$^{-1}$ (Welz et al., 2014; Chung et al., 2019; Sipilä et al., 2014).

**Table S3** Rate coefficients (cm$^3$ molecule$^{-1}$ s$^{-1}$) of each elementary pathway involved in the initiation reaction of CH$_2$OO with HCOOH computed at different temperatures

| T/K | $k$ (TS$_{\text{ent1}}$) | $k$ (TS$_{\text{ent2}}$) | $k$ (TS$_{\text{ent3}}$) | $k$ (TS$_{\text{ent4}}$) | $k_{\text{tot-CH2OO}}$ |
|---|---|---|---|---|---|
| 273 | $1.7 \times 10^{-10}$ | $3.6 \times 10^{-12}$ | $1.0 \times 10^{-22}$ | $3.6 \times 10^{-12}$ | $1.8 \times 10^{-10}$ |
| 280 | $1.6 \times 10^{-10}$ | $2.9 \times 10^{-12}$ | $1.2 \times 10^{-22}$ | $3.1 \times 10^{-12}$ | $1.7 \times 10^{-10}$ |
| 298 | $1.4 \times 10^{-10}$ | $1.9 \times 10^{-12}$ | $2.2 \times 10^{-22}$ | $2.3 \times 10^{-12}$ | $1.4 \times 10^{-10}$ |
| 300 | $1.4 \times 10^{-10}$ | $1.8 \times 10^{-12}$ | $2.4 \times 10^{-22}$ | $2.2 \times 10^{-12}$ | $1.4 \times 10^{-10}$ |
| 320 | $1.3 \times 10^{-10}$ | $1.2 \times 10^{-12}$ | $4.9 \times 10^{-22}$ | $1.6 \times 10^{-12}$ | $1.3 \times 10^{-10}$ |
| 340 | $1.3 \times 10^{-10}$ | $8.2 \times 10^{-13}$ | $1.0 \times 10^{-21}$ | $1.3 \times 10^{-12}$ | $1.3 \times 10^{-10}$ |
| 360 | $1.2 \times 10^{-10}$ | $5.9 \times 10^{-13}$ | $2.2 \times 10^{-21}$ | $1.0 \times 10^{-12}$ | $1.2 \times 10^{-10}$ |
| 380 | $1.2 \times 10^{-10}$ | $4.5 \times 10^{-13}$ | $4.5 \times 10^{-21}$ | $8.2 \times 10^{-13}$ | $1.2 \times 10^{-10}$ |
| 400 | $1.2 \times 10^{-10}$ | $3.5 \times 10^{-13}$ | $9.0 \times 10^{-21}$ | $6.9 \times 10^{-13}$ | $1.2 \times 10^{-10}$ |

**Table S4** Rate coefficients (cm$^3$ molecule$^{-1}$ s$^{-1}$) of each elementary pathway involved in the initiation reaction of *anti*-CH$_3$CHOO with HCOOH computed at different temperatures

| T/K | $k$ (TS$_{\text{ent1}}$-*anti*) | $k$ (TS$_{\text{ent2}}$-*anti*) | $k$ (TS$_{\text{ent3}}$-*anti*) | $k$ (TS$_{\text{ent4}}$-*anti*) | $k_{\text{tot-}anti}$ |
|---|---|---|---|---|---|
| 273 | $5.9 \times 10^{-10}$ | $4.2 \times 10^{-11}$ | $5.5 \times 10^{-22}$ | $6.1 \times 10^{-11}$ | $6.9 \times 10^{-10}$ |
| 280 | $5.7 \times 10^{-10}$ | $3.8 \times 10^{-11}$ | $6.7 \times 10^{-22}$ | $4.9 \times 10^{-11}$ | $6.6 \times 10^{-10}$ |
| 298 | $5.4 \times 10^{-10}$ | $2.3 \times 10^{-11}$ | $1.2 \times 10^{-21}$ | $3.0 \times 10^{-11}$ | $5.9 \times 10^{-10}$ |
| 300 | $5.3 \times 10^{-10}$ | $2.0 \times 10^{-11}$ | $1.3 \times 10^{-21}$ | $2.8 \times 10^{-11}$ | $5.8 \times 10^{-10}$ |
| 320 | $5.0 \times 10^{-10}$ | $1.5 \times 10^{-11}$ | $2.6 \times 10^{-21}$ | $1.7 \times 10^{-11}$ | $5.3 \times 10^{-10}$ |
| 340 | $4.7 \times 10^{-10}$ | $9.4 \times 10^{-12}$ | $5.4 \times 10^{-21}$ | $1.1 \times 10^{-11}$ | $4.9 \times 10^{-10}$ |

| | | | | |
|---|---|---|---|---|
| 360 | $4.5 \times 10^{-10}$ | $7.0 \times 10^{-12}$ | $1.1 \times 10^{-20}$ | $7.8 \times 10^{-12}$ | $4.7 \times 10^{-10}$ |
| 380 | $4.4 \times 10^{-10}$ | $3.6 \times 10^{-12}$ | $2.1 \times 10^{-20}$ | $5.6 \times 10^{-12}$ | $4.5 \times 10^{-10}$ |
| 400 | $4.3 \times 10^{-10}$ | $2.0 \times 10^{-12}$ | $4.0 \times 10^{-20}$ | $4.2 \times 10^{-12}$ | $4.4 \times 10^{-10}$ |

**Table S5** Rate coefficients ($cm^3$ molecule$^{-1}$ s$^{-1}$) of each elementary pathway involved in the initiation reaction of $syn$-$CH_3CHOO$ with HCOOH computed at different temperatures

| T/K | $k$ ($TS_{ent1}$-$syn$) | $k$ ($TS_{ent2}$-$syn$) | $k$ ($TS_{ent3}$-$syn$) | $k$ ($TS_{ent4}$-$syn$) | $k_{tot\text{-}syn}$ |
|---|---|---|---|---|---|
| 273 | $3.1 \times 10^{-10}$ | $9.5 \times 10^{-13}$ | $4.6 \times 10^{-27}$ | $7.5 \times 10^{-16}$ | $3.1 \times 10^{-10}$ |
| 280 | $2.8 \times 10^{-10}$ | $8.0 \times 10^{-13}$ | $7.1 \times 10^{-27}$ | $6.4 \times 10^{-16}$ | $2.8 \times 10^{-10}$ |
| 298 | $2.7 \times 10^{-10}$ | $5.4 \times 10^{-13}$ | $8.9 \times 10^{-26}$ | $5.5 \times 10^{-16}$ | $2.7 \times 10^{-10}$ |
| 300 | $2.7 \times 10^{-10}$ | $5.2 \times 10^{-13}$ | $9.9 \times 10^{-26}$ | $4.6 \times 10^{-16}$ | $2.7 \times 10^{-10}$ |
| 320 | $2.5 \times 10^{-10}$ | $3.6 \times 10^{-13}$ | $3.0 \times 10^{-25}$ | $3.8 \times 10^{-16}$ | $2.5 \times 10^{-10}$ |
| 340 | $2.5 \times 10^{-10}$ | $2.6 \times 10^{-13}$ | $9.1 \times 10^{-25}$ | $3.1 \times 10^{-16}$ | $2.5 \times 10^{-10}$ |
| 360 | $2.3 \times 10^{-10}$ | $2.0 \times 10^{-13}$ | $2.6 \times 10^{-24}$ | $3.0 \times 10^{-16}$ | $2.3 \times 10^{-10}$ |
| 380 | $2.2 \times 10^{-10}$ | $1.5 \times 10^{-13}$ | $7.2 \times 10^{-24}$ | $2.4 \times 10^{-16}$ | $2.2 \times 10^{-10}$ |
| 400 | $2.2 \times 10^{-10}$ | $1.2 \times 10^{-13}$ | $1.8 \times 10^{-23}$ | $2.2 \times 10^{-16}$ | $2.2 \times 10^{-10}$ |

**Table S6** Rate coefficients ($cm^3$ molecule$^{-1}$ s$^{-1}$) of each elementary pathway involved in the initiation reaction of $(CH_3)_2OO$ with HCOOH computed at different temperatures

| T/K | $k$ ($TS_{ent1}$-$dim$) | $k$ ($TS_{ent2}$-$dim$) | $k$ ($TS_{ent3}$-$dim$) | $k$ ($TS_{ent4}$-$dim$) | $k_{tot\text{-}dim}$ |
|---|---|---|---|---|---|
| 273 | $5.3 \times 10^{-10}$ | $6.8 \times 10^{-12}$ | $1.4 \times 10^{-26}$ | $4.4 \times 10^{-15}$ | $5.4 \times 10^{-10}$ |
| 280 | $5.1 \times 10^{-10}$ | $5.2 \times 10^{-12}$ | $2.2 \times 10^{-26}$ | $4.2 \times 10^{-15}$ | $5.2 \times 10^{-10}$ |
| 298 | $4.8 \times 10^{-10}$ | $2.8 \times 10^{-12}$ | $8.0 \times 10^{-26}$ | $4.0 \times 10^{-15}$ | $4.8 \times 10^{-10}$ |
| 300 | $4.7 \times 10^{-10}$ | $2.6 \times 10^{-12}$ | $9.2 \times 10^{-26}$ | $3.9 \times 10^{-15}$ | $4.7 \times 10^{-10}$ |
| 320 | $4.5 \times 10^{-10}$ | $1.4 \times 10^{-12}$ | $3.6 \times 10^{-25}$ | $3.7 \times 10^{-15}$ | $4.5 \times 10^{-10}$ |
| 340 | $4.2 \times 10^{-10}$ | $8.6 \times 10^{-13}$ | $1.3 \times 10^{-24}$ | $3.6 \times 10^{-15}$ | $4.2 \times 10^{-10}$ |
| 360 | $3.9 \times 10^{-10}$ | $5.5 \times 10^{-13}$ | $4.5 \times 10^{-24}$ | $3.5 \times 10^{-15}$ | $3.9 \times 10^{-10}$ |
| 380 | $3.7 \times 10^{-10}$ | $3.7 \times 10^{-13}$ | $1.4 \times 10^{-23}$ | $3.4 \times 10^{-15}$ | $3.7 \times 10^{-10}$ |
| 400 | $3.7 \times 10^{-10}$ | $2.6 \times 10^{-13}$ | $3.9 \times 10^{-23}$ | $3.4 \times 10^{-15}$ | $3.7 \times 10^{-10}$ |

Corresponding descriptions have been added in the page 7 line 173-190, page 11 line 303-315, page 12 line 330-338 and page 13 line 346-351 of the revised manuscript:

The rate coefficients for the barrierless reactions are determined by employing the inverse Laplace transformation (ILT) method. The ILT calculations are performed with the MESMER 6.0 program (Glowacki et al., 2012). In the ILT treatment, the rotational constants, vibrational frequencies, molecular weights, energies and other input parameters are obtained from the M06-2X/6-311+G(2df,2p) or M06-2X/ma-TZVP methods. For the barrierless reaction of 1,4 O-H insertion of SCIs into HCOOH, SCIs and HCOOH are assigned as the deficient and excess reactants, respectively. The concentration of HCOOH is given a value of $5.0 \times 10^{10}$ molecules $cm^{-3}$ in the simulation, which is taken from the typical concentration of HCOOH in the tropical forest environments (Vereecken et al., 2012). $N_2$ is applied as the buffer gas. A single exponential down model is employed to simulate the collision transfer ($<\Delta E>_{down} = 200\ cm^{-1}$). The collisional Lennard-Jones parameters are estimated with the empirical formula described by Gilbert and Smith (1990).

The rate coefficients for the bimolecular reactions with the tight transition states are calculated by using the canonical transition state theory (CTST) along with one-dimensional asymmetric Eckart tunneling correction (Truhlar et al., 1996; Eckart, 1930). The CTST/Eckart calculations are performed with the KiSThelP 2019 program (Canneaux et al., 2013).

The rate coefficients of each elementary pathway included in the initiation reactions of distinct SCIs with HCOOH are calculated in the temperature range of 273-400 K, as listed in Table S3-S6. As shown in Table S3, the total rate coefficients $k_{tot-CH2OO}$ of $CH_2OO$ reaction with HCOOH are in excess of $1.0 \times 10^{-10}$ $cm^3$ molecule$^{-1}$ s$^{-1}$, and they exhibit a slightly negative temperature dependence in the temperature range studied. $k_{tot-CH2OO}$ is estimated to be $1.4 \times 10^{-10}$ $cm^3$ molecule$^{-1}$ s$^{-1}$ at 298 K, which is in good agreement with the experimental values reported by Welz et al. (2014) ([$1.1 \pm 0.1$] $\times 10^{-10}$), Chung et al. (2019) ([$1.4 \pm 0.3$] $\times 10^{-10}$), and Peltola et al. (2020) ([$1.0 \pm 0.03$] $\times 10^{-10}$). $k(TS_{ent1})$ is approximately equal to $k_{tot-CH2OO}$ in the whole temperature range, and it decreases in the range of $1.7 \times 10^{-10}$ (273 K) to $1.2 \times 10^{-10}$ (400 K) $cm^3$ molecule$^{-1}$ s$^{-1}$ with increasing temperature. $k(TS_{ent1})$ is several orders of magnitude greater than $k(TS_{ent2})$, $k(TS_{ent3})$ and $k(TS_{ent4})$ over the temperature range from 273 to 400 K. The result again shows that the barrierless 1,4 O-H insertion reaction is predominant.

Equivalent to the case of $CH_2OO$ reaction with HCOOH, the rate coefficient of each elementary pathway involved in the anti-$CH_3CHOO$ + HCOOH reaction also decreases with the

*temperature increasing (Table S4). This table shows that Entry 1 is kinetically favored over Entry 2, 3 and 4, and Entry 2 is competitive with Entry 4 in the range 273-400 K. Similar conclusion is also obtained from the results of the rate coefficients for the reactions of syn-CH₃CHOO and (CH₃)₂COO with HCOOH that Entry 1 is the dominant pathway (Table S5-S6). It deserves mentioning that the competition of Entry 2 is significantly greater than that of Entry 4 in the syn-CH₃CHOO + HCOOH and (CH₃)₂COO + HCOOH systems. At ambient temperature, the total rate coefficients of HCOOH reactions with anti-CH₃CHOO, syn-CH₃CHOO and (CH₃)₂COO are estimated to be 5.9, 2.7 and 4.8 × 10⁻¹⁰ cm³ molecule⁻¹ s⁻¹, respectively, which are consistent with the prior experimental measurements of 5 ± 3, 2.5 ± 0.3 and 4.5 × 10⁻¹⁰ cm³ molecule⁻¹ s⁻¹ (Welz et al., 2014; Chung et al., 2019; Sipilä et al., 2014).*

3. Why are $k(TS_{ent2})$ and $k(TS_{ent4})$ decrease with increasing temperature as they both have positive energy barrier? (Table S3)

**Response:** Based on the Reviewer's suggestion, the relevance descriptions on the negative temperature dependence of $k(TS_{ent2})$ and $k(TS_{ent4})$ in Table S3 have been added in the revised manuscript. The rate coefficients for the bimolecular reactions with the tight transition states are calculated by using the canonical transition state theory (CTST) along with one-dimensional asymmetric Eckart tunneling correction. The initiation reaction of $CH_2OO$ with HCOOH proceeds through four possible pathways, namely (1) 1,4 O-H insertion (Entry 1), (2) 1,2 O-H insertion (Entry 2), (3) C-H insertion (Entry 3), and (4) C=O cycloaddition (Entry 4). A schematic PES for the possible entrance pathways is drawn in Fig. 1. As shown in Fig. 1, the entrance pathway Entry2 consists of two elementary steps: (i) an intermediate IMent2 is formed via a barrierless process; (ii) then, it rearranges to the product Pent2 through a tight transition state TSent2. The whole reaction process can be described as Eq. (1):

$$CH_2OO + HCOOH \underset{k_{-1}}{\overset{k_1}{\rightleftarrows}} IMent2 \xrightarrow{k_2} Pent2 \qquad (1)$$

Assuming the rapid equilibrium is established between the IMent2 and reactants. According to the steady-state approximation (SSA), the total rate coefficient is approximately expressed as Eq. (2):

$$k_{\text{tot}} = \frac{k_1}{k_{-1} + k_2} k_2 \approx \frac{k_1}{k_{-1}} k_2 = K_{\text{eq}} k_2 \tag{2}$$

The equilibrium constant $K_{\text{eq}}$ is written as Eq. (3):

$$K_{\text{eq}} = \sigma \frac{Q_{\text{IM}}(T)}{Q_{\text{R1}}(T) Q_{\text{R2}}(T)} \exp\left(\frac{G_{\text{R}} - G_{\text{IM}}}{RT}\right) \tag{3}$$

where $\sigma$ refers to the reaction symmetry number, $Q_{\text{IM}}(T)$, $Q_{\text{R1}}(T)$ and $Q_{\text{R2}}(T)$ denote the partition functions of intermediate, reactants R1 and R2, which are equal to the multiplication of translational, rotational, vibrational and electronic partition functions ($Q = Q_{\text{rot}} Q_{\text{vib}} Q_{\text{trans}} Q_{\text{elec}}$). $T$ is the temperature in Kelvin, $R$ is the ideal gas constant, $G_{\text{R}}$ and $G_{\text{IM}}$ are the total Gibbs free energies of reactant and intermediate, respectively. Similar methodology is adopted to calculate the rate coefficient of each elementary pathway in Entry 4.

The calculated $K_{\text{eq-ent2}}$, $k_{\text{2-ent2}}$, and $k(\text{TS}_{\text{ent2}})$ ($k(\text{TS}_{\text{ent2}}) = K_{\text{eq-ent2}} \times k_{\text{2-ent2}}$) in Entry 2 are listed in Table S7. This table shows that $K_{\text{eq-ent2}}$ significantly decreases with increasing temperature, and $k_{\text{2-ent2}}$ increases as the temperature is increased. However, the decreased value in $K_{\text{eq-ent2}}$ is greater than the increased value in $k_{\text{2-ent2}}$ under the same temperature range. For example, $K_{\text{eq-ent2}}$ deceases by a factor of 6.3 and $k_{\text{2-ent2}}$ increases by a factor of 2.9 at 298 K compared with the values of $K_{\text{eq-ent2}}$ and $k_{\text{2-ent2}}$ at 273 K. It is therefore that $k(\text{TS}_{\text{ent2}})$ decreases with the temperature increasing. Similar conclusion is also obtained from the results of the rate coefficients in Entry 4 that $k(\text{TS}_{\text{ent4}})$ exhibits a negative temperature dependence in the temperature range studied (Table S8). The aforementioned results imply that $k(\text{TS}_{\text{ent2}})$ and $k(\text{TS}_{\text{ent4}})$ are mediated by the pre-reactive complexes IMent2 and IMent4 in the Entry 2 and 4 of the $CH_2OO + HCOOH$ reaction.

**Table S7** $K_{\text{eq-ent2}}$ ($cm^3$ molecule$^{-1}$), $k_{\text{2-ent2}}$ ($s^{-1}$) and $k(\text{TS}_{\text{ent2}})$ ($cm^3$ molecule$^{-1}$ s$^{-1}$) in Entry 2 computed at different temperatures

| T/K | $K_{\text{eq-ent2}}$ | $k_{\text{2-ent2}}$ | $k(\text{TS}_{\text{ent2}})$ |
|---|---|---|---|
| 273 | $8.2 \times 10^{-17}$ | $4.4 \times 10^4$ | $3.6 \times 10^{-12}$ |
| 280 | $4.7 \times 10^{-17}$ | $6.3 \times 10^4$ | $2.9 \times 10^{-12}$ |
| 298 | $1.3 \times 10^{-17}$ | $1.5 \times 10^5$ | $1.9 \times 10^{-12}$ |
| 300 | $1.1 \times 10^{-17}$ | $1.6 \times 10^5$ | $1.8 \times 10^{-12}$ |
| 320 | $3.2 \times 10^{-18}$ | $3.7 \times 10^5$ | $1.2 \times 10^{-12}$ |
| 340 | $1.1 \times 10^{-18}$ | $7.6 \times 10^5$ | $8.2 \times 10^{-13}$ |

| | | | |
|---|---|---|---|
| 360 | $4.1 \times 10^{-19}$ | $1.5 \times 10^6$ | $5.9 \times 10^{-13}$ |
| 380 | $1.7 \times 10^{-19}$ | $2.6 \times 10^6$ | $4.5 \times 10^{-13}$ |
| 400 | $8.0 \times 10^{-20}$ | $4.4 \times 10^6$ | $3.5 \times 10^{-13}$ |

**Table S8** $K_{eq\text{-}ent4}$ (cm$^3$ molecule$^{-1}$), $k_{2\text{-}ent4}$ (s$^{-1}$) and $k(TS_{ent4})$ (cm$^3$ molecule$^{-1}$ s$^{-1}$) in Entry 4 computed at different temperatures

| T/K | $K_{eq\text{-}ent4}$ | $k_{2\text{-}ent4}$ | $k(TS_{ent4})$ |
|---|---|---|---|
| 273 | $6.3 \times 10^{-20}$ | $5.7 \times 10^7$ | $3.6 \times 10^{-12}$ |
| 280 | $4.5 \times 10^{-20}$ | $7.0 \times 10^7$ | $3.1 \times 10^{-12}$ |
| 298 | $2.0 \times 10^{-20}$ | $1.1 \times 10^8$ | $2.3 \times 10^{-12}$ |
| 300 | $1.8 \times 10^{-20}$ | $1.2 \times 10^8$ | $2.2 \times 10^{-12}$ |
| 320 | $8.4 \times 10^{-21}$ | $1.9 \times 10^8$ | $1.6 \times 10^{-12}$ |
| 340 | $4.3 \times 10^{-21}$ | $2.9 \times 10^8$ | $1.3 \times 10^{-12}$ |
| 360 | $2.4 \times 10^{-21}$ | $4.2 \times 10^8$ | $1.0 \times 10^{-12}$ |
| 380 | $1.4 \times 10^{-21}$ | $5.9 \times 10^8$ | $8.2 \times 10^{-13}$ |
| 400 | $8.8 \times 10^{-22}$ | $7.9 \times 10^8$ | $6.9 \times 10^{-13}$ |

Corresponding descriptions have been added in the page 7 line 186-206 and page 12 line 315-326 of the revised manuscript:

*The rate coefficients for the bimolecular reactions with the tight transition states are calculated by using the canonical transition state theory (CTST) along with one-dimensional asymmetric Eckart tunneling correction (Truhlar et al., 1996; Eckart, 1930). The CTST/Eckart calculations are performed with the KiSThelP 2019 program (Canneaux et al., 2013). As shown in Fig. 1, the entrance pathway Entry2 of R$_1$R$_2$COO reaction with HCOOH consists of two steps: (i) an intermediate IMent2 is formed via a barrierless process; (ii) then, it rearranges to the product Pent2 through a tight transition state TSent2. The whole reaction process can be described as Eq. (1):*

$$R_1R_2COO + HCOOH \underset{k_{-1}}{\overset{k_1}{\rightleftharpoons}} IMent2 \xrightarrow{k_2} Pent2 \qquad (1)$$

*Assuming the rapid equilibrium is established between the IMent2 and reactants. According to the steady-state approximation (SSA), the total rate coefficient is approximately expressed as Eq. (2) (Zhang et al., 2012):*

$$k_{\text{tot}} = \frac{k_1}{k_{-1} + k_2} k_2 \approx \frac{k_1}{k_{-1}} k_2 = K_{\text{eq}} k_2 \qquad (2)$$

*The equilibrium constant $K_{eq}$ is written as Eq. (3):*

$$K_{\text{eq}} = \sigma \frac{Q_{\text{IM}}(T)}{Q_{\text{R1}}(T) Q_{\text{R2}}(T)} \exp\left(\frac{G_{\text{R}} - G_{\text{IM}}}{RT}\right) \qquad (3)$$

*where $\sigma$ refers to reaction symmetry number, $Q_{IM}(T)$, $Q_{R1}(T)$ and $Q_{R2}(T)$ denote the partition functions of intermediate, reactants R1 and R2, which are equal to the multiplication of translational, rotational, vibrational and electronic partition functions ($Q = Q_{rot}Q_{vib}Q_{trans}Q_{elec}$) (Mendes et al., 2014), T is the temperature in Kelvin, R is the ideal gas constant, $G_R$ and $G_{IM}$ are the total Gibbs free energies of reactant and intermediate, respectively.*

*The calculated $K_{eq\text{-}ent2}$, $k_{2\text{-}ent2}$, and $k(TS_{ent2})$ ($k(TS_{ent2}) = K_{eq\text{-}ent2} \times k_{2\text{-}ent2}$) in Entry 2 are listed in Table S7. This table shows that $K_{eq\text{-}ent2}$ significantly decreases with increasing temperature, and $k_{2\text{-}ent2}$ increases as the temperature is increased. However, the decreased value in $K_{eq\text{-}ent2}$ is greater than the increased value in $k_{2\text{-}ent2}$ under the same temperature range. For example, $K_{eq\text{-}ent2}$ deceases by a factor of 6.3 and $k_{2\text{-}ent2}$ increases by a factor of 2.9 at 298 K compared with the values of $K_{eq\text{-}ent2}$ and $k_{2\text{-}ent2}$ at 273 K. It is therefore that $k(TS_{ent2})$ decreases with the temperature increasing. Similar conclusion is also obtained from the results of the rate coefficients in Entry 4 that $k(TS_{ent4})$ exhibits a negative temperature dependence in the temperature range studied (Table S8). The aforementioned results imply that $k(TS_{ent2})$ and $k(TS_{ent4})$ are mediated by the pre-reactive complexes IMent2 and IMent4 in the Entry 2 and 4.*

4. The oligomerization reactions are highly dependent on the concentration of the monomers. Here the monomer are highly reactive SCIs and usually has very low concentration in the atmosphere. It seems that the high exothermicity of the oligomerization reaction results from the "stabilization" of SCIs in oligomerization. Also, the calculated free energies represent standard condition. Could the authors correct the Gibbs free energies by incorporating the atmospheric concentrations of SCIs (i.e., RTln(P/Pref)) to check whether this oligomerization is favored in the atmospheric conditions?

**Response:** Based on the Reviewer's suggestion, the relative importance of distinct SCIs reactions with hydroperoxide esters and trace species (e.g., $H_2O$, HCOOH and $SO_2$) has been added in the revised manuscript. It is well known that the reactions with trace species are expected to be

[revised manuscript text omitted]

Corresponding descriptions have been added in the page 23 line 573-590 and page 24 line 591-610 of the revised manuscript:

*It is of interest to assess whether the reactions of distinct SCIs with HPMF can compete well with the losses to reactions with trace species (e.g., H$_2$O, HCOOH and SO$_2$), because it is well known that the reactions with trace species are expected to be the dominant chemical sinks for SCIs in the atmosphere (Taatjes et al., 2013; Long et al., 2016). The reported concentrations of coreactant, the rate coefficients k, and the effective pseudo-first-order rate constants ($k_{eff}$ = k[coreactant]) for distinct SCI reactions with H$_2$O, HCOOH, SO$_2$, and HPMF are summarized in Table 2. As seen in Table 2, the rate coefficient of a particular SCI reaction with trace species is strongly dependent on its structure. The methyl group substitution may alter the rate coefficient by several to tens of times. The atmospheric concentrations of H$_2$O, HCOOH and SO$_2$ in the tropical forest environments are measured to be 3.9-6.1 × 10$^{17}$, 5.0-10 × 10$^{10}$, and 1.7-9.0 × 10$^{10}$ molecules cm$^{-3}$, respectively (Vereecken, 2012). For the reactions of CH$_2$OO with H$_2$O, HCOOH, and SO$_2$, the experimental rate coefficients are determined to be < 1.5 × 10$^{-15}$, [1.1 ± 0.1] × 10$^{-10}$, and [3.9 ± 0.7] × 10$^{-11}$ cm$^3$ molecule$^{-1}$ s$^{-1}$, respectively (Welz et al., 2012 and 2014; Chao et al., 2015), which translate into $k_{eff(CH2OO+H2O)}$, $k_{eff(CH2OO+HCOOH)}$ and $k_{eff(CH2OO+SO2)}$ of 5.9-9.2 × 10$^2$, 5.5-11, and 0.7-3.5 s$^{-1}$, respectively. The result reveals that the reaction of CH$_2$OO with H$_2$O is the most important bimolecular reaction. $k_{eff(CH2OO+HCOOH)}$ is greater by a factor of 3-8 than $k_{eff(CH2OO+SO2)}$, indicating that the reaction of CH$_2$OO with HCOOH is favored over reaction with SO$_2$. Similar conclusion is also obtained from the results of $k_{eff}$ for the reactions of anti-CH$_3$CHOO, syn-CH$_3$CHOO and (CH$_3$)$_2$COO with H$_2$O, HCOOH and SO$_2$ that SCIs reactions with H$_2$O are faster than with HCOOH, which, in turn, are faster than with SO$_2$.*

*According to the results shown in the Table 2, the room temperature rate coefficient for the reaction of CH$_2$OO with HPMF is calculated to be 2.7 × 10$^{-11}$ cm$^3$ molecule$^{-1}$ s$^{-1}$. However, to the best of our knowledge, the atmospheric concentration of HPMF has not been reported up to now. If we assume that the concentration of HPMF is the same as that of HCOOH, $k_{eff(CH2OO+HPMF)}$ is*

*estimated to be 1.4-2.7 s$^{-1}$, which is significantly lower than $k_{eff(CH2OO+H2O)}$ and $k_{eff(CH2OO+HCOOH)}$. $k_{eff(CH2OO+HPMF)}$ is nearly identical to $k_{eff(CH2OO+SO2)}$, indicating that the CH$_2$OO + HPMF reaction is competitive with the CH$_2$OO + SO$_2$ system. Previous model-measurement studies have estimated the surface-level SCIs concentrations in the range of $1.0 \times 10^4$ to $1.0 \times 10^5$ molecules cm$^{-3}$ (Khan et al., 2018; Novelli et al., 2017). If we assume that the concentration of HPMF is equal to that of SCIs, $k_{eff(CH2OO+HPMF)}$ is calculated to be $2.7\text{-}27 \times 10^{-7}$ s$^{-1}$, which is several orders of magnitude lower than $k_{eff(CH2OO+H2O)}$, $k_{eff(CH2OO+HCOOH)}$ and $k_{eff(CH2OO+SO2)}$. This result indicates that the reaction of CH$_2$OO with HPMF is of less importance. Similar conclusion is also obtained from the reactions of anti-CH$_3$CHOO, syn-CH$_3$CHOO and (CH$_3$)$_2$COO with HPMF. Based on the above discussions, it can be concluded that the relative importance of carbonyl oxides reactions with hydroperoxide esters is significantly dependent on the concentrations of hydroperoxide esters. These reactions may play a certain role in the formation of organic new particle in some regions where low concentration of water vapour and high concentration of hydroperoxide esters occur.*

5. Additionally, it would be helpful if there is some estimation about how much the oligomerization process could contribute to the regional or global SOA.

   **Response:** Sakamoto et al. (2013) investigated the ozonolysis of ethylene in a Teflon bag reactor, and found that CH$_2$OO plays a critical role in the formations of oligomers and secondary organic aerosol (SOA) in the gas phase and particle phase. They proposed a possible formation mechanism for the oligomeric hydroperoxides, which includes the successive addition of CH$_2$OO to hydroperoxides. Sadezky et al. (2008) studied the gas-phase ozonolysis of small enol ethers in a 570 l spherical glass reactor at atmospheric conditions in the absence of seed aerosol. They found that the oligomers composed of Criegee intermediate as the repeated chain unit are the main constituents of SOA. Zhao et al. (2015) studied the ozonolysis of trans-3-hexene in both the static Teflon chamber and glass flow reactor under different relative humidity conditions. It was found that the oligomers having Criegee intermediate as the chain unit are the dominant components of SOA. These findings may help in understanding the potential pathway for the formation of SOA in the atmosphere. However, to the best of our knowledge, the contribution of the oligomerization reaction composed of Criegee intermediate as the chain unit to SOA remains unknown. In the future work, we will adopt the combination of quantum chemistry and numerical simulation to estimate

the contribution of oligomerization reaction to the regional and global SOA.

6. Line 39, "with increasing the number of SCIs" is a bit confusing, it would be better to say "with increasing the number of SCIs added to the oligomer".

    **Response:** The sentence "with increasing the number of SCIs" has been replaced by "with increasing the number of SCIs added to the oligomer" in the revised manuscript.

7. Line 491, "netative" should be "negative".

    **Response:** The word "netative" has been replaced by "negative" in the revised manuscript.

8. Line 499, "neartly" should be "nearly".

    **Response:** The word "neartly" has been replaced by "nearly" in the revised manuscript.

**References**

[revised manuscript text omitted]

---

## Author Response (AR2)

Prof. Yu Huang
State Key Lab of Loess and Quaternary Geology
Institute of Earth Environment, Chinese Academy
of Sciences, Xi'an, 710061, China
Tel./Fax: (86) 29-62336261
E-mail: huangyu@ieecas.cn

Sep. 30, 2022

Dear Prof. Kourtchev,

**Revision for Manuscript ACP-2022-376**

We thank you very much for giving us the opportunity to revise our manuscript. We highly appreciate the reviewers for their comments and suggestions on the manuscript entitled "**Oligomer formation from the gas-phase reactions of Criegee intermediates with hydroperoxide esters: mechanism and kinetics**". We have made revisions of our manuscript carefully according to the comments and suggestions of reviewers. The revised contents are marked in blue color. The response letter to reviewers is attached at the end of this cover letter.

We hope that the revised manuscript can meet the requirement of Atmospheric Chemistry & Physics. Any further modifications or revisions, please do not hesitate to contact us.

Look forward to hearing from you as soon as possible.

Best regards,

Yu Huang

**Comments of reviewer #1**

1. I have serious concerns with the calculation of rate constants for the barrierless 1,4 O-H insertion reactions. In the revised manuscript, the authors report using the Inverse Laplace Transform (ILT) method to compute rate constants for O-H insertion. However, what the ILT method in MESMER does is to convert thermal rate constants, as modeled by an Arrhenius expression, to microcanonical rate coefficients needed for master equation simulations. ILT cannot by itself predict rate constants; it is dependent on thermal rate constants that come from either experiment or theoretical methods like VTST, which was used in the original version of the manuscript.

**Response:** Based on the Reviewer's suggestion, the rate coefficients for the barrierless 1,4 O-H insertion reactions have been recalculated by employing the variable-reaction-coordinate variational transition-state theory (VRC-VTST) in the revised manuscript. The VRC-VTST calculations are performed with the potential surface obtained by direct dynamics using the M06-2X/6-311+G(2df,2p) method. Rate coefficients for the SCIs + HCOOH reactions are calculated using the $E,J$-resolved microcanonical variational theory ($E,J$-μVT) using a single-faceted dividing surface. In the VRC-VTST calculations, the reaction coordinate $s$ is defined by pivot points, which are used to orientate the reactants 1 and 2. $s$ is defined as the minimal value of $r_{ij}$, where $r_{ij}$ is the distance between pivot points $i$ and $j$, $i$ is a pivot point on reactant 1 and $j$ is a pivot point on reactant 2. Two of the pivot points are located at a distance $\pm d$ from the center of mass (COM) of SCIs, and the other two pivot points are located at a distance $\pm d$ from the COM of HCOOH with a fixed length of 0.05, 0.10, 0.15, 0.2 and 0.25 Å. Then, for a given choice of pivot points, the variationally lowest rate coefficients are minimized with respect to $s$ at each of the temperatures. We observed that d=0.05 produces the best variation results and only its value is reported. The best variational results obtained for the barrierless 1,4 O-H insertion reactions are presented in Table S3-S6.

From Table S3, it can be seen that the total rate coefficients $k_{tot-CH2OO}$ of $CH_2OO$ reaction with HCOOH are in excess of $1.0 \times 10^{-10}$ $cm^3$ molecule$^{-1}$ s$^{-1}$, and they exhibit a slightly negative temperature dependence in the temperature range of 273-400 K. At room temperature, $k_{tot-CH2OO}$ is estimated to be $1.29 \times 10^{-10}$ $cm^3$ molecule$^{-1}$ s$^{-1}$, which is in good agreement with the experimental values reported by Welz et al. (2014) ([$1.1 \pm 0.1 \times 10^{-10}$), Chung et al. (2019) ([$1.4 \pm 0.3] \times 10^{-10}$),

and Peltola et al. (2020) ($[1.0 \pm 0.03] \times 10^{-10}$). $k(TS_{ent1})$ is approximately equal to $k_{tot-CH2OO}$ in the whole temperature range, and it decreases in the range of $1.34 \times 10^{-10}$ (273 K) to $1.05 \times 10^{-10}$ (400 K) $cm^3$ molecule$^{-1}$ s$^{-1}$ with increasing temperature. $k(TS_{ent1})$ is several orders of magnitude greater than $k(TS_{ent2})$, $k(TS_{ent3})$ and $k(TS_{ent4})$ over the temperature range from 273 to 400 K. The result again shows that the barrierless 1,4 O-H insertion reaction is predominant. Similar conclusion is also obtained from the results of the rate coefficients for the reactions of HCOOH with *anti*-CH$_3$CHOO, *syn*-CH$_3$CHOO and (CH$_3$)$_2$COO (Table S4-S6). At ambient temperature, the total rate coefficients of HCOOH reactions with *anti*-CH$_3$CHOO, *syn*-CH$_3$CHOO and (CH$_3$)$_2$COO are estimated to be 5.22, 2.18 and $3.97 \times 10^{-10}$ $cm^3$ molecule$^{-1}$ s$^{-1}$, respectively, which are consistent with the prior experimental measurements of $5 \pm 3$, $2.5 \pm 0.3$ and $4.5 \pm 0.9 \times 10^{-10}$ $cm^3$ molecule$^{-1}$ s$^{-1}$ (Welz et al., 2014; Sipilä et al., 2014).

**Table S3** Rate coefficients ($cm^3$ molecule$^{-1}$ s$^{-1}$) of each elementary pathway involved in the initiation reaction of CH$_2$OO with HCOOH computed at different temperatures

| T/K | $k$ (TS$_{ent1}$) | $k$ (TS$_{ent2}$) | $k$ (TS$_{ent3}$) | $k$ (TS$_{ent4}$) | $k_{tot-CH2OO}$ |
|---|---|---|---|---|---|
| 273 | $1.34 \times 10^{-10}$ | $3.56 \times 10^{-12}$ | $1.03 \times 10^{-22}$ | $3.57 \times 10^{-12}$ | $1.41 \times 10^{-10}$ |
| 280 | $1.30 \times 10^{-10}$ | $2.94 \times 10^{-12}$ | $1.22 \times 10^{-22}$ | $3.12 \times 10^{-12}$ | $1.36 \times 10^{-10}$ |
| 298 | $1.25 \times 10^{-10}$ | $1.88 \times 10^{-12}$ | $2.18 \times 10^{-22}$ | $2.26 \times 10^{-12}$ | $1.29 \times 10^{-10}$ |
| 300 | $1.21 \times 10^{-10}$ | $1.80 \times 10^{-12}$ | $2.35 \times 10^{-22}$ | $2.20 \times 10^{-12}$ | $1.25 \times 10^{-10}$ |
| 320 | $1.17 \times 10^{-10}$ | $1.18 \times 10^{-12}$ | $4.86 \times 10^{-22}$ | $1.63 \times 10^{-12}$ | $1.20 \times 10^{-10}$ |
| 340 | $1.12 \times 10^{-10}$ | $8.16 \times 10^{-13}$ | $1.04 \times 10^{-21}$ | $1.26 \times 10^{-12}$ | $1.14 \times 10^{-10}$ |
| 360 | $1.11 \times 10^{-10}$ | $5.92 \times 10^{-13}$ | $2.20 \times 10^{-21}$ | $1.04 \times 10^{-12}$ | $1.13 \times 10^{-10}$ |
| 380 | $1.07 \times 10^{-10}$ | $4.48 \times 10^{-13}$ | $4.52 \times 10^{-21}$ | $8.23 \times 10^{-13}$ | $1.08 \times 10^{-10}$ |
| 400 | $1.05 \times 10^{-10}$ | $3.50 \times 10^{-13}$ | $9.01 \times 10^{-21}$ | $6.91 \times 10^{-13}$ | $1.06 \times 10^{-10}$ |

**Table S4** Rate coefficients ($cm^3$ molecule$^{-1}$ s$^{-1}$) of each elementary pathway involved in the initiation reaction of *anti*-CH$_3$CHOO with HCOOH computed at different temperatures

| T/K | $k$ (TS$_{ent1}$-*anti*) | $k$ (TS$_{ent2}$-*anti*) | $k$ (TS$_{ent3}$-*anti*) | $k$ (TS$_{ent4}$-*anti*) | $k_{tot-anti}$ |
|---|---|---|---|---|---|
| 273 | $4.94 \times 10^{-10}$ | $4.23 \times 10^{-11}$ | $5.53 \times 10^{-22}$ | $6.12 \times 10^{-11}$ | $5.98 \times 10^{-10}$ |
| 280 | $4.82 \times 10^{-10}$ | $3.75 \times 10^{-11}$ | $6.73 \times 10^{-22}$ | $4.92 \times 10^{-11}$ | $5.69 \times 10^{-10}$ |
| 298 | $4.69 \times 10^{-10}$ | $2.34 \times 10^{-11}$ | $1.20 \times 10^{-21}$ | $2.95 \times 10^{-11}$ | $5.22 \times 10^{-10}$ |
| 300 | $4.56 \times 10^{-10}$ | $2.01 \times 10^{-11}$ | $1.29 \times 10^{-21}$ | $2.80 \times 10^{-11}$ | $5.04 \times 10^{-10}$ |

| | | | | | |
|---|---|---|---|---|---|
| 320 | $4.42 \times 10^{-10}$ | $1.48 \times 10^{-11}$ | $2.61 \times 10^{-21}$ | $1.72 \times 10^{-11}$ | $4.74 \times 10^{-10}$ |
| 340 | $4.28 \times 10^{-10}$ | $9.42 \times 10^{-12}$ | $5.36 \times 10^{-21}$ | $1.12 \times 10^{-11}$ | $4.49 \times 10^{-10}$ |
| 360 | $4.27 \times 10^{-10}$ | $7.04 \times 10^{-12}$ | $1.08 \times 10^{-20}$ | $7.77 \times 10^{-12}$ | $4.42 \times 10^{-10}$ |
| 380 | $4.14 \times 10^{-10}$ | $3.64 \times 10^{-12}$ | $2.12 \times 10^{-20}$ | $5.60 \times 10^{-12}$ | $4.23 \times 10^{-10}$ |
| 400 | $4.09 \times 10^{-10}$ | $2.02 \times 10^{-12}$ | $4.01 \times 10^{-20}$ | $4.18 \times 10^{-12}$ | $4.15 \times 10^{-10}$ |

**Table S5** Rate coefficients ($cm^3$ molecule$^{-1}$ s$^{-1}$) of each elementary pathway involved in the initiation reaction of *syn*-$CH_3CHOO$ with HCOOH computed at different temperatures

| T/K | $k$ ($TS_{ent1}$-*syn*) | $k$ ($TS_{ent2}$-*syn*) | $k$ ($TS_{ent3}$-*syn*) | $k$ ($TS_{ent4}$-*syn*) | $k_{tot-syn}$ |
|---|---|---|---|---|---|
| 273 | $2.34 \times 10^{-10}$ | $9.50 \times 10^{-13}$ | $4.58 \times 10^{-27}$ | $7.46 \times 10^{-16}$ | $2.35 \times 10^{-10}$ |
| 280 | $2.25 \times 10^{-10}$ | $8.03 \times 10^{-13}$ | $7.06 \times 10^{-27}$ | $6.43 \times 10^{-16}$ | $2.26 \times 10^{-10}$ |
| 298 | $2.17 \times 10^{-10}$ | $5.37 \times 10^{-13}$ | $8.92 \times 10^{-26}$ | $5.46 \times 10^{-16}$ | $2.18 \times 10^{-10}$ |
| 300 | $2.08 \times 10^{-10}$ | $5.15 \times 10^{-13}$ | $9.94 \times 10^{-26}$ | $4.58 \times 10^{-16}$ | $2.09 \times 10^{-10}$ |
| 320 | $1.99 \times 10^{-10}$ | $3.55 \times 10^{-13}$ | $3.03 \times 10^{-25}$ | $3.78 \times 10^{-16}$ | $1.99 \times 10^{-10}$ |
| 340 | $1.89 \times 10^{-10}$ | $2.57 \times 10^{-13}$ | $9.14 \times 10^{-25}$ | $3.05 \times 10^{-16}$ | $1.89 \times 10^{-10}$ |
| 360 | $1.88 \times 10^{-10}$ | $1.95 \times 10^{-13}$ | $2.64 \times 10^{-24}$ | $3.03 \times 10^{-16}$ | $1.88 \times 10^{-10}$ |
| 380 | $1.79 \times 10^{-10}$ | $1.53 \times 10^{-13}$ | $7.15 \times 10^{-24}$ | $2.43 \times 10^{-16}$ | $1.79 \times 10^{-10}$ |
| 400 | $1.76 \times 10^{-10}$ | $1.24 \times 10^{-13}$ | $1.82 \times 10^{-23}$ | $2.22 \times 10^{-16}$ | $1.76 \times 10^{-10}$ |

**Table S6** Rate coefficients ($cm^3$ molecule$^{-1}$ s$^{-1}$) of each elementary pathway involved in the initiation reaction of $(CH_3)_2OO$ with HCOOH computed at different temperatures

| T/K | $k$ ($TS_{ent1}$-*dim*) | $k$ ($TS_{ent2}$-*dim*) | $k$ ($TS_{ent3}$-*dim*) | $k$ ($TS_{ent4}$-*dim*) | $k_{tot-dim}$ |
|---|---|---|---|---|---|
| 273 | $4.10 \times 10^{-10}$ | $6.81 \times 10^{-12}$ | $1.38 \times 10^{-26}$ | $4.37 \times 10^{-15}$ | $4.17 \times 10^{-10}$ |
| 280 | $4.02 \times 10^{-10}$ | $5.20 \times 10^{-12}$ | $2.24 \times 10^{-26}$ | $4.20 \times 10^{-15}$ | $4.07 \times 10^{-10}$ |
| 298 | $3.94 \times 10^{-10}$ | $2.78 \times 10^{-12}$ | $7.95 \times 10^{-26}$ | $4.03 \times 10^{-15}$ | $3.97 \times 10^{-10}$ |
| 300 | $3.86 \times 10^{-10}$ | $2.61 \times 10^{-12}$ | $9.18 \times 10^{-26}$ | $3.86 \times 10^{-15}$ | $3.89 \times 10^{-10}$ |
| 320 | $3.77 \times 10^{-10}$ | $1.44 \times 10^{-12}$ | $3.63 \times 10^{-25}$ | $3.71 \times 10^{-15}$ | $3.78 \times 10^{-10}$ |
| 340 | $3.68 \times 10^{-10}$ | $8.60 \times 10^{-13}$ | $1.33 \times 10^{-24}$ | $3.55 \times 10^{-15}$ | $3.69 \times 10^{-10}$ |
| 360 | $3.63 \times 10^{-10}$ | $5.48 \times 10^{-13}$ | $4.47 \times 10^{-24}$ | $3.54 \times 10^{-15}$ | $3.64 \times 10^{-10}$ |
| 380 | $3.59 \times 10^{-10}$ | $3.69 \times 10^{-13}$ | $1.37 \times 10^{-23}$ | $3.41 \times 10^{-15}$ | $3.59 \times 10^{-10}$ |

| | | | | | |
|---|---|---|---|---|---|
| 400 | $3.56 \times 10^{-10}$ | $2.60 \times 10^{-13}$ | $3.86 \times 10^{-23}$ | $3.37 \times 10^{-15}$ | $3.56 \times 10^{-10}$ |

Corresponding descriptions have been added in the page 7 line 180-195 and page 12 line 315-329 and page 12 line 333-355 of the revised manuscript:

[revised manuscript text omitted]

2. The authors have done a good job providing a rationale for the trend in the exothermicities of the Criegee intermediate (CI) + HCOOH reactions. In the revised manuscript, the authors report computed enthalpies of formation for the four CIs under consideration, and relate the trend in these formation enthalpies to the trend in the CI + HCOOH reaction enthalpies.

**Response:** Thank the reviewer for his positive comments on the trend in the exothermicities of the reactions of distinct SCIs with HCOOH in the revised manuscript. The exothermicities of 1,4 O-H insertion reactions of distinct SCIs with HCOOH are assessed by the reaction enthalpies ($\Delta_r H_{298}^o$), which are defined as the difference between the enthalpies of formation ($\Delta_f H_{298}^o$) of the

products and reactants ( $\Delta_r H^{\mathrm{o}}_{298} = \sum\limits_{\mathrm{products}} \Delta_f H^{\mathrm{o}}_{298} - \sum\limits_{\mathrm{reactants}} \Delta_f H^{\mathrm{o}}_{298}$ ). The enthalpies of formation of

carbonyl oxides and hydroperoxide esters are estimated by using the isodesmic reaction method, and the results are listed in Table S2. As seen in Table S2, the enthalpies of formation of carbonyl oxides and hydroperoxide esters significantly decrease with increasing the number of methyl groups. Notably, the decreased values in the enthalpies of formation of carbonyl oxides are greater than those of hydroperoxide esters under the condition of the same number of methyl groups. For example, the enthalpy of formation of *anti*-CH$_3$CHOO decreases by 12.95 kcal·mol$^{-1}$ compared to the enthalpy of formation of CH$_2$OO (23.23 kcal·mol$^{-1}$), and the enthalpy of formation of Pent1b decreases by 12.12 kcal·mol$^{-1}$ compared to the enthalpy of formation of Pent1a (-112.08 kcal·mol$^{-1}$). The reaction enthalpies of the reactions of distinct SCIs with HCOOH decrease in the order of -44.69 (CH$_2$OO + HCOOH → Pent1a) < -43.86 (*anti*-CH$_3$CHOO + HCOOH → Pent1b) < -38.13 (*syn*-CH$_3$CHOO + HCOOH → Pent1c) < -37.12 kcal·mol$^{-1}$ ((CH$_3$)$_2$COO + HCOOH → Pent1d), indicating that the reaction enthalpies are highly dependent on the number and location of methyl groups. The trend in reaction enthalpies is consistent with the trend in the enthalpies of formation of carbonyl oxides.

**Table S2** Enthalpies of formation ( $\Delta_f H^{\mathrm{o}}_{298}$ ) for the various carbonyl oxides and hydroperoxide esters computed at the CCSD(T)//M06-2X/6-311+G(2df,2p) level of theory

| Species | Cal (kcal·mol$^{-1}$) | Refs. (kcal·mol$^{-1}$) |
|---|---|---|
| CH$_2$OO | 23.23 | 22.92[a]
 24.59[b] |
| *anti*-CH$_3$OO | 10.28 | |
| *syn*-CH$_3$CHOO | 6.73 | |
| (CH$_3$)$_2$COO | -6.77 | |
| HCOOH | | -90.62 (exp) |
| HC(O)OCH$_2$OOH (Pent1a) | -112.08 | |
| HC(O)OCH(CH$_3$)OOH (Pent1b) | -124.20 | |
| HC(O)OCH(CH$_3$)OOH (Pent1c) | -122.02 | |
| HC(O)OC(CH$_3$)$_2$OOH (Pent1d) | -134.51 | |

Exp is taken from NIST Chemistry Webbook

[a] the value is obtained at the G4 level of theory (Chen et al., 2016)

[b] the value is obtained at the W3-F12 level of theory (Karton et al., 2013)

Corresponding descriptions have been added in the page 10 line 253-283 of the revised manuscript:

*The exothermicities of 1,4 O-H insertion reactions of distinct SCIs with HCOOH are assessed by the reaction enthalpies ($\Delta_r H^o_{298}$), which are defined as the difference between the enthalpies of formation ($\Delta_f H^o_{298}$) of the products and reactants ($\Delta_r H^o_{298} = \sum\limits_{products} \Delta_f H^o_{298} - \sum\limits_{reactants} \Delta_f H^o_{298}$). To the best of our knowledge, there are no literature values available on the enthalpies of formation of carbonyl oxides and hydroperoxide esters except the simplest carbonyl oxide CH₂OO. Therefore, the isodesmic reaction method is adopted to obtain the enthalpies of formation, and the results are listed in Table S2. An isodesmic reaction is a hypothetical reaction, in which the type of chemical bonds in the reactants is the similar as that of chemical bonds in the products. The following isodesmic reaction is constructed because the experimental values of H₂, CH₄ and H₂O are available ($\Delta_f H^o_{298}$ (H₂) = 0.00 kcal·mol⁻¹; $\Delta_f H^o_{298}$ (CH₄) = -17.82 kcal·mol⁻¹; $\Delta_f H^o_{298}$ (H₂O) = -57.79 kcal·mol⁻¹).*

$$SCIs + nH_2 \;\rightarrow\; CH_4 + mH_2O \qquad (4)$$

*As seen in Table S2, the enthalpy of formation of CH₂OO is calculated to be 23.23 kcal·mol⁻¹, which is in good agreement with the available literature values (Chen et al., 2016; Karton et al., 2013). This result implies that the theoretical method employed herein is reasonable to predict the thermochemical parameters. The enthalpies of formation of carbonyl oxides and hydroperoxide esters significantly decrease with increasing the number of methyl groups. Notably, the decreased values in the enthalpies of formation of carbonyl oxides are greater than those of hydroperoxide esters under the condition of the same number of methyl groups. For example, the enthalpy of formation of anti-CH₃CHOO decreases by 12.95 kcal·mol⁻¹ compared to the enthalpy of formation of CH₂OO (23.23 kcal·mol⁻¹), and the enthalpy of formation of Pent1b decreases by 12.12 kcal·mol⁻¹ compared to the enthalpy of formation of Pent1a (-112.08 kcal·mol⁻¹). The reaction enthalpies of the reactions of distinct SCIs with HCOOH decrease in the order of -44.69 (CH₂OO + HCOOH → Pent1a) < -43.86 (anti-CH₃CHOO + HCOOH → Pent1b) < -38.13 (syn-CH₃CHOO + HCOOH → Pent1c) < -37.12 kcal·mol⁻¹ ((CH₃)₂COO + HCOOH → Pent1d), indicating that the reaction enthalpies are highly dependent on the number and location of methyl*

*groups. The trend in reaction enthalpies is consistent with the trend in the enthalpies of formation of carbonyl oxides.*

3. In the original version of the manuscript, the authors had already answered the question I posed: how fast do all four CIs under consideration react with the most important bimolecular reaction partners in the atmosphere. I apologize for missing Table 2.

**Response:** As the Reviewer's said, the reactions with trace species (e.g., $H_2O$, HCOOH and $SO_2$) are expected to be the dominant chemical sinks for the considered all four SCIs ($CH_2OO$, *syn*-$CH_3CHOO$, *anti*-$CH_3CHOO$ and $(CH_3)_2COO$) in the atmosphere. The reported concentrations of coreactant, the rate coefficients $k$, and the effective pseudo-first-order rate constants ($k_{eff} = k$[coreactant]) for the reactions of distinct SCIs with $H_2O$, HCOOH, $SO_2$ are summarized in Table 2. As seen in Table 2, the rate coefficient of a particular SCI reaction with trace species is strongly dependent on its structure. The methyl group substitution may alter the rate coefficient by several to tens of times. The atmospheric concentrations of $H_2O$, HCOOH and $SO_2$ in the tropical forest environments are measured to be $3.9$-$6.1 \times 10^{17}$, $5.0$-$10 \times 10^{10}$, and $1.7$-$9.0 \times 10^{10}$ molecules cm$^{-3}$, respectively (Vereecken et al., 2012). For the reactions of $CH_2OO$ with $H_2O$, HCOOH, and $SO_2$, the experimental rate coefficients are determined to be $< 1.5 \times 10^{-15}$, $[1.1 \pm 0.1] \times 10^{-10}$, and $[3.9 \pm 0.7] \times 10^{-11}$ cm$^3$ molecule$^{-1}$ s$^{-1}$, respectively (Welz et al., 2012 and 2014; Chao et al., 2015), which translate into $k_{eff(CH2OO+H2O)}$, $k_{eff(CH2OO+HCOOH)}$ and $k_{eff(CH2OO+SO2)}$ of $5.9$-$9.2 \times 10^2$, $5.5$-$11$, and $0.7$-$3.5$ s$^{-1}$, respectively. The result reveals that the reaction of $CH_2OO$ with $H_2O$ is the most important bimolecular reaction. $k_{eff(CH2OO+HCOOH)}$ is greater by a factor of 3-8 than $k_{eff(CH2OO+SO2)}$, indicating that $CH_2OO$ reaction with HCOOH is favored over reaction with $SO_2$. Similar conclusion is also obtained from the results of $k_{eff}$ for the reactions of *anti*-$CH_3CHOO$, *syn*-$CH_3CHOO$ and $(CH_3)_2COO$ with $H_2O$, HCOOH and $SO_2$ that SCIs reactions with $H_2O$ are faster than with HCOOH, which, in turn, are faster than with $SO_2$.

**Table 2** The reported concentrations of coreactant, the rate coefficients $k$, and the effective pseudo-first-order rate constants ($k_{eff} = k$[coreactant]) for distinct SCI reactions with HPMF, $H_2O$, HCOOH and $SO_2$ at the tropical forest environments

| SCIs | Coreactant | [Coreactant] (molecules cm$^{-3}$) | $k$ (cm$^3$ molecule$^{-1}$ s$^{-1}$) | $k_{eff}$ (s$^{-1}$) | Reference |
|------|-----------|-----------------------------------|--------------------------------------|---------------------|-----------|
| $CH_2OO$ | $H_2O$ | $3.9$-$6.1 \times 10^{17}$ | $< 1.5 \times 10^{-15}$ | $5.9$-$9.2 \times 10^2$ | Chao et al., (2015) |

| | | | | | |
|---|---|---|---|---|---|
| | HCOOH | $5.0\text{-}10.0 \times 10^{10}$ | $[1.1 \pm 0.1] \times 10^{-10}$ | 5.5-11 | Welz et al., (2014) |
| | $SO_2$ | $1.7\text{-}9.0 \times 10^{10}$ | $[3.9 \pm 0.7] \times 10^{-11}$ | 0.7-3.5 | Welz et al., (2012) |
| | HPMF | - | $2.7 \times 10^{-11}$ | - | This work |
| | $H_2O$ | $3.9\text{-}6.1 \times 10^{17}$ | $[1.0 \pm 0.4] \times 10^{-14}$ | $3.9\text{-}6.1 \times 10^3$ | Taatjes et al., (2013) |
| *anti*-$CH_3CHOO$ | HCOOH | $5.0\text{-}10.0 \times 10^{10}$ | $[5 \pm 3] \times 10^{-10}$ | 25.0-50.0 | Welz et al., (2014) |
| | $SO_2$ | $1.7\text{-}9.0 \times 10^{10}$ | $[6.7 \pm 1.0] \times 10^{-11}$ | 1.1-6.0 | Taatjes et al., (2013) |
| | HPMF | - | $3.3 \times 10^{-10}$ | - | This work |
| | $H_2O$ | $3.9\text{-}6.1 \times 10^{17}$ | $< 4.0 \times 10^{-15}$ | $1.6\text{-}2.4 \times 10^3$ | Taatjes et al., (2013) |
| *syn*-$CH_3CHOO$ | HCOOH | $5.0\text{-}10.0 \times 10^{10}$ | $[2.5 \pm 0.3] \times 10^{-10}$ | 12.5-25.0 | Welz et al., (2014) |
| | $SO_2$ | $1.7\text{-}9.0 \times 10^{10}$ | $[2.4 \pm 0.3] \times 10^{-11}$ | 0.4-2.2 | Taatjes et al., (2013) |
| | HPMF | - | $1.7 \times 10^{-13}$ | - | This work |
| | $H_2O$ | $3.9\text{-}6.1 \times 10^{17}$ | $< 1.5 \times 10^{-16}$ | 58.5-91.5 | Huang et al., (2015) |
| $(CH_3)_2COO$ | HCOOH | $5.0\text{-}10.0 \times 10^{10}$ | $4.5 \times 10^{-10}$ | 22.5-45.0 | Sipilä et al., (2014) |
| | $SO_2$ | $1.7\text{-}9.0 \times 10^{10}$ | $1.3 \times 10^{-10}$ | 2.2-11.7 | Huang et al., (2015) |
| | HPMF | - | $2.2 \times 10^{-11}$ | - | This work |
| | $H_2O$ | $3.9\text{-}6.1 \times 10^{17}$ | $< 4.0 \times 10^{-17}$ | 15.6-24.4 | Caravan et al., (2020) |
| *syn-trans*-MVK-OO | HCOOH | $5.0\text{-}10.0 \times 10^{10}$ | $[3.0 \pm 0.1] \times 10^{-10}$ | 15.0-30.0 | Caravan et al., (2020) |
| | $SO_2$ | $1.7\text{-}9.0 \times 10^{10}$ | $[4.2 \pm 0.6] \times 10^{-11}$ | 0.7-3.8 | Caravan et al., (2020) |
| | HPMF | - | $3.0 \times 10^{-11}$ | - | This work |

Corresponding descriptions have been revised in the page 23 line 581-603 of the revised manuscript:

*It is of interest to assess whether the reactions of distinct SCIs with HPMF can compete well*

*with the losses to reactions with trace species (e.g., $H_2O$, HCOOH and $SO_2$), because it is well known that the reactions with trace species are expected to be the dominant chemical sinks for SCIs in the atmosphere (Taatjes et al., 2013; Long et al., 2016). The reported concentrations of coreactant, the rate coefficients k, and the effective pseudo-first-order rate constants ($k_{eff}$ = k[coreactant]) for the reactions of distinct SCIs with $H_2O$, HCOOH, $SO_2$, and HPMF are summarized in Table 2. As seen in Table 2, the rate coefficient of a particular SCI reaction with trace species is strongly dependent on its structure. The methyl group substitution may alter the rate coefficient by several to tens of times. The atmospheric concentrations of $H_2O$, HCOOH and $SO_2$ in the tropical forest environments are measured to be 3.9-6.1 × $10^{17}$, 5.0-10 × $10^{10}$, and 1.7-9.0 × $10^{10}$ molecules cm$^{-3}$, respectively (Vereecken et al., 2012). For the reactions of $CH_2OO$ with $H_2O$, HCOOH, and $SO_2$, the experimental rate coefficients are determined to be < 1.5 × $10^{-15}$, [1.1 ± 0.1] × $10^{-10}$, and [3.9 ± 0.7] × $10^{-11}$ cm$^3$ molecule$^{-1}$ s$^{-1}$, respectively (Welz et al., 2012 and 2014; Chao et al., 2015), which translate into $k_{eff(CH2OO+H2O)}$, $k_{eff(CH2OO+HCOOH)}$ and $k_{eff(CH2OO+SO2)}$ of 5.9-9.2 × $10^2$, 5.5-11, and 0.7-3.5 s$^{-1}$, respectively. The result reveals that the reaction of $CH_2OO$ with $H_2O$ is the most important bimolecular reaction. $k_{eff(CH2OO+HCOOH)}$ is greater by a factor of 3-8 than $k_{eff(CH2OO+SO2)}$, indicating that $CH_2OO$ reaction with HCOOH is favored over reaction with $SO_2$. Similar conclusion is also obtained from the results of $k_{eff}$ for the reactions of anti-$CH_3CHOO$, syn-$CH_3CHOO$ and $(CH_3)_2COO$ with $H_2O$, HCOOH and $SO_2$ that SCIs reactions with $H_2O$ are faster than with HCOOH, which, in turn, are faster than with $SO_2$.*

4. The authors now quantify the pseudo-first-order rate constants for the reaction of CIs with HPMF with the more accurate assumption that the concentration of HPMF will be equal to the atmospheric concentration of CIs.

**Response:** As the Reviewer's said, we make the more accurate assumption that the concentration of HPMF is approximately equal to the atmospheric concentration of SCIs in the calculation of the pseudo-first-order rate constants for the bimolecular reaction of SCIs with HPMF. It is mainly because that the SCIs is the deficient reactant in the bimolecular reaction of SCIs with HCOOH. The competition between SCIs + HPMF and SCIs + trace species (e.g., $H_2O$, HCOOH and $SO_2$) reactions is taken into consideration in the revised manuscript, because it is well known that the reactions with trace species are expected to be the dominant chemical sinks

[revised manuscript text omitted]

5. I commend the authors for the work they have done to include estimates of vapor pressures and saturation concentrations in the revised manuscript. One question I have is why, for the $nCH_2OO$ + HCOOH series, the vapor pressures do not decrease monotonically with increasing n.

**Response:** Based on the Reviewer's suggestion, the vapour pressure of the adduct products formed from the successive reactions of SCIs with HCOOH has been recalculated in the revised manuscript. To further evaluate the reliability of the considered methods for the calculations of vapor pressure, some selected compounds with experimental data are calculated by using the combination of boiling point and vapour pressure method proposed by Nannoolal et al. (2004 and 2008) (Nan-Nan) and the EVAPORATION method proposed by Compernolle et al. (2011) The calculated results are listed in Table R1. This table shows that the saturated vapour pressure ($P^0$) obtained using the EVAPORATION method is consistent with the experimentally reported ones. The $P^0$ obtained using the Nan-Nan method is about one order of magnitude greater than the experimental data, suggesting that the Nan-Nan method overestimates the saturated vapour pressure. Therefore, in the revised manuscript, the saturated vapour pressure of adduct products at room temperature is estimated by using the EVAPORATION method, and the results are summarized in Table S8. As show in Table S8, the $P^0$ of the adduct products decreases significantly as the number of SCIs is increased. Notably, the $P^0$ of the adduct products decreases when the size of SCIs increases. For example, the $P^0$ of the adduct product $HC(O)O(CH_2OO)_3H$ in the $nCH_2OO$ + HCOOH reaction is estimated to be $3.41 \times 10^{-5}$ atm, which is greater than those of the corresponding adduct products in the $n$*anti*-$CH_3CHOO$ + HCOOH ($4.73 \times 10^{-6}$ atm), $n$*syn*-$CH_3CHOO$ + HCOOH ($4.73 \times 10^{-6}$ atm), and $n(CH_3)_2COO$ + HCOOH ($1.03 \times 10^{-6}$ atm) reactions by 7.21, 7.21 and 33.11 times, respectively.

A classify scheme of various organic compounds is based on their volatility, as presented by Donahue et al. (2012) The volatility of organic compounds is described by their effective saturation concentrations. The saturated concentrations ($c^0$) of the adduct products formed from the successive reactions of SCIs with HCOOH are listed in Table S8. As shown in Table S8, the $c^0$ of the adduct products decrease significantly as the number of SCIs is increased. It deserves mentioning that the $c^0$ of the adduct products decrease with increasing the size of SCIs. For the $nCH_2OO$ + HCOOH reaction, the $c^0$ of the adduct products are estimated to be $1.03 \times 10^8$ (n=1), $5.42 \times 10^6$ (n=2), $2.53 \times 10^5$ (n=3), $1.11 \times 10^4$ (n=4) and $4.67 \times 10^2$ (n=5) ug/m$^3$, respectively.

According to the Volatility Basis Set (VBS) of organic compounds (Donahue et al., 2012), the adduct products belong to volatile organic compounds (VOC, $c^0 > 3 \times 10^6$ ug/m$^3$) when the number of SCIs is less than or equal to two, while they belong to intermediate volatility organic compounds (IVOC, $300 < c^0 < 3 \times 10^6$ ug/m$^3$) when the number of SCIs is greater than or equal to three. Similarly, the adduct products in the n$anti$-CH$_3$CHOO + HCOOH, n$syn$-CH$_3$CHOO + HCOOH, and n(CH$_3$)$_2$COO + HCOOH reactions belong to IVOC when the number of SCIs ranges from 2 to 4, whereas they belong to semivolatile organic compounds (SVOC, $0.3 < c^0 < 300$ ug/m$^3$) when the number of SCIs is equal to 5. Based on the above discussions, it can be concluded that the volatility of the adduct products is significantly affected by the number and size of SCIs in the successive reaction of SCIs with HCOOH. The formed adduct products may participate in the formation and growth processes of organic new particle in the atmosphere.

**Table R1** Saturated vapour pressure ($P^0$) of some selected compounds predicted by using the Nan-Nan and EVAPORATION methods

| Compounds | Nan-Nan (Pa) | EVAPORATION (Pa) | experimental data (Pa) |
|---|---|---|---|
| methanol | $5.81 \times 10^5$ | $1.58 \times 10^4$ | $1.67 \times 10^4$ |
| ethanol | $2.27 \times 10^4$ | $8.12 \times 10^3$ | $7.96 \times 10^3$ |
| isoprene | $3.51 \times 10^5$ | $7.35 \times 10^4$ | $7.33 \times 10^4$ |
| cyclohexene | $5.61 \times 10^4$ | $1.15 \times 10^4$ | $1.30 \times 10^4$ |
| n-hexane | $1.41 \times 10^5$ | $2.00 \times 10^4$ | $2.02 \times 10^4$ |
| n-heptane | $4.55 \times 10^4$ | $6.12 \times 10^3$ | $6.09 \times 10^3$ |
| methylbenzene | $2.17 \times 10^4$ | $3.16 \times 10^3$ | $3.79 \times 10^3$ |

**Table S8** Predicted saturated vapour pressure ($P^0$) and saturated concentrations ($c^0$) for the adduct products of the successive reactions of SCIs with HCOOH

|  | formula | $P^0$ (atm) | $c^0$ (ug/m$^3$) |
|---|---|---|---|
| n CH$_2$OO + HCOOH | | | |
| n = 1 | HC(O)OCH$_2$OOH | $2.77 \times 10^{-2}$ | $1.03 \times 10^8$ |
| n = 2 | HC(O)O(CH$_2$OO)$_2$H | $9.73 \times 10^{-4}$ | $5.42 \times 10^6$ |
| n = 3 | HC(O)O(CH$_2$OO)$_3$H | $3.41 \times 10^{-5}$ | $2.53 \times 10^5$ |
| n = 4 | HC(O)O(CH$_2$OO)$_4$H | $1.20 \times 10^{-6}$ | $1.11 \times 10^4$ |
| n = 5 | HC(O)O(CH$_2$OO)$_5$H | $4.19 \times 10^{-8}$ | $4.67 \times 10^2$ |
| n $anti$-CH$_3$CHOO + HCOOH | | | |
| n = 1 | HC(O)OCH(CH$_3$)OOH | $1.44 \times 10^{-2}$ | $6.15 \times 10^7$ |

| | | | |
|---|---|---|---|
| n = 2 | $HC(O)O(CH(CH_3)OO)_2H$ | $2.61 \times 10^{-4}$ | $1.75 \times 10^{6}$ |
| n = 3 | $HC(O)O(CH(CH_3)OO)_3H$ | $4.73 \times 10^{-6}$ | $4.32 \times 10^{4}$ |
| n = 4 | $HC(O)O(CH(CH_3)OO)_4H$ | $8.59 \times 10^{-8}$ | $9.92 \times 10^{2}$ |
| n = 5 | $HC(O)O(CH(CH_3)OO)_5H$ | $1.56 \times 10^{-9}$ | $2.18 \times 10^{1}$ |
| n $syn$-$CH_3CHOO$ + HCOOH | | | |
| n = 1 | $HC(O)OCH(CH_3)OOH$ | $1.44 \times 10^{-2}$ | $6.15 \times 10^{7}$ |
| n = 2 | $HC(O)O(CH(CH_3)OO)_2H$ | $2.61 \times 10^{-4}$ | $1.75 \times 10^{6}$ |
| n = 3 | $HC(O)O(CH(CH_3)OO)_3H$ | $4.73 \times 10^{-6}$ | $4.32 \times 10^{4}$ |
| n = 4 | $HC(O)O(CH(CH_3)OO)_4H$ | $8.59 \times 10^{-8}$ | $9.92 \times 10^{2}$ |
| n = 5 | $HC(O)O(CH(CH_3)OO)_5H$ | $1.56 \times 10^{-9}$ | $2.18 \times 10^{1}$ |
| n $(CH_3)_2COO$ + HCOOH | | | |
| n = 1 | $HC(O)OC(CH_3)_2OOH$ | $1.86 \times 10^{-3}$ | $9.02 \times 10^{6}$ |
| n = 2 | $HC(O)O(C(CH_3)_2OO)_2H$ | $4.38 \times 10^{-5}$ | $3.43 \times 10^{5}$ |
| n = 3 | $HC(O)O(C(CH_3)_2OO)_3H$ | $1.03 \times 10^{-6}$ | $1.11 \times 10^{4}$ |
| n = 4 | $HC(O)O(C(CH_3)_2OO)_4H$ | $2.42 \times 10^{-8}$ | $3.35 \times 10^{2}$ |
| n = 5 | $HC(O)O(C(CH_3)_2OO)_5H$ | $5.70 \times 10^{-10}$ | $9.57 \times 10^{0}$ |

Corresponding descriptions have been added in the page 26 line 649-681 of the revised manuscript:

*The saturated vapour pressure ($P^0$) of the adduct products formed from the successive reactions of SCIs with HCOOH is estimated by using the EVAPORATION method proposed by Compernolle et al. (2011), and the room temperature results are summarized in Table S8. This table shows that the $P^0$ of the adduct products decreases significantly as the number of SCIs is increased. Notably, the $P^0$ of the adduct products decreases when the size of SCIs increases. For example, the $P^0$ of the adduct product $HC(O)O(CH_2OO)_3H$ in the $nCH_2OO$ + HCOOH reaction is estimated to be $3.41 \times 10^{-5}$ atm, which is greater than those of the corresponding adduct products in the $nanti$-$CH_3CHOO$ + HCOOH ($4.73 \times 10^{-6}$ atm), $nsyn$-$CH_3CHOO$ + HCOOH ($4.73 \times 10^{-6}$ atm), and $n(CH_3)_2COO$ + HCOOH ($1.03 \times 10^{-6}$ atm) reactions by 7.21, 7.21 and 33.11 times, respectively.*

*A classify scheme of various organic compounds is based on their volatility, as presented by Donahue et al. (2012) The volatility of organic compounds is described by their effective saturation concentrations. The saturated concentrations ($c^0$) of the adduct products formed from the successive reactions of SCIs with HCOOH are listed in Table S8. As shown in Table S8, the $c^0$ of the adduct products decrease significantly as the number of SCIs is increased. It deserves mentioning that the $c^0$ of the adduct products decrease with increasing the size of SCIs. For the*

*nCH₂OO + HCOOH reaction, the $c^0$ of the adduct products are estimated to be $1.03 \times 10^8$ (n=1), $5.42 \times 10^6$ (n=2), $2.53 \times 10^5$ (n=3), $1.11 \times 10^4$ (n=4) and $4.67 \times 10^2$ (n=5) ug/m³, respectively. According to the Volatility Basis Set (VBS) of organic compounds (Donahue et al., 2012), the adduct products belong to volatile organic compounds (VOC, $c^0 > 3 \times 10^6$ ug/m³) when the number of SCIs is less than or equal to two, while they belong to intermediate volatility organic compounds (IVOC, $300 < c^0 < 3 \times 10^6$ ug/m³) when the number of SCIs is greater than or equal to three. Similarly, the adduct products in the nanti-CH₃CHOO + HCOOH, nsyn-CH₃CHOO + HCOOH, and n(CH₃)₂COO + HCOOH reactions belong to IVOC when the number of SCIs ranges from 2 to 4, whereas they belong to semivolatile organic compounds (SVOC, $0.3 < c^0 < 300$ ug/m³) when the number of SCIs is equal to 5. Based on the above discussions, it can be concluded that the volatility of the adduct products is significantly affected by the number and size of SCIs in the successive reaction of SCIs with HCOOH. The formed adduct products may participate in the formation and growth processes of organic new particle in the atmosphere.*

6. Finally, the Conclusion to the revised manuscript should contain some discussion of the atmospheric significance of the reactions they have considered. Key points to address are the very small pseudo-first-order rate constants for the CI + hydroperoxy ester reaction and the likelihood of oligomers of the dimethyl CI to be IVOCs.

**Response:** Based on the Reviewer's suggestion, the effective pseudo-first-order rate constants for the reactions of SCIs with hydroperoxide ester and the saturated vapour pressure and saturated concentration of the formed oligomers have been added in the Conclusion of the revised manuscript.

(e) In the tropical forest environments, the effective pseudo-first-order rate constants for the reactions of distinct SCIs with HPMF ($k_{\text{eff(SCIs+HPMF)}}$) are several orders of magnitude lower than those for the reactions of distinct SCIs with H₂O ($k_{\text{eff(SCIs+H2O)}}$), HCOOH ($k_{\text{eff(SCIs+HCOOH)}}$) and SO₂ ($k_{\text{eff(SCIs+SO2)}}$). $k_{\text{eff(SCIs+H2O)}}$ is greater than $k_{\text{
[revised manuscript text omitted]